# UltraLightUNet: Rethinking U-shaped Network with Multi-kernel Lightweight Convolutions for Medical Image Segmentation

## Abstract

In this paper, we introduce UltraLightUNet (2D and 3D), an ultra-lightweight, multi-kernel U-shaped network for medical image segmentation. The core of UltraLightUNet consists of a new Multi-kernel Inverted Residual (MKIR) block, which can efficiently process images through multiple kernels while capturing complex spatial relationships. Additionally, our Multi-kernel Inverted Residual Attention (MKIRA) block refines and emphasizes image salient features via sophisticated convolutional multi-focal attention mechanisms. UltraLightUNet strategically employs the MKIR block in the encoder for feature extraction and the MKIRA block in the decoder for feature refinement, thus ensuring targeted feature enhancement at each stage. With only 0.316M #Params and 0.314G #FLOPs, UltraLightUNet offers an ultra-lightweight yet powerful segmentation solution that outperforms state-of-the-art (SOTA) methods across twelve medical imaging benchmarks. Notably, UltraLightUNet surpasses TransUNet on DICE score while using $333\times$ fewer #Params and $123\times$ fewer #FLOPs. Compared to the lightweight model, UNeXt, UltraLightUNet improves DICE scores by up to 6.7% with $4.7\times$ fewer parameters. UltraLightUNet also outperforms recent lightweight models such as MedT, CMUNeXt, EGE-UNet, Rolling-UNet, and UltraLight_VM_UNet, while using significantly fewer #Params and #FLOPs. Furthermore, our 3D version, UltraLightUNet3D-M (1.42M #Params and 7.1G #FLOPs), outperforms SwinUNETR (62.19M #Params, 328.6G #FLOPs) and nn-UNet (31.2M #Params, 110.4G #FLOPs) on the FETA, MSD Brain Tumor, Prostate, and Lung Cancer segmentation benchmarks. This remarkable performance, combined with substantial computational gains, makes UltraLightUNet an ideal solution for real-time and point-of-care services in resource-constrained environments. We will make the code publicly available upon paper acceptance.

## 1 Introduction

The field of medical image segmentation has been revolutionized through the development of U-shaped convolutional neural network (CNN) architectures (Ronneberger et al., 2015; Oktay et al., 2018; Zhou et al., 2018; Fan et al., 2020) such as UNet (Ronneberger et al., 2015), ResUNet (Zhang et al., 2018), UNet++ (Zhou et al., 2018), AttnUNet (Oktay et al., 2018), PraNet (Fan et al., 2020), UACANet (Kim et al., 2021), DeepLabv3+ (Chen et al., 2017), and ACC-UNet (Ibtehaz & Kihara, 2023). These models excel at segmenting medical images, thus enabling precise segmentation of critical areas like tumors, lesions, or polyps. The attention mechanisms (Oktay et al., 2018; Fan et al., 2020; Woo et al., 2018) integrated into these architectures help refine the feature maps, thus enhancing pixel-level classification. However, the substantial computational demands of these models, including those with attention mechanisms, limit their applicability in resource-constrained environments such as point-of-care diagnostics.

The introduction of vision transformers (Chen et al., 2021; Cao et al., 2021; Rahman & Marculescu, 2023b; Valanarasu et al., 2021), including TransUNet (Chen et al., 2021), SwinUNet (Cao et al., 2021), MedT (Valanarasu et al., 2021), EMCAD (Rahman et al., 2024), and DeformableLKA (Azad et al., 2024), marked a shift towards leveraging self-attention to capture long-range dependencies within images for a comprehensive global view. However, transformers tend to neglect crucial local

spatial relationships among pixels which are essential for precise segmentation. Moreover, transformers usually have high memory and computational demands for calculation and fusing attention with convolutional mechanisms, which limits their practical uses.

In recent years, a good number of lightweight architectures such as MobileNets (Howard et al., 2017; Sandler et al., 2018), UNeXt (Valanarasu & Patel, 2022), CMUNeXt (Tang et al., 2023), MALUNet (Ruan et al., 2022), EGE-UNet (Ruan et al., 2023), Rolling-UNet (Liu et al., 2024), and UltraLight VM-UNet (Wu et al., 2024), helped bridge this gap by combining the strengths of CNNs and/or multi-layer perceptrons (MLPs). However, most of these architectures are designed for less complex or easy-to-segment applications such as skin lesions, breast cancer with ultrasound, and microscopic cell nuclei/structure segmentation. Consequently, these architectures show poor performance in challenging applications like polyp segmentation due to the high variability in the shape, size, and texture of polyps.

Aiming to improve segmentation performance and accuracy, several 3D medical image segmentation networks have been also introduced, such as 3D U-Net (Çiçek et al., 2016), SwinUNETR (Hatamizadeh et al., 2021), 3D UX-NET (Lee et al., 2022), UNETR (Hatamizadeh et al., 2022), nn-UNet (Isensee et al., 2021), and nn-Former (Zhou et al., 2021). However, the high computational demands (particularly the large #FLOPs and significant memory consumption) of these 3D networks, make it challenging to implement them in clinical settings. Recently, SlimUNETER (Pang et al., 2023) introduces a lightweight architecture, however, with the cost of significantly low segmentation accuracy. These limitations highlight the need for extremely lightweight models that can deliver highly accurate segmentation while being practical for use in real-time, particularly in resource-constrained settings.

Aiming to extreme lightweight efficiency for both 2D and 3D medical image segmentation in resource-constraint settings, we design UltraLightUNet, a significant breakthrough in medical image segmentation, which leverages *multi-kernel (i.e., $k_1 = k_2$ or $k_1 \neq k_2$ for $k_1, k_2 \in Kernels$) lightweight convolutions*. Our lightweight convolution blocks drastically reduce the computational cost, making the network ultra-lightweight without sacrificing the ability to capture detailed features within an image. Additionally, our multi-kernel property enables the model to effectively handle feature representations at *same or varying* receptive fields, thus allowing for a more robust and comprehensive analysis of complex images in diverse applications. Moreover, by incorporating sophisticated convolutional multi-focal attention mechanisms *only* in our decoder further refines the feature maps by capturing the image salient features. We note that our network is effective for segmentation in both scenarios, whether the regions of interest vary significantly in size and shape or remain relatively uniform. By integrating these new ideas, UltraLightUNet achieves a fine balance between computational efficiency and segmentation accuracy, thus offering an ultra-lightweight model that not only surpasses the performance of heavyweight counterparts (in DICE scores), but it does so with significantly fewer #Params and #FLOPs. Our contributions are as follows:

- **New Ultra Lightweight UNet:** We propose a new end-to-end network, UltraLightUNet, for *both 2D and 3D* medical image segmentation, which encodes an image using lightweight multi-kernel convolutions. UltraLightUNet also progressively refines the multi-resolution spatial representations using multi-kernel convolutional attention. Of note, our UltraLightUNet-T has only 0.027M and 0.062G #Params and #FLOPs, respectively, yet provides SOTA performance. Moreover, UltraLightUNet has only 0.316M #Params and 0.314G #FLOPs. The extremely low model size (#Params) and computations (#FLOPs) make our UltraLightUNet easy to deploy in point-of-care diagnostics or resource-constraint environments (e.g., mobile or edge devices).

- **Lightweight Multi-kernel Inverted Residual:** We introduce MKIR, a new Multi-Kernel Inverted Residual block that performs depth-wise convolutions with multiple kernels (i.e., $k_1 = k_2$ or $k_1 \neq k_2$, for $k_1, k_2 \in Kernels$). Our encoder extracts features using the MKIR block; this choice is motivated by the need to efficiently process and encode diverse and complex structures in medical images, thus providing a rich representation with minimal computational costs.

- **Lightweight Multi-kernel Inverted Residual Attention:** We propose Multi-Kernel Inverted Residual Attention (MKIRA), a new block to refine and enhance multi-scale salient features by suppressing irrelevant regions. In our decoder, MKIRA enhances features discrimination by focusing on key feature channels and highlighting the important spatial

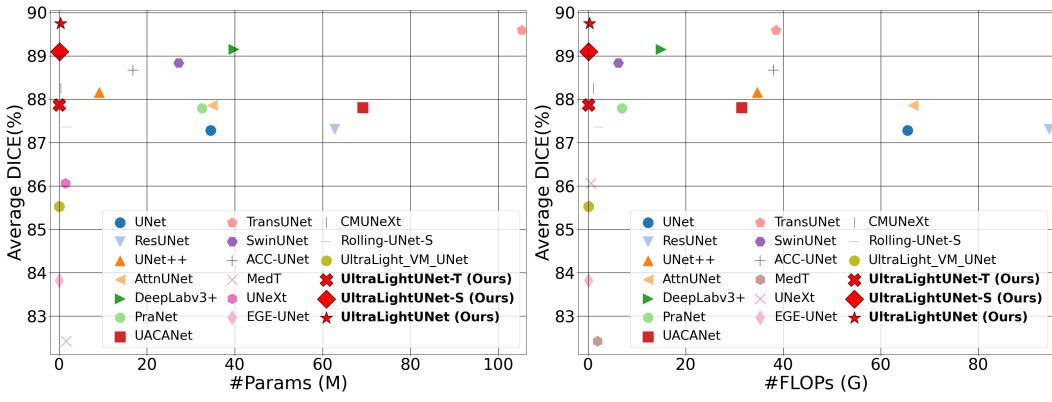

(a) Average DICE scores vs. #Params (b) Average DICE scores vs. #FLOPs

Figure 1: Comparison of our UltraLightUNet against different SOTA methods over six binary medical image segmentation datasets. As shown, UltraLightUNet has the third lowest #Params and #FLOPs (behind EGE-UNet (Ruan et al., 2023), UltaLight_VM_UNet (Wu et al., 2024)), yet the highest DICE scores. However, our UltraLightUNet-T achieves significantly better DICE score than EGE-UNet and UltraLight_VM_UNet, with much lower #Params and comparable #FLOPs.

      regions in an image. This ensures that the decoder can reconstruct precise and accurate segmentation maps by focusing only on the most critical aspects of the encoded features.

- **Improved Performance across Various Datasets:** Our experiments evident that Ultra-LightUNet significantly improves the performance of medical image segmentation compared to SOTA models with an extreme efficiency (as shown in Fig. 1) on twelve medical image segmentation datasets (e.g., BUSI, ClinicDB, ColonDB, ISIC2018, DSB2018, EM, Synapse, ACDC, FETA, MSD BrainTumour, Prostate, and Lung) that belong to ten different segmentation tasks (e.g., breast cancer, polyp, skin lesion, cell, abdomen organs, cardiac organs, brain organs, brain tumor, prostate, and lung cancer).

The remaining of this paper is organized as follows: Section 2 summarizes related work. Section 3 describes our proposed method. Section 4 explains our experimental setup and results on multiple medical image segmentation benchmarks. Section 5 covers different ablation experiments. Lastly, Section 6 concludes the paper by summarizing our findings.

## 2 RELATED WORK

### 2.1 CONVOLUTIONAL NEURAL NETWORKS (CNNS)

The advent of CNNs marks a significant shift in medical image segmentation (Ronneberger et al., 2015; Oktay et al., 2018; Zhou et al., 2018; Fan et al., 2020; Kim et al., 2021; Chen et al., 2017; Ibtehaz & Kihara, 2023). Pioneering works such as Fully Convolutional Networks (FCNs) (Long et al., 2015) laid the foundation for end-to-end segmentation models. FCNs replace fully connected layers with convolutional layers, thus enabling pixel-wise predictions and efficient learning of spatial hierarchies in images. Afterward, U-Net (Ronneberger et al., 2015) became a *key model* in medical image segmentation due to its encoder-decoder architecture with skip connections. U-Net effectively combines the high-resolution features from the encoder with the context information from the decoder, hence leading to precise segmentations even with limited training data. The sophisticated design of U-shaped architecture for pixel-level segmentation tasks motivates us to choose the U-shaped design in our proposed network.

U-Net's success has inspired numerous variants and improvements. Inspired by residual learning in ResNet (He et al., 2016), ResUNet (Zhang et al., 2018) employs residual blocks to facilitate gradient flow and improve convergence, addressing the vanishing gradient problem in deep networks. Zhou et al. (2018) introduce UNet++, which uses nested and dense skip connections to further enhance the feature propagation and improve the segmentation accuracy. AttnUNet (Oktay et al., 2018) incorporates attention mechanisms that focus on the relevant regions in the feature maps, thus enhancing the segmentation performance by suppressing irrelevant background noise. Fan et al. (2020) introduce PraNet for precise polyp segmentation by employing parallel reverse attention and edge-guidance to

refine segmentation boundaries. UACANet (Kim et al., 2021) leverages uncertainty-aware mechanisms to improve the reliability and robustness of segmentation outcomes. DeepLabv3+ (Chen et al., 2017) integrates atrous convolutions and spatial pyramid pooling to capture multi-scale context information. ACC-UNet (Ibtehaz & Kihara, 2023) employs adaptive context capture mechanisms to dynamically adjust the receptive fields based on the input image.

## 2.2 Vision Transformers

Vision Transformers (ViTs) (Dosovitskiy et al., 2020; Liu et al., 2021) have emerged as a powerful alternative to CNNs, i.e., a new paradigm for medical image analysis tasks that leverages the self-attention mechanism (Chen et al., 2021; Cao et al., 2021; Rahman et al., 2024; Rahman & Marculescu, 2023a; Valanarasu et al., 2021). Moreover, by combining the strengths of CNNs for local feature extraction and Transformers for capturing long-range dependencies, TransUNet (Chen et al., 2021) achieves superior performance in medical image segmentation. SwinUNet (Cao et al., 2021) is introduced based on the Swin Transformer (Liu et al., 2021) architecture, which utilizes shifted windows to achieve hierarchical feature representation, enabling efficient computation. Rahman et al. introduces, CASCADE (Rahman & Marculescu, 2023a), a cascaded-attention decoding network using standard convolutions. Recently, EMCAD (Rahman et al., 2024) introduces a depthwise convolutions-based multi-scale decoder. Although, CASCADE and EMCAD perform well in medical image segmentation, their segmentation accuracy and computational complexity solely depend on the strength and complexity of the exiting pretrained transformer encoder they use, thus making them less suitable for resource-constrained settings. In contrast, we propose to design an extremely efficient (ultra-lightweight) end-to-end (both encoder and decoder) architecture using the multi-kernel trick (where, $k_1 = k_2$ or $k_1 \neq k_2$ for $k_1, k_2 \in Kernels$) with depthwise convolutions.

## 2.3 Lightweight Networks

Recent efforts have focused on making CNNs more efficient for real-time and resource-constrained environments. MobileNets (Howard et al., 2017) and EfficientNets (Tan & Le, 2019) introduce depthwise separable convolutions and compound scaling, respectively, to create lightweight models with competitive performance. Additionally, several novel lightweight architectures have been developed to further enhance the efficiency of medical image segmentation (Valanarasu & Patel, 2022; Tang et al., 2023; Ruan et al., 2023; Liu et al., 2024; Yang et al., 2023; Lin et al., 2023b). UNeXt (Valanarasu & Patel, 2022) leverages hybrid convolutional and transformer blocks to capture both local and global features efficiently, improving segmentation accuracy while maintaining computational efficiency. CMUNeXt (Tang et al., 2023) combines convolutional and multi-scale features to enhance segmentation performance. EGE-UNet (Ruan et al., 2023) integrates edge-guided mechanisms to refine segmentation boundaries. Rolling-UNet (Liu et al., 2024) incorporates rolling convolutional blocks to enhance the model's ability to capture long-range dependencies.

## 2.4 3D Networks

Recent advancements in 3D medical image segmentation have introduced several techniques to improve performance, though many face challenges related to computational and memory efficiency. 3D U-Net (Çiçek et al., 2016) is a widely-used U-shaped network, but suffers from high #FLOPs and memory usage. nn-UNet (Isensee et al., 2021) automates the architecture optimization for specific datasets, but still remains resource-intensive. Transformer-based models like nn-Former (Zhou et al., 2021), UNETR (Hatamizadeh et al., 2022), and SwinUNETR (Hatamizadeh et al., 2021) capture global dependencies, but are computationally heavy. Recently, mimicking the large-kernel depthwise convolutions in ConvNeXt (Liu et al., 2022) for global feature representation (like transformers), 3D UX-Net (Lee et al., 2022) introduces a lightweight encoder using the 3D large-kernel depthwise convolutions to improve global feature learning. However, 3D UX-Net has high computational costs due to using the existing heavyweight UNETR decoder (Hatamizadeh et al., 2022). Recently, SlimUNETR (Azad et al., 2024) introduces a lightweight 3D network for volumetric segmentation. However, SlimUNETR performs poorly on complex datasets. In contrast, we propose to design an ultra-lightweight 3D network without compromising the segmentation accuracy.

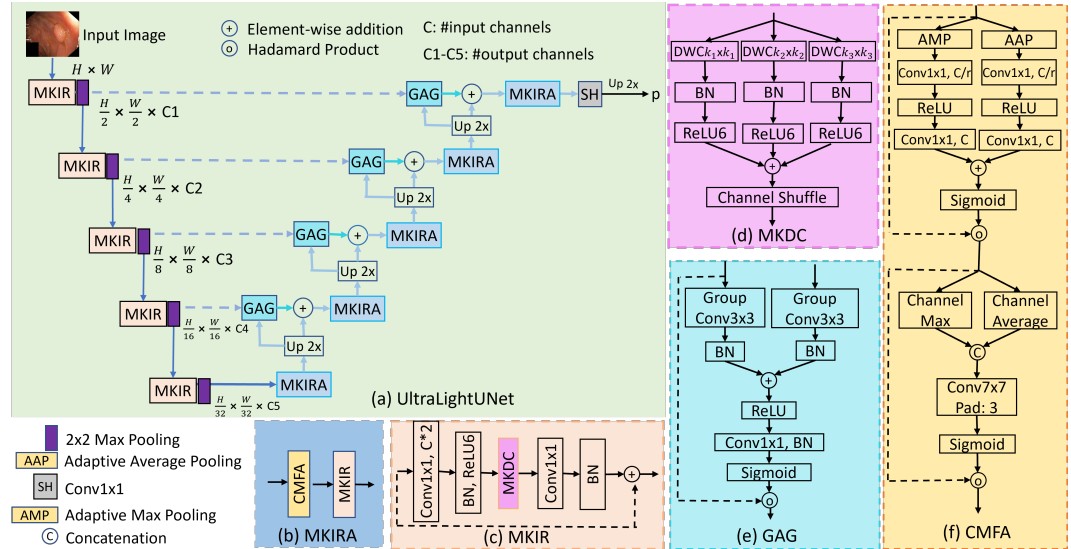

Figure 2: The proposed network. (a) UltraLightUNet network (b) Multi-kernel inverted residual attention (MKIRA), (c) Multi-kernel inverted residual (MKIR), (d) Multi-kernel (parallel) depth-wise convolution (MKDC), (e) Grouped attention gate (GAG), (f) Convolutional multi-focal attention (CMFA). We propose the 3D version of all the above modules (b-f) for volumetric context extraction, and design our UltraLightUNet3D network for volumetric segmentation.

## 3 METHOD

We introduce next our Multi-Kernel Inverted Residual (MKIR), Convolutional Multi-focal Attention (CMFA), Multi-Kernel Inverted Residual Attention (MKIRA) and Grouped Attention Gate (GAG) blocks. Then, we introduce our complete UltraLightUNet architecture by integrating these new blocks into the UNeXt (Valanarasu & Patel, 2022) (Fig. 2a in green box).

### 3.1 MULTI-KERNEL INVERTED RESIDUAL (MKIR)

We first introduce the multi-kernel inverted residual ($MKIR$) block to generate and refine feature maps (Fig. 2c). By utilizing multiple (same or different) kernel sizes, $MKIR$ allows for better understanding of both fine-grained details and broader contexts, thereby enabling a comprehensive representation of the input. As shown in Fig. 2c, the process begins by expanding the #channels (i.e., expansion_factor = 2) through point-wise convolution $PWC_1$, batch normalization $BN$ (Ioffe & Szegedy, 2015), and $ReLU6$ activation (Krizhevsky & Hinton, 2010). This is followed by multi-kernel depth-wise convolution $MKDC$ for capturing application-specific complex spatial contexts. A subsequent point-wise convolution $PWC_2$ and $BN$ restore the original #channels. The MKIR (Equation 1) significantly reduces the computational cost while ensuring rich feature representation:

$$MKIR(x) = BN(PWC_2(MKDC(ReLU6(BN(PWC_1(x))))))$$ (1)

where $MKDC$ for multiple kernels ($K$) is defined in Equation 2 and Fig. 2d:

$$MKDC(x) = CS(\sum_{k \in K} DWCB_k(x))$$ (2)

where $DWCB_k(x) = ReLU6(BN(DWC_k(x)))$. Here, $DWC_k(.)$ is a depth-wise convolution with the kernel $k \times k$. To address the channel independence in depth-wise convolution, a channel shuffle ($CS$) is used to ensure the inter-channel information flow. Our $MKDC(.)$ differs from $MSCB$ (in EMCAD (Rahman et al., 2024)) in their core theoretical concepts. Our multi-kernel trick supports both $k_1 = k_2$ (same-size kernels) and $k_1 \neq k_2$ (different-size kernels) for $k_1, k_2 \in K$ versus conventional multi-scale (only $k_1 \neq k_2$) designs (Rahman et al., 2024; Lin et al., 2023a; Seo et al., 2022), thus allowing adaptable context extraction. This conceptual distinction allows UltraLightUNet to adapt kernel sizes based on application-specific needs (e.g., large kernels for large objects, small kernels for small objects, or mixed for both objects segmentation).

### 3.2 CONVOLUTIONAL MULTI-FOCAL ATTENTION (CMFA)

Our CMFA (Fig. 2f) leverages a unified attention mechanism that effectively captures both channel-wise and spatial features (Rahman & Marculescu, 2023a), thereby optimizing the network ability to focus on critical aspects of the image while suppressing irrelevant details. We first enhance the relevant channels by applying both adaptive max pooling ($AMP$) and average pooling ($AAP$) to condense spatial information, which allows the network to maintain robustness to variations in local structures. The pooled outputs are then passed through a series of point-wise convolutions ($PWC$) to reduce ($r = 16$) dimensions and are activated by $ReLU$ (Nair & Hinton, 2010), followed by a second $PWC$ layer for expansion. The $Sigmoid$ activation generates the attention weights, which are then multiplied element-wise ($\circledast$) with the input, thus emphasizing the important channels. This attention process is defined in Equation 3:

$$CA(x) = Sigmoid(PWC_2(ReLU(PWC_1(AMP(x)))) + PWC_2(ReLU(PWC_1(AAP(x))))) \circledast x \tag{3}$$

Subsequently, to capture the spatial dependencies and refine the feature maps further, we apply pooling operations across channels to generate two spatial descriptors: $Channel_{max}(x)$ and $Channel_{avg}(x)$. By applying a large-kernel convolution ($LKC$) to the concatenated pooled values, we capture contextual relationships across a broader spatial context, thus reinforcing the network focus on important regions of an image. The refined feature maps are derived as Equation 4:

$$SA(x) = Sigmoid(LKC([Channel_{max}(x), Channel_{avg}(x)])) \circledast x \tag{4}$$

In essence, combining both mean and max pooling helps balance the focus between high-intensity (max) regions and overall feature consistency (mean). Similarly, the integration of both channel and spatial attention facilitates precise reconstruction and segmentation, even in complex scenarios, thereby leading to improved segmentation performance.

### 3.3 MULTI-KERNEL INVERTED RESIDUAL ATTENTION (MKIRA)

Our new MKIRA block (Fig. 2b) effectively refines the feature maps by leveraging a convolutional multi-focal attention mechanism ($CMFA$) and a multi-kernel inverted residuals ($MKIR$). The use of $CMFA$ enhances the network ability to focus on critical channels and spatial regions, thereby ensuring that the most salient features are enhanced (see the CMFA activation heatmaps in Fig. 3 in Appendix). This dual attention mechanism aids in improving feature discrimination and representation, especially in challenging scenarios where important structures may vary significantly. Additionally, the incorporation of the $MKIR$ block further enriches the feature maps by capturing contextual relationships through multiple kernels (see the MKIRA activation heatmaps in Fig. 3). Taken together, these components enable the network to maintain high accuracy while minimizing the computational overhead. $MKIRA$ is given in Equation 5:

$$MKIRA(x) = MKIR(CMFA(x)) \tag{5}$$

### 3.4 GROUPED ATTENTION GATE (GAG)

We use a grouped attention gate ($GAG$, Fig. 2e) that mixes the feature maps with the attention coefficients for enhancing the relevant features and suppressing the irrelevant ones. By utilizing a gating signal from higher-resolution features, $GAG$ directs the information flow, thus improving medical image segmentation accuracy. Unlike Attention UNet (Oktay et al., 2018), which processes signals with $1 \times 1$ convolution, our method applies $3 \times 3$ group convolutions to both gating ($g$) and input ($x$) feature maps separately (Rahman et al., 2024). After convolution, the features undergo batch normalization ($BN$) and get combined via addition, followed by $ReLU$ activation. Subsequently, a $1 \times 1$ convolution and batch normalization ($BN$) produce a unified feature map which, after the $Sigmoid$ activation ($\sigma$), generates the attention coefficients. These coefficients adjust the input feature $x$, and create an attention-enhanced output. $GAG$ is defined in Equations 6:

$$GAG(g, x) = x \circledast \sigma(BN(Conv(ReLU(BN(GroupConv_g(g) + BN(GroupConv_x(x)))))))) \tag{6}$$

### 3.5 ULTRALIGHTUNET

Our complete UltraLightUNet architecture employs multi-kernel convolutions across five encoding and decoding stages to generate high-resolution segmentation maps, as depicted in Fig. 2a. Each encoding stage uses an MKIR block to produce $C_i$ feature maps, followed by max pooling for down-sampling while retaining crucial information. The output from the final encoding (bottleneck) stage passes through an MKIRA block in the decoder initial stage, significantly refining the feature maps

Table 1: Results of binary (breast cancer, skin lesion, polyp, and cell) segmentation. We reproduce the results of SOTA methods using their publicly available implementations with our 80:10:10 train-val-test splits. FLOPs of all the methods are reported for $256 \times 256$ inputs. The FLOPs of all methods for polyp segmentation with $352 \times 352$ inputs will be higher. We report the DICE scores (%) averaging over five runs, thus having 1-4% standard deviations. Best results are shown in bold.

| Architecture | Pretrain | #Params | FLOPs | BUSI | ISIC18 | Polyp | | Cell | | Avg. |
| --- | --- | --- | --- | --- | --- | --- | --- | --- | --- | --- |
| | | | | | | Clinic | Colon | DSB18 | EM | |
| UNet (Ronneberger et al., 2015) | No | 34.53M | 65.53G | 74.04 | 86.67 | 91.43 | 83.95 | 92.23 | 95.36 | 87.28 |
| ResUNet (Zhang et al., 2018) | No | 62.74M | 94.56G | 74.12 | 86.75 | 91.46 | 84.02 | 92.16 | 95.32 | 87.31 |
| UNet++ (Zhou et al., 2018) | No | 9.16M | 34.65G | 74.76 | 87.46 | 91.52 | 87.88 | 91.97 | 95.38 | 88.16 |
| AttnUNet (Oktay et al., 2018) | No | 34.88M | 66.64G | 74.48 | 87.05 | 91.50 | 86.46 | 92.22 | 95.45 | 87.86 |
| DeepLabv3+ (Chen et al., 2017) | Yes | 39.76M | 14.92G | 76.81 | 88.64 | 92.46 | 89.86 | 92.14 | 94.96 | 89.15 |
| PraNet (Fan et al., 2020) | Yes | 32.55M | 6.93G | 75.14 | 88.46 | 91.71 | 89.16 | 89.89 | 92.37 | 87.79 |
| UACANet (Kim et al., 2021) | Yes | 69.16M | 31.51G | 76.96 | 88.72 | 93.29 | 89.76 | 88.86 | 89.28 | 87.81 |
| TransUNet (Chen et al., 2021) | Yes | 105.32M | 38.52G | 78.01 | **89.04** | 93.18 | 89.97 | 92.04 | 95.27 | 89.59 |
| SwinUNet (Cao et al., 2021) | Yes | 27.17M | 6.2G | 77.38 | 88.66 | 92.42 | 89.07 | 91.03 | 94.47 | 88.84 |
| ACC-UNet (Ibtehaz & Kihara, 2023) | No | 16.8M | 38.0G | 77.02 | 88.57 | 92.56 | 89.13 | 90.05 | 94.67 | 88.67 |
| DeformableLKA (Azad et al., 2024) | No | 102.76M | 26.03G | 74.62 | 88.17 | 91.05 | 85.93 | 92.12 | 94.45 | 87.72 |
| Rolling-UNet-S (Liu et al., 2024) | No | 1.78M | 2.1G | 76.38 | 87.35 | 90.23 | 82.48 | 92.50 | 95.23 | 87.36 |
| MedT (Valanarasu et al., 2021) | No | 1.57M | 1.95G | 69.23 | 86.78 | 83.44 | 68.90 | 92.28 | 93.87 | 82.42 |
| UNeXt (Valanarasu & Patel, 2022) | No | 1.47M | 0.57G | 74.71 | 87.78 | 90.20 | 83.84 | 86.01 | 93.81 | 86.06 |
| CMUNeXt (Tang et al., 2023) | No | 0.418M | 1.09G | 77.34 | 87.51 | 92.82 | 83.85 | 92.58 | 95.38 | 88.25 |
| **UltraLightUNet (Ours)** | No | 0.316M | 0.314G | **78.04** | 88.74 | **93.48** | **90.01** | 92.71 | 95.52 | **89.75** |
| **UltraLightUNet-S (Ours)** | No | 0.093M | 0.125G | 77.26 | 88.57 | 92.31 | 88.78 | 92.45 | 95.22 | 89.10 |
| EGE-UNet (Ruan et al., 2023) | No | 0.054M | 0.072G | 71.34 | 86.95 | 84.76 | 76.03 | 90.10 | 93.76 | 83.82 |
| UltraLight_VM_UNet (Wu et al., 2024) | No | 0.050M | 0.060G | 72.31 | 87.85 | 87.11 | 80.06 | 91.88 | 93.96 | 85.53 |
| **UltraLightUNet-T (Ours)** | No | **0.027M** | **0.062G** | 75.64 | 88.19 | 91.26 | 85.03 | 92.38 | 94.69 | 87.87 |

by emphasizing and grouping relevant pixels. These are then upsampled using bilinear interpolation for subsequent decoding stages. Decoder stages integrate skip-connections with refined features using a GAG followed by additive aggregation. The resultant feature maps are refined through the MKIRA block and up-sampled (only bilinear $2\times$, no convolutions) to align with the later stages.

The segmentation head (SH) at the last stage outputs the segmentation map $p$. We obtain the final segmentation output by employing a $Sigmoid$ on $p$ for binary segmentation or a $Softmax$ for multi-class segmentation. We optimize the loss of only the prediction $p$ for all segmentation tasks.

We directly extended our UltraLightUNet architecture to UltraLightUNet3D for 3D medical image segmentation by replacing all 2D operations (such as Conv2d, BatchNorm2d, and channel shuffle) with their corresponding 3D operations. This adaptation allows the network to capture the volumetric features and spatial relationships inherent in 3D medical imaging data. Therefore, our MKDC is extended to MKDC3D, thus enabling our network to process complex volumetric features more efficiently across different layers in a 3D space. Additionally, attention mechanisms such as channel attention (CA) and spatial attention (SA) are adapted to focus on relevant 3D regions to enhance feature discrimination and segmentation accuracy.

Note: The different versions of our network (UltraLightUNet-T (Tiny), UltraLightUNet-S (Small), UltraLightUNet (Base), UltraLightUNet-M (Medium), and UltraLightUNet-L (Large)) are distinguished by the number of channels used in the five stages of our network (see Tables 9, and 10).

## 4 EXPERIMENTS AND RESULTS

The implementation details, binary segmentation results, multi-class segmentation results, and 3D segmentation results are described below. **The dataset description, evaluation metrics, and more results including qualitative are provided in the Appendix A.1, A.2, and A.9-A.13, respectively.**

### 4.1 IMPLEMENTATION DETAILS

Our networks are developed and evaluated using Pytorch 1.11.0, operating on a single NVIDIA RTX A6000 GPU equipped with 48GB of RAM. We utilize multi-scale kernels $[1, 3, 5]$ within our MKDC, based on an ablation study. The architecture employs a series of parallel depth-wise convolutions in the UltraLightUNet network, standardizing on channel configurations of $[16, 32, 64, 96, 160]$ across all experiments, unless specified otherwise. Model optimization is achieved via the AdamW (Loshchilov & Hutter, 2017) optimizer. **The dataset specific implementation details are in Appendix A.3.**

## 4.2 RESULTS ON BINARY SEGMENTATION

Table 1 and Fig. 1 compare our UltraLightUNet with SOTA CNNs and Transformers on six datasets for four binary medical segmentation tasks. Our UltraLightUNet achieves the top average DICE score of 89.75% with an ultra-lightweight footprint of only 0.316M #Params and 0.314G #FLOPs. Our UltraLightUNet-T with 0.027M #Params and 0.062G #FLOPs, outperforms the existing tiny model EGE-UNet (Ruan et al., 2023) by on an average 5.93% DICE score over six datasets. The multi-kernel inverted residuals, alongside convolutional multi-focal attention mechanisms, play a crucial role in these strong results. The UltraLightUNet's performance on different datasets highlights its superior ability to balance accuracy with computational efficiency, setting a new benchmark for point-of-care services. The quantitative results of four different tasks are described next.

**Breast cancer segmentation:** Our UltraLightUNet shows superior performance on the BUSI dataset (Al-Dhabyani et al., 2020) with a DICE score of 78.04% by segmenting complex breast cancer lesions with diverse appearances. UltraLightUNet achieves comparable results with far fewer #Params and #FLOPs compared to heavyweight networks like TransUNet (78.01%) and SwinUNet (77.38%). Against lightweight networks such as UNeXt (74.71%), UltraLightUNet shows a 3.3% improvement with 4.7× lower #Params. Additionally, compared to ultra-lightweight networks like EGE-UNet (71.34%), UltraLightUNet exhibits 6.7% higher DICE scores.

**Skin lesion segmentation:** UltraLightUNet outperforms most SOTA methods on the ISIC18 dataset (Codella et al., 2019) with a DICE score of 88.74% by effectively handling the diverse lesion shapes and sizes in ISIC18. Among heavyweight networks, UltraLightUNet achieves comparable performance to TransUNet (89.04%) and DeepLabv3+ (88.64%) with significantly fewer #Params and FLOPs. Compared to lightweight networks like UNeXt (87.78%) and Roling-UNet-S (87.35%), UltraLightUNet shows a 1.0-1.4% improvement with 4.7× and 5.7× fewer #Params. Even against ultra-lightweight methods such as EGE-UNet (86.95%) and UltraLight_VM_UNet (87.85%), UltraLightUNet-T (88.19%) demonstrates up to 1.2% better DICE score.

**Polyp segmentation:** In polyp segmentation on Clinic (Bernal et al., 2015) and Colon (Vázquez et al., 2017) datasets, our UltraLightUNet excels with leading scores of 93.48% and 90.01%, respectively, by effectively capturing variations in polyp shapes, sizes, and textures. UltraLightUNet achieves comparable performance with fewer #Params (0.316M) compared to heavyweight networks like TransUNet (105.32M) and SwinUNet (27.17M). Against lightweight networks like UNeXt and CMUNeXt, UltraLightUNet delivers a higher DICE score. Even among ultra-lightweight networks, UltraLightUNet-T (0.027M #Params) outperforms EGE-UNet and UltraLight_VM_UNet.

**Microscopic cell nuclei/structure segmentation:** For cell structure segmentation on the DSB18 and EM datasets, UltraLightUNet achieves DICE scores of 92.71% and 95.52%, respectively, by capturing complex cellular structures effectively even with its ultra-lightweight design. In contrast, networks like TransUNet and UNeXt, despite their heavyweight design and higher #Params and FLOPs, do not surpass the DICE score of UltraLightUNet. For instance, TransUNet achieves lower scores on DSB18 (92.04%) and EM (95.27%), while UNeXt falls 6.70% behind the UltraLightUNet.

## 4.3 RESULTS ON SYNAPSE MULTI-ORGAN SEGMENTATION

Table 2 shows that our UltraLightUNet networks achieve superior or comparable DICE scores compared to various SOTA lightweight and traditional methods on the Synapse Multi-Organ Segmentation benchmark. Traditional architectures like UNet and Att_UNet exhibit high parameter counts (34.53M and 34.88M), yet only achieve modest DICE scores of 70.11% and 71.70%. Advanced models such as TransUNet (77.61% DICE) and SwinUNet (79.13% DICE) show improved performance, but at a significant computational cost, with 105.28M and 27.17M #Params (7.2× of our UltraLightUNet-L (78.68% DICE)), respectively, thus making them less suitable for real-time tasks.

Among lightweight models, our UltraLightUNet-L outperforms the SOTA models by achieving the top DICE score of 78.68% with 3.76M #Params and 2.51G #FLOPs, thus surpassing Rolling_UNet (73.15%) and CMUNeXt (72.69%) with far fewer computational resources. Even UltraLightUNet-M achieves a competitive 76.01% DICE score, surpassing UNeXt (72.60%) and EGE-UNet (59.32%) with fewer #Params and #FLOPs. Our ultra-lightweight networks, UltraLightUNet-S and UltraLightUNet-T, also show a solid balance between performance and efficiency.

We note that the improved performances of our UltraLightUNet stems from the use of MKIR and CMFA blocks, which focus on extracting multi-scale features while reducing redundant computa-

Table 2: Experimental results of Synapse Multi-organ segmentation. #FLOPs are reported for 224×224 images. The average DICE scores of three runs are reported here. Our models have orders of magnitude fewer #Params and #FLOPs.

| Network | #Params (M) | FLOPs (G) | DICE (%) |
|---|---|---|---|
| UNet (Ronneberger et al., 2015) | 34.53 | 50.19 | 70.11 |
| Att_UNet (Oktay et al., 2018) | 34.88 | 51.04 | 71.70 |
| UNet++ (Zhou et al., 2018) | 9.164 | 26.74 | 74.87 |
| DeepLabV3+ (Chen et al., 2017) | 39.76 | 11.45 | 78.40 |
| TransUNet (Chen et al., 2021) | 105.28 | 24.73 | 77.61 |
| SwinUNet (Cao et al., 2021) | 27.17 | 6.2 | **79.13** |
| **UltraLightUNet-L (Ours)** | **3.76** | **2.51** | 78.68 |
| MedT (Valanarasu et al., 2021) | 1.564 | 1.957 | 62.29 |
| Rolling_UNet_S (Liu et al., 2024) | 1.783 | 1.613 | 73.15 |
| CMUNeXt (Tang et al., 2023) | 0.418 | 0.838 | 72.69 |
| UNeXt (Valanarasu & Patel, 2022) | 1.474 | 0.449 | 72.60 |
| **UltraLightUNet-M (Ours)** | 1.15 | 0.760 | **76.01** |
| **UltraLightUNet (Ours)** | **0.316** | **0.257** | 73.31 |
| EGE-UNet (Ruan et al., 2023) | 0.053 | 0.056 | 59.32 |
| UltraLight_VM_UNet (Wu et al., 2024) | 0.050 | **0.047** | 61.56 |
| **UltraLightUNet-S (Ours)** | 0.093 | 0.104 | **70.83** |
| **UltraLightUNet-T (Ours)** | **0.027** | 0.053 | 65.69 |

Table 3: Experimental results of 3D segmentation on MSD Prostate and FETA datasets. Our models have orders of magnitude fewer #Params and #FLOPs. We report the average DICE scores (%) of three runs. **Bold** and underlined values denote the best and second-best values, respectively.

| Architecture | Params (M) ↓ | FLOPs (G) ↓ | MSD Prostate ↑ | FETA ↑ |
|---|---|---|---|---|
| 3D U-Net (Çiçek et al., 2016) | 4.81 | 135.9 | 62.53 | 85.93 |
| nn-UNet (Isensee et al., 2021) | 31.2 | 743.3 | 67.85 | 87.24 |
| TransBTS (Wenxuan et al., 2021) | 31.6 | 110.4 | 68.02 | 87.52 |
| UNETR (Hatamizadeh et al., 2022) | 92.78 | 82.6 | 65.22 | 86.72 |
| nnFormer (Zhou et al., 2021) | 159.3 | 204.2 | 66.63 | 87.03 |
| SwinUNETR (Hatamizadeh et al., 2021) | 62.19 | 328.6 | 65.12 | 87.75 |
| 3D UX-Net (Lee et al., 2022) | 53.01 | 632.0 | 68.92 | **88.67** |
| SlimUNETR (Pang et al., 2023) | 1.78 | 11.99 | 59.01 | 86.98 |
| **UltraLightUNet3D-S (Ours)** | **0.163** | **2.03** | 69.20 | 87.15 |
| **UltraLightUNet3D (Ours)** | 0.453 | 3.42 | 70.52 | 87.92 |
| **UltraLightUNet3D-M (Ours)** | 1.42 | 7.1 | **71.51** | 88.40 |

tions. This allows UltraLightUNet to capture complex structure of organs more effectively than other lightweight methods, thus achieving SOTA results with significantly lower computational overhead.

### 4.4 RESULTS ON 3D SEGMENTATION

Table 3 presents the performance of our UltraLightUNet3D networks against several SOTA 3D medical image segmentation methods on the MSD Prostate (Antonelli et al., 2022) and FETA (Payette et al., 2021) datasets. Despite using significantly fewer #Params and #FLOPs, our models consistently achieve superior or comparable DICE scores. Notably, UltraLightUNet3D-M achieves the highest DICE score of 71.51% on MSD Prostate, outperforming large-scale models like nnFormer (66.63%) and SwinUNETR (65.12%), with only 1.42M parameters and 7.1G #FLOPs — substantially lower than nnFormer (159.3M, 204.2G) and SwinUNETR (62.19M, 328.6G). Moreover, compared to 3D UX-Net, UltraLightUNet3D-M not only improves the DICE score by 2.59% on MSD Prostate, but also reduces the #Params and #FLOPs by 37.3× and 89×, respectively. This performance gain can be attributed to our multi-kernel design and attention-based refinement strategy, which effectively capture multi-scale contextual features and enhance critical regions.

## 5 ABLATION STUDIES

We describe two critical ablation studies here and provide more in Appendix A.5-A.8.

### 5.1 IMPACT OF DIFFERENT COMPONENTS

Table 4 presents the performance of various configurations within the UltraLightUNet network across six medical image segmentation datasets, highlighting the impact of integrating different

Table 4: Effect of different components of UltraLightUNet with #channels = $[16, 32, 64, 96, 160]$ and $[1, 3, 5]$ kernels. UNeXt has #channels = $[16, 32, 128, 160, 256]$. We design Mobile UNet following the structure of UNeXt network. However, we use the #channels = $[16, 32, 64, 96, 160]$ and kernel size of $[3]$ with the original inverted residual block (IRB) in the Mobile UNet. We report the DICE scores (%) averaged over five runs. Best results are shown in bold.

| Network | #Params | #FLOPs | BUSI | Clinic | Colon | ISIC18 | DSB18 | EM |
|---|---|---|---|---|---|---|---|---|
| UNeXt | 1.47M | 0.57G | 74.71 | 90.20 | 83.84 | 87.78 | 86.01 | 93.81 |
| Mobile UNet | 0.271M | 0.230G | 72.41 | 90.90 | 84.15 | 87.20 | 90.52 | 94.87 |
| MKIR | 0.306M | 0.300G | 74.74 | 92.63 | 86.46 | 88.22 | 92.40 | 95.31 |
| MKIR + GAG | 0.310M | 0.311G | 74.98 | 91.97 | 86.56 | 88.34 | 92.67 | 95.48 |
| MKIR + MKIRA | 0.311M | 0.303G | 76.61 | 92.64 | 89.40 | 88.56 | 92.64 | 95.37 |
| **MKIR + GAG + MKIRA (Ours)** | 0.316M | 0.314G | **78.04** | **93.48** | **90.01** | **88.64** | **92.71** | **95.52** |

Table 5: Effect of multiple kernels in the depth-wise convolution of MKDC on BUSI dataset. The results for kernels beyond $7 \times 7$ are not reported as the performance does not scale proportionally with the computational cost of larger kernels. We use the UltraLightUNet network with #channels= $[16, 32, 64, 96, 160]$ for these experiments and report the FLOPs for $256 \times 256$ inputs. We report the DICE scores (%) averaging over five runs. Best results are highlighted in bold.

| Convolution kernels | #Params(M) | FLOPs(G) | DICE | Convolution kernels | #Params(M) | FLOPs(G) | DICE |
|---|---|---|---|---|---|---|---|
| $1 \times 1$ | 0.272 | 0.220 | 70.83 | $5 \times 5$ | 0.299 | 0.276 | 76.81 |
| $1 \times 1, 1 \times 1$ | 0.275 | 0.229 | 71.11 | $1 \times 1, 5 \times 5$ | 0.303 | 0.286 | 77.05 |
| $3 \times 3$ | 0.281 | 0.239 | 76.42 | $3 \times 3, 3 \times 3, 3 \times 3$ | 0.306 | 0.295 | 76.86 |
| $1 \times 1, 3 \times 3$ | 0.284 | 0.248 | 76.81 | $3 \times 3, 5 \times 5$ | 0.312 | 0.304 | 77.62 |
| $1 \times 1, 1 \times 1, 3 \times 3$ | 0.288 | 0.257 | 77.08 | $\mathbf{1 \times 1, 3 \times 3, 5 \times 5}$ | 0.316 | 0.314 | **78.04** |
| $3 \times 3, 3 \times 3$ | 0.294 | 0.267 | 76.83 | $5 \times 5, 5 \times 5$ | 0.331 | 0.342 | 77.88 |
| $1 \times 1, 3 \times 3, 3 \times 3$ | 0.297 | 0.276 | 77.26 | $5 \times 5, 5 \times 5, 5 \times 5$ | 0.362 | 0.408 | 77.80 |

components like MKIR, GAG, and MKIRA. The comparison spans models from UNeXt to the advanced MKIR + GAG + MKIRA variant, revealing a progressive improvement in the DICE scores with the addition of each component. Notably, the multi-kernel trick, implemented through MKIR (in the encoder) and MKIRA (in the decoder) blocks, is the most critical component for improving the segmentation accuracy, increasing the DICE score from 72.41% to 76.61% on the BUSI dataset. This indicates the significant contribution of the multi-kernel approach to feature extraction and refinement. However, when we integrate all proposed modules (MKIR + GAG + MKIRA), our model achieves the highest overall DICE score of 78.04% on the same dataset, with minimal computational resources (0.316M #Params and 0.314G #FLOPs). This exhibits the efficacy of combining multi-kernel convolution with attention mechanisms within UltraLightUNet.

## 5.2 EFFECT OF MULTIPLE KERNELS

Table 5 evaluates the influence of different convolutional kernel combinations on the performance of MKDC within the UltraLightUNet network, specifically for the BUSI dataset. By experimenting with a variety of kernel sizes ranging from $1$ to $3, 5, 7$, it becomes evident that a mix of $1, 3, 5$ kernels stands out by achieving the best DICE score of 78.04% with a moderate increase in computational resources (0.316M #Params and 0.314G #FLOPs). This finding highlights the effectiveness of a multi-scale kernel approach in capturing diverse feature representations, thus significantly improving segmentation accuracy without a substantial rise in computational demands. Drawing from these empirical findings, we opt for the kernel combination of $[1, 3, 5]$ across all our experiments.

## 6 CONCLUSION

In this paper, we have presented UltraLightUNet, a new network for medical image segmentation that achieves high accuracy with an ultra-lightweight design. UltraLightUNet outperforms state-of-the-art models across multiple benchmarks while maintaining a significantly lower computational footprint. For example, UltraLightUNet surpasses the performance of TransUNet in DICE scores with $333\times$ fewer #Params and $123\times$ fewer #FLOPs. Similarly, UltraLightUNet improves segmentation accuracy by up to 6.7% compared to UNeXt, while using $4.7\times$ fewer #Params. Our design efficiently captures complex spatial relationships and refines salient features, thus making it ideal for resource-constrained environments such as point-of-care services, where real-time, high-fidelity diagnostics are essential.

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

# A APPENDIX / SUPPLEMENTAL MATERIAL

## A.1 DATASETS

We evaluate the UltraLightUNet's efficacy across 11 datasets covering eight segmentation tasks. Our six datasets from four binary segmentation tasks includes breast cancer (BUSI (Al-Dhabyani et al., 2020), 647 images: 437 benign and 210 malignant), polyp (ClinicDB (Bernal et al., 2015) with 612 images, and ColonDB (Vázquez et al., 2017) with 379 images), skin lesion (ISIC18 (Codella et al., 2019), 2,594 images), and cell nuclei/structure segmentation (DSB18 (Caicedo et al., 2019) with 670 images, and EM (Cardona et al., 2010) with 30 images). These datasets, collected from various imaging centers, offer a broad diversity in image characteristics, ensuring a comprehensive evaluation. An 80:10:10 train-val-test split was applied across all the binary segmentation datasets and the DICE score of testset is reported.

Our two 2D multi-class segmentation datasets are Synapse Multi-organs [1] and ACDC cardiac organs [2]. The Synapse multi-organ dataset is used for abdominal organ segmentation and includes 30 abdominal CT scans with 3,779 axial slices of $512 \times 512$ pixels. Following the TransUNet (Chen et al., 2021), 18 scans (2,212 slices) are used for training and 12 for validation. We segment eight organs: aorta, gallbladder, left kidney, right kidney, liver, pancreas, spleen, and stomach. For cardiac organ segmentation, the ACDC dataset contains 100 cardiac MRI scans segmented into three sub-organs: right ventricle, myocardium, and left ventricle. We follow the TransUNet protocol using 70 cases (1,930 slices) for training, 10 for validation, and 20 for testing.

We perform experiments on five public multi-modality datasets for 3D volumetric segmentation: (1) MICCAI 2021 FeTA Challenge dataset (FeTA2021) (Payette et al., 2021), (2) Medical Segmentation Decathlon (MSD) (Antonelli et al., 2022) Task01_BrainTumour, (3) MSD Task05_Prostate, (4) MSD Task06_Lung, and (5) Synapse Multi-organ.

(1) For FeTA2021, we use 80 T2-weighted infant brain MRIs from the University Children's Hospital, acquired using 1.5T and 3T clinical whole-body scanners, for brain tissue segmentation with annotations of seven distinct tissues. We perform a five-fold cross-validation and report the average DICE score.

(2) The MSD Task01_BrainTumour dataset consists of multi-modal 848 MRI scans, including T1, T1-contrast enhanced (T1ce), T2, and FLAIR modalities. The dataset focuses on the segmentation of three tumor sub-regions: Tumor Core (TC), Whole Tumor (WT), and Non-enhancing Tumor (NET). A major challenge in this dataset lies in the heterogeneous appearance and spatial distribution of the tumor sub-regions, caused by differences in tumor morphology, size, and imaging characteristics across patients. We use 396 scans for training and 96 scans for validation.

(3) The MSD Task05_Prostate dataset comprises 32 annotated MRI scans across two modalities, targeting the prostate peripheral zone (PZ) and transition zone (TZ). One of the major challange of this dataset is the significant inter-subject variability. We use 26 MRI scans for training and remaining 6 scans for validation.

(4) The MSD Task06_Lung dataset comprises 63 annotated CT scans, aiming to segment lung cancer lesions. The dataset presents significant challenges due to the variable sizes, shapes, and locations of lesions, as well as the presence of similar-looking non-tumorous structures like blood vessels and nodules. The diversity in scan quality and inter-subject anatomical variability further adds to the complexity in this dataset. We use 51 scans for training and 12 scans for validation.

(5) As described earlier, Synapse Multi-organ dataset contains 30 CT scans with the annotation of 13 abdominal organs (Spleen, Right Kidney, Left Kidney, Gallbladder, Esophagus, Liver, Stomach, Aorta, IVC, Portal and Splenic Veins, Pancreas, Right adrenal gland, Left adrenal gland). Following the splits of TransUNet (18 for training, 12 for validation), we perform both 13 and 8 class segmentation. For all of these 3D datasets, we report the results on validation set.

---

[1]https://www.synapse.org/#!Synapse:syn3193805/wiki/217789
[2]https://www.creatis.insa-lyon.fr/Challenge/acdc/

Table 6: Original Inverted Residual Block (IRB) (Sandler et al., 2018) vs our Multi-Kernel Inverted Residual (MKIR) with #channels = $[16, 32, 64, 96, 160]$. We use the kernel size of $[3]$ and $[1, 3, 5]$ for IRB and MKIR, respectively. We report the DICE scores (%) averaging over five runs. Best results are shown in bold.

| Blocks | #Params | #FLOPs | BUSI | Clinic | Colon | ISIC18 | DSB18 | EM |
|---|---|---|---|---|---|---|---|---|
| IRB | 0.271M | 0.230G | 72.41 | 90.90 | 84.15 | 87.20 | 90.52 | 94.87 |
| MKIR (**Ours**) | 0.306M | 0.300G | **74.74** | **92.63** | **86.46** | **88.22** | **92.40** | **95.31** |

Table 7: Effect of MKIRA in the encoder and decoder of UltraLightUNet with #channels = $[16, 32, 64, 96, 160]$ and $[1, 3, 5]$ kernels. We report the DICE scores (%) averaging over five runs. Best results are shown in bold.

| Encoder | Decoder | #Params | #FLOPs | BUSI | Clinic | Colon | ISIC18 | DSB18 | EM |
|---|---|---|---|---|---|---|---|---|---|
| MKIRA | MKIRA | 0.321M | 0.346G | 77.28 | 92.81 | 89.63 | 88.61 | 92.65 | 95.43 |
| (**Ours**) MKIR | MKIRA | 0.316M | 0.314G | **78.04** | **93.48** | **90.01** | **88.64** | **92.71** | **95.52** |

## A.2 EVALUATION METRICS

We use the DICE score to evaluate performance on all the datasets. The DICE score $DSC(Y, P)$ is calculated using Equations 7:

$$DSC(Y, P) = \frac{2 \times |Y \cap P|}{|Y| + |P|} \times 100 \tag{7}$$

where $Y$ and $P$ are the ground truth and predicted segmentation map, respectively.

## A.3 DATASET SPECIFIC IMPLEMENTATION DETAILS

For binary segmentation, training spans over 200 epochs with batches of 16, learning rate of $1e - 4$, and weight decay, during which we save the model achieving the highest DICE score. Image dimensions are set to $256 \times 256$ pixels for BUSI (Al-Dhabyani et al., 2020), ISIC18 (Codella et al., 2018), EM (Cardona et al., 2010), and DSB18 (Caicedo et al., 2019) datasets, while for ClinicDB (Bernal et al., 2015) and ColonDB (Vázquez et al., 2017), the resolution is adjusted to $352 \times 352$ pixels. We utilize a multi-scale training approach, with scales of $\{0.75, 1.0, 1.25\}$, and enforce gradient clipping at 0.5. For binary segmentation, we do not apply any form of augmentation and use a hybrid loss function that combines (1:1) weighted BinaryCrossEntropy (BCE) and Intersection over Union (IoU) loss.

For multi-class segmentation in Synapse Multi-organs and ACDC datasets, we use an input size of $224 \times 224$, employ random rotation and flipping as data augmentation, and optimize the combined Cross-entropy (0.3) and DICE (0.7) with a learning rate of $1e - 4$. We train models for 300 and 400 epochs with a batch size of 6 and 12 for Synapse and ACDC datasets, respectively. In the case of 3D segmentation in MSD Prostate, FETA, and Synapse Multi-organs datasets, the DiceCELoss is optimized for 40000 iterations with a learning rate of $1e - 3$. We use an input size of $96 \times 96 \times 96$ and augmentations the same as 3D UX-Net (Lee et al., 2022).

We consider MSD Task01_BrainTumour as a multi-level segmentation problem and use the *Sigmoid* activation on the prediction. While we use *Softmax* activation on the prediction for multi-class segmentation on MSD Task05_Prostate and binary segmentation on MSD Task06_Lung datasets. We train the model for 60000 iterations on MSD Task01_BrainTumour and use 40000 iterations for MSD Task05_Prostate and MSD Task06_Lung datasets. For these three MSD datasets, we use an input size of $96 \times 96 \times 96$, learning rate of $1e - 3$, and augmentations the same as SwinUNETR (Tang et al., 2022).

Table 8: Original Attention Gate (AG) (Sandler et al., 2018) vs our Grouped Attention Gate (GAG) with #channels = $[16, 32, 64, 96, 160]$ in UltraLightUNet. We use the kernel size of 3 for GAG. We report the DICE scores (%) averaging over five runs. Best results are shown in bold.

| Blocks | #Params | #FLOPs | BUSI | Clinic | Colon | ISIC18 | DSB18 | EM |
|---|---|---|---|---|---|---|---|---|
| AG | 0.326M | 0.320G | 77.61 | 93.02 | 89.78 | 88.38 | 92.48 | 95.31 |
| **GAG (Ours)** | **0.316M** | **0.314G** | **78.04** | **93.48** | **90.01** | **88.64** | **92.71** | **95.52** |

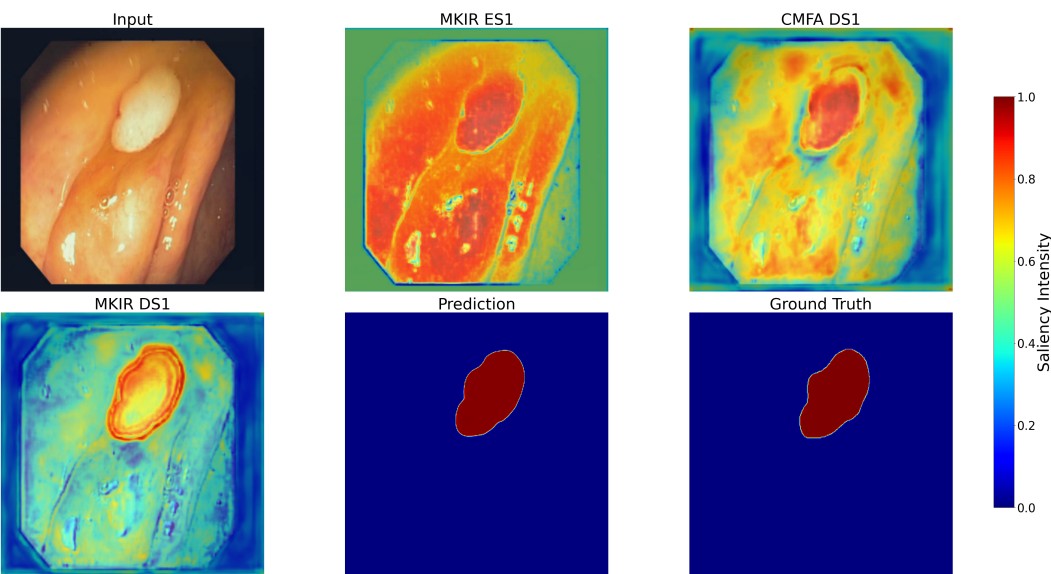

Figure 3: Activation heatmaps visualization of CMFA and MKIR.

## A.4 EFFECTIVENESS OF OUR MULTI-KERNEL INVERTED RESIDUAL (MKIR) OVER INVERTED RESIDUAL BLOCK (IRB) (SANDLER ET AL., 2018)

Table 6 reports the results of the original IRB of MobileUNetv2 (Sandler et al., 2018) and our proposed MKIR block. It can be concluded from the table that our MKIR significantly outperforms (up to 2.33%) IRB in all the datasets with only an additional 0.035M #Params and 0.07G #FLOPs. The use of lightweight convolutions with multiple kernels contributes to these performance improvements with nominal additional computational resources.

## A.5 EFFECTIVENESS OF OUR MULTI-KERNEL INVERTED RESIDUAL (MKIR) OVER MULTI-KERNEL INVERTED RESIDUAL ATTENTION (MKIRA) IN ENCODER

The experimental results in Table 7 demonstrate that employing MKIR in the encoder and MKIRA in the decoder yields superior performance across all datasets. Specifically, this configuration achieves the best average DICE scores of 78.04% (BUSI), 93.48% (Clinic), 90.01% (Colon), 88.64% (ISIC18), 92.71% (DSB18), and 95.52% (EM). The MKIR block in the encoder effectively extracts complex features by leveraging multiple kernels to capture a diverse range of spatial patterns and global contexts without the need for localized attention, which is more computationally intensive. Since the encoder primarily focuses on feature extraction, this design helps preserve critical details while maintaining lightweightness. In contrast, localized attention is crucial in the decoder to facilitate precise reconstruction. The MKIRA block in the decoder attends to key spatial regions, enabling effective feature refinement. This complementary setup leads to an optimal balance between performance and computational cost, as evidenced by the superior results achieved with only 0.316M parameters and 0.314G #FLOPs.

Table 9: Analysis of the number of channels on different datasets. We report #FLOPs for $256 \times 256$ inputs and the DICE scores (%) averaging over five runs, thus having 1-4% standard deviations.

| Network | C1 | C2 | C3 | C4 | C5 | #Params | #FLOPs | BUSI | Clinic | Colon | ISIC18 | DSB18 | EM |
|---------|----|----|----|----|----|---------|--------|------|--------|-------|--------|-------|----|
| UltraLightUNet-T | 4 | 8 | 16 | 24 | 32 | **0.027M** | **0.062G** | 75.64 | 91.26 | 85.03 | 88.19 | 92.38 | 94.69 |
| UltraLightUNet-S | 8 | 16 | 32 | 48 | 80 | 0.093M | 0.125G | 77.26 | 92.31 | 88.78 | 88.57 | 92.45 | 95.22 |
| UltraLightUNet | 16 | 32 | 64 | 96 | 160 | 0.316M | 0.314G | 78.04 | 93.48 | 90.01 | 88.74 | 92.71 | 95.52 |
| UltraLightUNet-M | 32 | 64 | 128 | 192 | 320 | 1.15M | 0.951G | 78.27 | 93.67 | 90.27 | 89.08 | 92.74 | 95.62 |
| UltraLightUNet-L | 64 | 128 | 256 | 384 | 512 | 3.76M | 3.19G | **79.02** | **93.85** | **91.82** | **89.25** | **92.80** | **95.67** |

Table 10: Analysis of the number of channels on different 3D datasets. The #FLOPs are reported for $96 \times 96 \times 96$ 3D input volumes. We report the average DICE scores (%) of three runs.

| Architecture | #Params(M) | #FLOPs(G) | MSD Prostate | FETA |
|--------------|-----------|-----------|--------------|------|
| UltraLightUNet3D-T | **0.061** | **1.45** | 61.21 | 84.24 |
| UltraLightUNet3D-S | 0.163 | 2.03 | 69.20 | 87.15 |
| UltraLightUNet3D | 0.453 | 3.42 | 70.52 | 87.92 |
| UltraLightUNet3D-M | 1.42 | 7.1 | **71.51** | **88.40** |
| UltraLightUNet3D-L | 4.28 | 18.0 | 71.04 | 88.11 |

## A.6 EFFECTIVENESS OF OUR GROUPED ATTENTION GATE (GAG) OVER ATTENTION GATE (AG) (OKTAY ET AL., 2018)

Table 8 reports the results of the original AG of Attention UNet (Oktay et al., 2018) and our proposed GAG block. It can be seen from the table that our GAG surpasses AG in all the datasets with 0.01M less #Params and 0.06G less #FLOPs. The use of group convolutions with a larger kernel (3) contributes to these performance improvements with less computational costs.

## A.7 ACTIVATION HEATMAPS VISUALIZATION

In Fig. 3, we plot the average activation heatmaps for all channels in high-resolution layers, focusing on Encoder Stage 1 (ES1) and Decoder Stage 1 (DS1). In ES1, the MKIR block attends to diverse regions, including the polyp region, thus capturing broad spatial features as expected in the initial stages of the encoder. In contrast, the CMFA layer in DS1 sharpens attention, thus focusing more locally on the polyp region. Subsequently, the MKDC within the MKIR block of DS1 further refines these attended features, thus concentrating exclusively on the polyp region (indicated by deep red areas). This progression highlights the effectiveness of our architecture in capturing and refining features, thus resulting in a segmentation map that strongly overlaps with the ground truth.

## A.8 ANALYSIS OF THE NUMBER OF CHANNELS

We conduct an ablation study with the different number of channel dimensions in different stages of the network to show the scalability of our network. Table 9 reports the results of this set of experiments. The progression from UltraLightUNet-T to UltraLightUNet-L in Table 9 demonstrates a clear positive correlation between model complexity and performance. Starting with UltraLightUNet-T's minimal resource use (0.027M #Params, 0.062G #FLOPs) yielding a 75.64% DICE score on BUSI, the score increases to 78.04% with UltraLightUNet's moderate complexity (0.316M #Params, 0.314G #FLOPs), and peaks at 79.02% with UltraLightUNet-L's higher resource demand (3.76M #Params, 3.19G #FLOPs). This trend of increasing DICE score with model complexity is consistent across datasets.

Additionally, Table 10 shows the impact of varying channel sizes on the 3D segmentation on MSD Prostate and FETA datasets. As channels increase, performance improves, with UltraLightUNet3D-M achieving the best DICE scores (71.51% for MSD Prostate and 88.40% for FETA) at 1.42M parameters and 7.1G #FLOPs. Further increasing to UltraLightUNet3D-L offers minimal gains, thus highlighting diminishing returns in performance beyond a certain point for 3D volumetric segmentation. The smallest model, UltraLightUNet3D-T, performs the worst, thereby demonstrating that too few channels limit segmentation accuracy. Overall, UltraLightUNet3D-M shows the best balance between model size and performance.

Table 11: Results of cardiac organ segmentation on ACDC dataset. Our models have orders of magnitude fewer #Params and #FLOPs. DICE scores (%) are reported for individual organs. Best results are shown in bold.

| Network | #Params (M) | #FLOPs (G) | Avg. | RV | Myo | LV |
|---|---|---|---|---|---|---|
| UNet (Ronneberger et al., 2015) | 35.53 | 50.19 | 87.55 | 87.10 | 80.63 | 94.92 |
| Attn_UNet (Oktay et al., 2018) | 34.88 | 51.04 | 86.75 | 87.58 | 79.20 | 93.47 |
| TransUNet (Chen et al., 2021) | 105.28 | 24.73 | 89.71 | 86.67 | 87.27 | 95.18 |
| SwinUNet (Cao et al., 2021) | 27.17 | 6.20 | 88.07 | 85.77 | 84.42 | 94.03 |
| **UltraLightUNet-L (Ours)** | **3.76** | **2.51** | **90.49** | **88.36** | **87.78** | **95.33** |
| MedT (Valanarasu et al., 2021) | 1.564 | 1.957 | 80.43 | 77.98 | 73.74 | 89.59 |
| Rolling_UNet_S (Liu et al., 2024) | 1.783 | 1.613 | 87.59 | 85.02 | 83.59 | 94.17 |
| CMUNeXt (Tang et al., 2023) | 0.418 | 0.838 | 85.19 | 81.30 | 82.54 | 91.74 |
| UNeXt (Valanarasu & Patel, 2022) | 1.474 | 0.449 | 84.68 | 81.06 | 81.22 | 91.76 |
| **UltraLightUNet-M (Ours)** | 1.15 | 0.760 | **89.93** | **87.76** | **86.9** | **95.14** |
| **UltraLightUNet (Ours)** | 0.316 | 0.257 | 88.80 | 86.03 | 85.9 | 94.46 |
| EGE-UNet (Ruan et al., 2023) | 0.053 | 0.056 | 80.68 | 76.6 | 75.21 | 90.23 |
| UltraLight_VM_UNet (Wu et al., 2024) | 0.050 | 0.047 | 81.82 | 78.63 | 76.48 | 90.36 |
| **UltraLightUNet-S (Ours)** | 0.093 | 0.104 | **87.32** | **84.41** | **83.50** | **94.03** |
| **UltraLightUNet-T (Ours)** | **0.027** | **0.053** | 82.42 | 80.02 | 76.26 | 91.00 |

Table 12: Experimental Results of the 3D Version of UltraLightUNet on Synapse Multi-Organ Segmentation. Our models have orders of magnitude fewer #Params and #FLOPs. We report the average DICE scores (%) of three runs. Best results are shown in bold.

| Architecture | #Params (M) ↓ | #FLOPs (G) ↓ | Synapse (8 organs) | Synapse (13 organs) |
|---|---|---|---|---|
| 3D U-Net (Çiçek et al., 2016) | 4.81 | 135.9 | 80.12 | 73.96 |
| nn-UNet (Isensee et al., 2021) | 31.2 | 743.3 | 82.96 | 78.58 |
| TransBTS (Wenxuan et al., 2021) | 31.6 | 110.4 | 82.74 | 77.42 |
| UNETR (Hatamizadeh et al., 2022) | 92.78 | 82.6 | 81.28 | 75.43 |
| nnFormer (Zhou et al., 2021) | 159.3 | 204.2 | 82.94 | 77.86 |
| SwinUNETR (Hatamizadeh et al., 2021) | 62.19 | 328.61 | 83.98 | **80.49** |
| 3D UX-Net (Lee et al., 2022) | 53.01 | 631.97 | **84.12** | 78.78 |
| SlimUNETR (Pang et al., 2023) | 1.78 | 11.99 | 80.42 | 72.56 |
| **UltraLightUNet3D-S (Ours)** | **0.163** | **2.03** | 81.89 | 74.81 |
| **UltraLightUNet3D (Ours)** | 0.453 | 3.42 | 81.87 | 76.33 |
| **UltraLightUNet3D-M (Ours)** | 1.42 | 7.1 | 82.58 | 77.46 |
| **UltraLightUNet3D-L (Ours)** | 4.28 | 18.00 | 82.90 | 77.24 |

## A.9    RESULTS ON CARDIAC ORGAN SEGMENTATION ON ACDC DATASET

Table 11 presents the performance comparison of our UltraLightUNet networks against several SOTA models on the ACDC cardiac organ segmentation dataset. Our UltraLightUNet-L model achieves the highest average DICE score of 90.49%, significantly outperforming traditional models like UNet (87.55%) and Attn_UNet (86.75%) despite having far fewer #Params (3.76M vs. 35.53M and 34.88M) and #FLOPs (2.51G vs. 50.19G and 51.04G). Even compared to more advanced models like TransUNet and SwinUNet, UltraLightUNet-L surpasses them in performance (90.49% vs. 89.71% and 88.07%) with a fraction of the computational costs. Among lightweight models, our UltraLightUNet-M (1.15M #Params) and UltraLightUNet (0.316M #Params) achieve superior results compared to Rolling_UNet_S (87.59%) and UNeXt (84.68%). The improved performance of our models can be attributed to the MKIR and CMFA blocks, which enable effective feature encoding, attention, and refinement, thus resulting in better discrimination of critical patterns of cardiac organs. The exceptionally low #Params and #FLOPs of UltraLightUNet-T and UltraLightUNet-S further highlight the efficiency of our method while maintaining competitive performance.

Table 13: Experimental results (DICE %) of 3D Brain tumor and Lung cancer segmentation on MSD Task01_BrainTumour (4-channel inputs) and MSD Task06_Lung datasets. #FLOPs are reported for 4-channel inputs with 96x96x96 volumes. The **bold** and underlined values highlight the best and second best values in each column. **Note:** Tumor Core (TC), Whole Tumor (WT), Non-enhancing Tumor (NET).

| Architecture | #Params (M) ↓ | #FLOPs (G) ↓ | Task01_BrainTumour | | | | Task06_Lung |
|---|---|---|---|---|---|---|---|
| | | | TC | WT | NET | Avg. | Cancer |
| UNETR (Hatamizadeh et al., 2022) | 92.78 | 82.6 | 79.77 | 89.83 | 57.47 | 75.69 | 65.38 |
| TransBTS (Wenxuan et al., 2021) | 31.60 | 110.4 | 80.09 | 88.38 | 55.89 | 74.79 | 63.57 |
| nnFormer (Zhou et al., 2021) | 159.03 | 204.2 | 83.19 | 90.14 | 60.15 | 77.82 | 69.79 |
| 3D UX-Net (Lee et al., 2022) | 53.01 | 632.0 | 82.90 | 91.13 | 61.72 | 78.58 | 71.46 |
| SwinUNETR (Hatamizadeh et al., 2021) | 62.19 | 328.6 | 83.19 | 91.36 | 62.62 | 79.06 | 65.12 |
| SlimUNETR (Pang et al., 2023) | 1.78 | 5.25 | 79.86 | 87.95 | 50.18 | 72.66 | 67.66 |
| **UltraLightUNet3D (Ours)** | **0.453** | **3.68** | 82.98 | 90.30 | 60.23 | 77.92 | 70.32 |
| **UltraLightUNet3D-M (Ours)** | 1.42 | 7.33 | **83.41** | **91.51** | 61.92 | 78.95 | **71.53** |

Figure 4: Qualitative results of our UltraLightUNet and SOTA methods. The incorrect segmented regions by different methods are highlighted using the red rectangular box.

A.10 3D SEGMENTATION RESULTS ON SYNAPSE DATASET

Table 12 presents the results of our UltraLightUNet3D models on the Synapse Multi-Organ Segmentation benchmark, compared to several state-of-the-art (SOTA) methods. Our models demonstrate competitive performance across both 8-organ and 13-organ segmentation tasks, while requiring significantly fewer #Params and #FLOPs. For example, UltraLightUNet3D-M achieves a DICE score of 82.58% for the 8-organ segmentation with only 1.42M #Params and 7.1G #FLOPs, whereas SwinUNETR achieves a slightly higher score of 83.98% but with 62.19M #Params and 328.61G #FLOPs. Similarly, nn-UNet performs comparably (82.96%), but it requires 31.2M #Params and 743.3G #FLOPs, thereby making it less suitable for resource-constrained applications.

Even our lightweight versions, UltraLightUNet3D-S and UltraLightUNet3D, perform strongly, with DICE scores of 81.89% and 81.87%, respectively, on the 8-organ task, significantly outperforming 3D U-Net (80.12%) with a much smaller model size. Although UltraLightUNet3D-T, our smallest model, achieves lower scores (78.78%), it still outperforms 3D U-Net while using only 0.061M parameters. The comparatively lower performance of our models in the 13-organ task can be attributed to the added complexity of handling a greater number of organs, yet UltraLightUNet3D-M and UltraLightUNet3D-L still deliver comparable results with much lower computational costs. These results showcase the capability of our UltraLightUNet3D models to achieve high segmentation accuracy with minimal computational resources, thus making them well-suited for point-of-care services and real-time applications.

A.11 3D BRAIN TUMOR AND LUNG CANCER SEGMENTATION RESULTS

Table 13 shows that our UltraLightUNet3D-M achieves the best DICE scores in Tumor core (83.41%), Whole tumor (91.51%), and Lung cancer (71.53%) segmentation while maintaining remarkably lower computational costs compared to heavyweight methods like SwinUNETR (62.19M parameters, 328.6G FLOPs) and 3D UX-Net (53.01M parameters, 632.0G FLOPs). Furthermore,

Table 14: Computational complexity (#Params, #FLOPs, Training Time (sec.), Inference Time (sec.)) comparisons of different architectures including our UltraLightUNet. We train each model for 200 epochs using a batch size of 16 with a total 1000 sample images of resolution 256×256 to get the total training time (sec.) on a NVIDIA RTX A6000 GPU. While we run the inference on 500 samples on the same GPU with a batch size of 1 and report the average inference time (ms) per image. We report the average DICE scores of six binary segmentation datasets (i.e., BUSI, ClinicDB, ColonDB, ISIC2018, DSB2018, EM) here for reasonable comparison. **Note:** The **bold** and underlined values highlight the best and second best values in each column.

| Architecture | #Params (M) ↓ | #FLOPs (G) ↓ | Training Time (sec.) ↓ | Inference Time (sec.) ↓ | Avg DICE (%) ↑ |
|---|---|---|---|---|---|
| U-Net (Ronneberger et al., 2015) | 34.53 | 65.53 | 1732.11 | 0.0084 | 87.28 |
| AttUNet (Oktay et al., 2018) | 34.88 | 66.64 | 1988.31 | 0.0092 | 87.86 |
| UNet++ (Zhou et al., 2018) | 9.16 | 34.65 | 794.38 | 0.0073 | 88.16 |
| PraNet (Fan et al., 2020) | 32.55 | 6.93 | 685.37 | 0.0156 | 87.79 |
| DeepLabv3+ (Chen et al., 2017) | 39.76 | 14.92 | 695.82 | 0.0078 | 89.15 |
| UACANet (Kim et al., 2021) | 69.16 | 31.51 | 850.53 | 0.0231 | 87.81 |
| TransUNet (Chen et al., 2021) | 105.32 | 38.52 | 1523.68 | 0.0153 | 89.59 |
| SwinUNet (Cao et al., 2021) | 27.17 | 6.20 | 828.99 | 0.0124 | 88.84 |
| DeformableLKA (Azad et al., 2024) | 102.76 | 26.03 | 5450.26 | 0.0663 | **89.92** |
| MedT (Valanarasu et al., 2021) | 1.57 | 1.95 | 7138.91 | 0.1191 | 82.42 |
| Rolling-UNet-S (Liu et al., 2024) | 1.78 | 2.10 | 635.69 | 0.0175 | 87.36 |
| CMUNeXt (Tang et al., 2023) | 0.418 | 1.09 | 450.86 | **0.0057** | 88.25 |
| UNeXt (Valanarasu & Patel, 2022) | 1.47 | 0.57 | **216.26** | 0.0058 | 86.06 |
| EGE-UNet (Ruan et al., 2023) | 0.054 | 0.072 | 360.69 | 0.0099 | 83.82 |
| UltraLight_VM_UNet (Wu et al., 2024) | 0.050 | **0.060** | 318.04 | 0.0102 | 85.53 |
| **UltraLightUNet-T (Ours)** | **0.027** | 0.062 | 312.08 | 0.0071 | 87.87 |
| **UltraLightUNet-S (Ours)** | 0.093 | 0.125 | 348.78 | 0.0072 | 89.10 |
| **UltraLightUNet (Ours)** | 0.316 | 0.314 | 474.02 | 0.0072 | 89.75 |

UltraLightUNet3D-M achieves the second-best DICE scores in Non-enhancing tumor and average Brain tumor segmentation (78.95%), which demonstrates UltraLightUNet3D's balanced performance across tumor subregions.

Compared to the existing lightweight method SlimUNETR, UltraLightUNet3D-M achieves better segmentation results on all tasks while maintaining similar computational efficiency (1.42M parameters, 7.33 GFLOPs vs. SlimUNETR's 1.78M parameters, 5.25 GFLOPs).

Additionally, our base model, UltraLightUNet3D, demonstrates competitive performance compared to heavyweight models like 3D UX-Net and SwinUNETR, while significantly outperforming SlimUNETR, achieving the lowest computational cost (0.453M parameters, 3.68 GFLOPs). These results validate UltraLightUNet's ability to generalize to complex segmentation tasks with an excellent balance between accuracy and efficiency.

## A.12 QUALITATIVE RESULTS

In Figure 4, we report the segmentation maps of breast tumors, skin lesions, polyps, and cell segmentation for representative test images. In breast tumor segmentation, UNet, UNet++, and UNeXt show greater false segmentation, while TransUNet and our UltraLightUNet produce near-perfect segmentation maps. Similarly, in skin lesion segmentation, UNet, ResUNet, UNet++, AttnUNet, DeepLabV3+, PraNet, SwinuNet, and UNeXt miss part of the lesion (in red rectangular box). However, UACANet, TransUNet, ACC-UNet, and our UltraLightUNet can segment that challenging region well. Our UltraLightUNet can also segment the polyp correctly, while all other methods incorrectly segment another region as a polyp. In general, our UltraLightUNet produces the best overlapping segmentation map across all four tasks. The reason behind this well-rounded performance by our UltraLightUNet with a very low computational budget is the use of multi-kernel depth-wise convolutions along with gated and local attention mechanisms.

## A.13 TRAINING AND INFERENCE TIME COMPARISONS

Table 14 highlights the trade-offs between training/inference time and efficiency. Our UltraLightUNet variants achieve competitive or superior DICE scores with significantly fewer #Params and #FLOPs compared to all other architectures. For example, UltraLightUNet (474.02 sec training, 0.0072 sec inference) achieves 89.75% DICE with only 0.316M params and 0.314G FLOPs, thus

outperforming heavier models like DeepLabv3+ (89.15% DICE, 39.76M params, 14.92G FLOPs) and TransUNet (89.59% DICE, 105.32M params, 38.52G FLOPs).

While depth-wise convolutions slightly increase the training time due to reduced parallelism, they enable extreme computational efficiency (#Params and #FLOPs), thus making UltraLightUNet ideal for resource-constrained environments.

Finally, the inference time of our UltraLightUNet (0.0072 sec, 89.75% DICE) remains competitive when compared to the best lightweight baseline (based on inference time), CMUNeXt (0.0057 sec, 88.25% DICE), while offering higher DICE score.

### A.14   LIMITATIONS AND FUTURE DIRECTIONS

While UltraLightUNet excels in computational efficiency, its focus on extreme lightweight design occasionally results in slightly lower performance compared to SOTA methods on complex datasets (e.g., Synapse in Tables 2 and 12). This tradeoff reflects its primary goal of addressing resource constraints in real-time and point-of-care applications.

Future work will explore hybrid architectures that combine lightweight and high-capacity components to handle challenging tasks without sacrificing efficiency. Additionally, strategies like self-supervised pretraining and domain-specific optimizations can enhance its performance further. We also plan to extend UltraLightUNet to other dense prediction tasks, such as 2D/3D image reconstruction, translation, enhancement, and denoising. This opens pathways to broaden the UltraLightUNet's applicability across various computer vision tasks.

