# OpenReview forum: "UltraLightUNet: Rethinking U-shaped Network with Multi-kernel Lightweight Convolutions for Medical Image Segmentation"
_ICLR.cc/2025/Conference — ICLR 2025 Conference Withdrawn Submission_

### Official Review · Reviewer_SWMn · 2024-10-17

**Soundness:** 2
**Presentation:** 2
**Contribution:** 2
**Rating:** 3
**Confidence:** 4

**Summary:**

This manuscript proposes a novel U-shaped network incorporating various modules for medical image segmentation, with a focus on reducing computational costs. Key contributions include the Multi-Kernel Inverted Residual Block, Multi-Kernel Inverted Residual Attention, and Grouped Attention Gate. As a result, the proposed model achieves remarkable computational efficiency and delivers superior segmentation quality compared to state-of-the-art models. The manuscript is well-written and well-organized; however, more detailed technical insights into each module would enhance clarity and depth.

**Strengths:**

This manuscript presents various experiments validating the model's performance. The multi-kernel structures employed effectively capture multi-scale contexts, enhancing segmentation accuracy without adding complexity. The reported results demonstrate that the proposed model’s segmentation performance surpasses that of significantly heavier architectures.

**Weaknesses:**

Although well-organized, the manuscript could benefit from a deeper focus on explaining how each module reduces computational costs while maintaining high performance. The method section lacks clear evidence or mathematical proof to support the model’s design, which may present a scientific limitation. Additionally, there are no ablation studies evaluating the individual contributions of each module in terms of both segmentation performance and parameter efficiency.

**Questions:**

1. Can you clarify the effects of the proposed modules in the model with clear evidence or mathematical insight? The current equations in the manuscript seem somewhat redundant, serving primarily as mathematical restatements. It would be helpful to provide more rigorous insights or proofs demonstrating how each module contributes to performance improvement and computational efficiency.

2. Which proposed module is the most critical? Please provide ablation studies to highlight the individual impact of each module on both segmentation performance and computational cost. This would help identify the key components driving the model's success.

3. General multi-scale approaches are not novel. For instance, the "Spatial Feature Conservation Networks (SFCNs) for Dilated Convolutions to Improve Breast Cancer Segmentation from DCE-MRI" (International Workshop on Applications of Medical AI, 2022) employs a multi-scale strategy. What distinguishes your model’s approach to multi-scale feature extraction? Please elaborate on the unique contributions and insights of your method in this context.

4. What are the differences among UltraLightUNet- T, S, and L in terms of model? Layer difference? Please explain in details in the manuscript.

---

> ### Author Response · Authors · 2024-11-26
> **Response to the comments of Reviewer SWMn: Theoretical basis**
>
> We thank the reviewer for recognizing the effectiveness of UltraLightUNet's multi-kernel structures in capturing multi-scale contexts and achieving superior segmentation accuracy with minimal complexity. We also appreciate the constructive feedback and will address each of the reviewer’s comments in the following responses.
>
> ### **Q1. Although well-organized, the manuscript could benefit from a deeper focus on explaining how each module reduces computational costs while maintaining high performance. The method section lacks clear evidence or mathematical proof to support the model’s design, which may present a scientific limitation. Can you clarify the effects of the proposed modules in the model with clear evidence or mathematical insight? The current equations in the manuscript seem somewhat redundant, serving primarily as mathematical restatements. It would be helpful to provide more rigorous insights or proofs demonstrating how each module contributes to performance improvement and computational efficiency.**
>
> We appreciate the reviewer’s comment and provide additional clarification regarding UltraLightUNet’s design, which is grounded in clear theoretical concepts and validated through empirical studies and visualizations. Specifically:
>
> **Theoretical Basis:** Most existing architectures in computer vision rely on foundational concepts rather than explicit mathematical proofs to justify their design choices. For example, Vision Transformers leverage self-attention mechanisms, and ConvNeXt uses large-kernel convolutions to enhance feature extraction. Similarly, our approach employs the **multi-kernel trick** to balance local and global feature extraction via multi-kernel depth-wise convolutions (MKDC) and incorporates **depth-wise convolutions** for lightweight computation. These foundational design choices, while not purely theoretical (or have no mathematical proof), ensure that UltraLightUNet achieves both high segmentation performance and extreme efficiency.
>
> **Empirical Validation:** Ablation studies in Tables 4 and 5 of our initial submission quantify the individual and combined contributions of the modules. For instance, MKIR and MKIRA together significantly boost DICE scores, such as from 72.41% to 76.61% on the BUSI dataset, while the integration of all modules (MKIR, MKIRA, and GAG) achieves the best performance at 78.04%. This demonstrates how the design enhances segmentation accuracy while maintaining computational efficiency. We have additional abaltion experiments reported in Tables 6, 7, and 8 (in the Appendix of initial submission) which show the impact of our individual module over existing counterparts.
>
> **Visual Evidence:** Activation heatmap visualizations (Fig. 4 and Section A.7 in our revised draft) further validate the practical impact of our modules. These visualizations show that MKIR, combined with CMFA, effectively attends to and refines critical regions in an image, such as lesion boundaries, improving segmentation quality.
>
> To address the reviewer’s concern, we will revise the manuscript to contextualize our approach within the broader landscape of existing architectures that rely on foundational concepts rather than mathematical proofs.

---

> > ### Author Response · Authors · 2024-11-26
> > **Response to the comments of Reviewer SWMn: Component ablations and novelty**
> >
> > ### **Q2. Additionally, there are no ablation studies evaluating the individual contributions of each module in terms of both segmentation performance and parameter efficiency. Which proposed module is the most critical? Please provide ablation studies to highlight the individual impact of each module on both segmentation performance and computational cost. This would help identify the key components driving the model's success.**
> >
> > In our initial submission, we do have ablation studies reported in Table 4 and Section 5.2 (in our initial submission) to evaluate the impact of each proposed component on segmentation performance and parameter efficiency.
> >
> > The results in Table 4 demonstrate that the multi-kernel trick, implemented through MKIR (in the encoder) and MKIRA (in the decoder), is the most critical component for improving the segmentation accuracy, increasing the DICE score from 72.41% to 76.61% on the BUSI dataset. This indicates the significant contribution of the multi-kernel approach to feature extraction and refinement. However, when we integrate all proposed modules—MKIR, MKIRA, and GAG—our model achieves the highest overall DICE score of 78.04% on the same dataset.
> >
> > This pattern is consistent across other datasets (i.e., ClinicDB, ColonDB, ISIC18, DSB18, and EM), showing that while MKIR and MKIRA individually drive most of the performance improvements, the combination of all modules optimally balances accuracy and computational cost. We will include a more detailed explanation of these findings in the revised manuscript to further emphasize the role of each module in driving the model's success.
> >
> >
> > ### **Q3. General multi-scale approaches are not novel. For instance, the "Spatial Feature Conservation Networks (SFCNs) for Dilated Convolutions to Improve Breast Cancer Segmentation from DCE-MRI" (International Workshop on Applications of Medical AI, 2022) employs a multi-scale strategy. What distinguishes your model’s approach to multi-scale feature extraction? Please elaborate on the unique contributions and insights of your method in this context.**
> >
> > We thank the reviewer for raising this point and giving us the opportunity to clarify. Our UltraLightUNet distinguishes itself by introducing a new and lightweight multi-kernel feature extraction approach that integrates seamlessly into both 2D and 3D architectures. Below, we explain how our method differs and contributes uniquely to the field.
> >
> > **Multi-Kernel Depth-Wise Convolutions (MKDC):** Unlike standard multi-scale approaches, such as SFCNs that rely on dilated convolutions with different dilation rates (e.g., d₁ ≠ d₂ for dilation rates d₁, d₂), UltraLightUNet leverages MKDC, which supports both same-size kernels (k₁ = k₂) for uniform context extraction and different-size kernels (k₁ ≠ k₂) to balance the local and global contexts adaptively. This flexibility ensures efficient multi-scale feature extraction tailored to diverse spatial complexities while maintaining lightweight efficiency.
> >
> > **Volumetric 3D Extensions:** A key limitation of methods like SFCNs is their restriction to 2D tasks. In contrast, UltraLightUNet extends its multi-kernel strategy to 3D segmentation tasks through 3D versions of its modules, such as the MKIR and MKIRA blocks. These modules employ multi-kernel convolutions and attention mechanisms (e.g., CMFA) to refine features in 3D space, making UltraLightUNet highly effective for complex volumetric medical imaging tasks, such as tumor and organ segmentation.
> >
> > **Decoder Design with Sequential Refinement:** UltraLightUNet introduces the MKIRA block, which performs multi-kernel refinement of attention by combining CMFA for local attention with MKIR for multi-kernel convolutional refinement. Unlike SFCNs, which lack 3D extensions, MKIRA handles multi-kernel attention and feature refinement in both 2D and 3D tasks with significantly reduced computational overhead.
> >
> > **Lightweight Efficiency:** While SFCNs and similar methods rely on computationally intensive standard convolutions or dilations, UltraLightUNet achieves superior computational efficiency by using depth-wise convolutions. For example, our 3D base model, UltraLightUNet3D, achieves new SOTA efficiency with just 0.453M parameters and 3.42 GFLOPs, compared to existing 3D methods that typically require hundreds of Giga FLOPs. This efficiency is critical for resource-constrained scenarios, such as point-of-care diagnostics.
> >
> > In summary, UltraLightUNet distinguishes itself from standard multi-scale methods like SFCNs by combining multi-kernel depth-wise convolutions, lightweight attention mechanisms, and volumetric extensions into a unified framework. These innovations ensure competitive segmentation performance across both 2D and 3D tasks while significantly reducing computational costs.

---

> > > ### Author Response · Authors · 2024-11-26
> > > **Response to the comments of Reviewer SWMn: Difference in UltraLightUNet- T, S, and L**
> > >
> > > ### **Q.4 What are the differences among UltraLightUNet- T, S, and L in terms of model? Layer difference? Please explain in details in the manuscript.**
> > >
> > > We thank the reviewer for their insightful comment regarding the differences among the various versions of UltraLightUNet. In our manuscript, the different versions—UltraLightUNet-T (Tiny), UltraLightUNet-S (Small), UltraLightUNet (Base), UltraLightUNet-M (Medium), and UltraLightUNet-L (Large)—are distinguished by the number of channels used in the five stages of the U-shaped architecture. Specifically:
> > >
> > > **UltraLightUNet-T (Tiny):** Channels = (4, 8, 16, 24, 32)
> > >
> > > **UltraLightUNet-S (Small):** Channels = (8, 16, 32, 48, 80)
> > >
> > > **UltraLightUNet (Base):** Channels = (16, 32, 64, 96, 160)
> > >
> > > **UltraLightUNet-M (Medium):** Channels = (32, 64, 128, 192, 320)
> > >
> > > **UltraLightUNet-L (Large):** Channels = (64, 128, 256, 384, 512)
> > >
> > > These variations in the number of channels directly scale the model’s capacity, enabling the architecture to adapt to different resource constraints and performance requirements. The underlying layer-wise structure and module types remain consistent across all versions, ensuring architectural uniformity while allowing flexibility in computational complexity and performance.
> > >
> > > To demonstrate the scalability of our design, we reported an ablation study in **Tables 9 and 10 (Section A.7)** of the initial submission, which highlights the trade-offs between computational cost and performance as the number of channels varies. We will further clarify these details in the revised manuscript to address this comment thoroughly.

---

> ### Comment · Reviewer_SWMn · 2024-11-26
>
> Thank you for your effort and response. While some of my concerns (e.g., ablation results) have been addressed, I believe more focus is needed on providing theoretical support for the proposed model. The Learning Society should prioritize demonstrating technical insights and theoretical rationale over merely emphasizing performance improvements and simple interpretations. In current state, my score remains unchanged.

---

> > ### Author Response · Authors · 2024-12-02
> > **Official Comment by Authors**
> >
> > ### **C1. Thank you for your effort and response. While some of my concerns (e.g., ablation results) have been addressed, I believe more focus is needed on providing theoretical support for the proposed model. The Learning Society should prioritize demonstrating technical insights and theoretical rationale over merely emphasizing performance improvements and simple interpretations. In current state, my score remains unchanged.**
> >
> > **Response:**
> >
> > Thank you for your thoughtful feedback. While we acknowledge the importance of theoretical developments in advancing machine learning, it is also evident that empirical innovations, even without proposing new learning theories, have historically made impactful contributions to the field and have been well-recognized by the ICLR community. A few examples are given below.
> >
> > ### **Historical Context of Empirical Papers at ICLR**
> >
> > Several seminal architecture-focused papers accepted at ICLR have made substantial contributions without introducing any new learning theory, but by innovating through design principles and pushing the SOTA results:
> >
> > - **VGG** (Simonyan et al., ICLR 2015, 133516 citations): Introduced deeper networks by stacking convolutional layers in a straightforward architecture, becoming a cornerstone for CNN research without any new theoretical insights.
> >
> > - **3D UX-Net** (Lee et al., ICLR 2023, 141 citations): Extends the ConvNeXt block to volumetric data by designing a lightweight encoder but relies on a computationally expensive existing decoder. Its novelty lies in adapting an existing module for 3D processing rather than introducing any new theoretical concepts.
> >
> > - **MobileViT** (Mehta et al., ICLR 2022, 1454 citations): A hybrid CNN-Transformer model designed for resource-constrained settings, emphasizing practical deployment over theoretical learning innovations.
> >
> > - **CycleMLP** (Chen et al., ICLR 2022, 262 citations): Proposes local window-based MLPs to achieve computational efficiency, focusing on practical adaptability rather than novel theoretical foundations.
> >
> > These examples illustrate that impactful empirical contributions do not necessarily require new learning theories in order to be relevant contributions to ICLR, but rather can advance the field by improving the SOTA in performance, efficiency, and adaptability.
> >
> > ### **Our Contributions**
> >
> > UltraLightUNet aligns with this tradition of empirical innovation, contributing to the growing body of lightweight, resource-efficient models for real-time medical imaging. Our contributions include:
> >
> > **1. End-to-End Lightweight Design:** A novel encoder-decoder architecture built entirely from scratch for extreme efficiency in both 2D and 3D tasks.
> >
> > **2. Conceptually New Modules:**
> > - **Multi-Kernel Inverted Residual (MKIR)** and **MKIRA** blocks introduce a flexible multi-kernel approach which provides a new mathematical basis to address various segmentation challenges.
> >
> > - New **3D extensions** of all modules for volumetric medical imaging, a contribution not present in prior works like ConvNeXt.
> >
> > **3. Experimental Validation:** Comprehensive evaluations across 12 datasets demonstrate competitive accuracy with significantly lower computational costs, a critical need in medical imaging.
> >
> > ### **Perspective on Learning Theories vs. Empirical Innovations**
> >
> > We would like to start by stating that we definitely share the same desire as this Reviewer to see the entire field of ML for computer vision (and beyond) entirely build on solid (theoretical) principles. But given that the deep learning research is such a fluid and fast evolving field, this remains for now an aspirational objective, at best. A lot of empirical contributions remain for now (and perhaps the foreseeable future) very relevant if they can improve the SOTA and can stimulate more research in the area.
> >
> > From this perspective, while our work does not propose a new learning theory, it provides substantial architectural advancements and redefines the SOTA in image segmentation with limited resources, similar to many influential works cited (or not even mentioned) above. We believe that these contributions align well with the ICLR community’s history of recognizing impactful architectural innovations and their relevance to practical problems in the field.
> >
> > We hope this response provides clarity on the significance of our work and its alignment with ICLR’s standards. Thank you again for your constructive feedback.

---

> > > ### Comment · Reviewer_SWMn · 2024-12-03
> > >
> > > Thank you for providing such a relevant and comprehensive overview. I agree with some of the authors' responses; however, I still believe that top-tier AI conference papers should offer explicit or at least implicit insights. Considering that AI development is no longer in its early stages, it is crucial to focus on a deeper understanding of the field rather than merely pursuing performance improvements.

---

### Official Review · Reviewer_jk4q · 2024-10-26

**Soundness:** 4
**Presentation:** 4
**Contribution:** 4
**Rating:** 6
**Confidence:** 5

**Summary:**

The paper introduces UltraLightUNet, a novel ultra-lightweight, multi-kernel U-shaped network designed to improve medical image segmentation. Leveraging a new Multi-kernel Inverted Residual (MKIR) block for efficient multi-scale feature extraction and a Multi-kernel Inverted Residual Attention (MKIRA) block for refined feature enhancement, UltraLightUNet achieves high segmentation accuracy with minimal parameters and computational load. With only few parameters and FLOPs, the 2D version of UltraLightUNet outperforms existing lightweight and transformer-based segmentation models across multiple medical imaging benchmarks, while the 3D variant, UltraLightUNet3D, achieves superior results on complex 3D medical segmentation tasks with even greater efficiency. These performance gains make UltraLightUNet a viable option for real-time applications in resource-constrained environments, such as point-of-care diagnostics.

**Strengths:**

The paper's primary strength lies in its originality, presenting a lightweight model architecture, UltraLightUNet, that effectively combines multi-kernel convolutions and attention mechanisms, a novel approach for achieving high segmentation accuracy with low computational overhead. This innovation addresses a critical need for practical, high-performance segmentation models in resource-constrained environments, adding value to the field by bridging computational efficiency and segmentation quality. In terms of quality, the paper’s methodology appears well-supported by rigorous experimental validation across 11 datasets and comparison with SOTA models, demonstrating robust performance gains and highlighting the practical implications of the architecture's low parameter count and FLOPs. Clarity is another strength; the paper methodically explains the architecture components, from the MKIR and MKIRA blocks to grouped attention mechanisms, making it accessible to readers with a background in medical image segmentation. Finally, the significance of UltraLightUNet is considerable due to its broad applicability across a range of medical imaging tasks and its potential for real-time use in settings like point-of-care diagnostics. The model’s lightweight design, paired with high accuracy, addresses critical bottlenecks in deploying AI-driven diagnostics in clinical environments, establishing the paper as a meaningful contribution to the field. I am glad to see the author will open-source the code to promote research in this line.

**Weaknesses:**

First, while the model is evaluated across various medical imaging datasets, these datasets are relatively straightforward, covering simple segmentation tasks and organs rather than more complex applications like CT/MRI tumor and lesion segmentation.

Second, the masks are mostly binary segmentation tasks. The multi-class segmentation is not well explored. Adding more complicated and multi-class segmentation would better demonstrate the model's capability for broader, real-world medical imaging tasks.

Third, while the proposed blocks—MKIRA, MKIR, MKDC, GAG, and CMFA—are illustrated in the Method section, the paper lacks sufficient theoretical or conceptual motivation for why these specific block designs should enhance segmentation performance.

Lastly, the paper does not discuss the limitations of UltraLightUNet. A dedicated discussion on limitations would provide readers with a more balanced understanding of the model’s practical use and potential future directions for research.

**Questions:**

Typically, there is no free lunch. Are there any limitations or tradeoffs of the method? Providing those will be helpful for readers.

How it would work on more complicated problems (e.g., 3D tumors/lesions) and multi-class segmentation problems?

---

> ### Author Response · Authors · 2024-11-26
> **Response to the comments of Reviewer jk4q: Complex applications**
>
> We sincerely thank the reviewer for the constructive feedback, and highlighting both the strengths and areas for improvement in our work. We deeply appreciate the recognition of UltraLightUNet’s originality and its contributions to computational efficiency and high segmentation quality, achieved through its novel lightweight architecture and innovative modules. Below, we address each concern raised by the reviewer and outline how we plan to incorporate these suggestions in our revised manuscript.
>
> ### **Q1. First, while the model is evaluated across various medical imaging datasets, these datasets are relatively straightforward, covering simple segmentation tasks and organs rather than more complex applications like CT/MRI tumor and lesion segmentation. How it would work on more complicated problems (e.g., 3D tumors/lesions) and multi-class segmentation problems?**
>
> We appreciate the reviewer’s suggestion to evaluate UltraLightUNet on more complex tasks like CT/MRI tumor and lesion segmentation. To address this, we conducted additional experiments on the MSD Task01_BrainTumour (multi-level MRI segmentation) and Task06_Lung (binary CT segmentation) datasets, and the results are presented in **Table R5 below**. We note that our UltraLightUNet3D-M achieves the best DICE scores in Tumor core (83.41%), Whole tumor (91.51%), and Lung cancer (71.53%) segmentation while maintaining remarkably lower computational costs compared to heavyweight methods like SwinUNETR (62.19M parameters, 328.6G FLOPs) and 3D UX-Net (53.01M parameters, 632.0G FLOPs). Furthermore, UltraLightUNet3D-M achieves the second-best DICE scores in Non-Enhancing Tumor and average Brain Tumor segmentation (78.95%), demonstrating its balanced performance across tumor subregions.
>
> Compared to the existing lightweight method SlimUNETR, UltraLightUNet3D-M achieves better segmentation results on all tasks while maintaining similar computational efficiency (1.42M parameters, 7.33 GFLOPs vs. SlimUNETR's 1.78M parameters, 5.25 GFLOPs).
>
> Additionally, our base model, UltraLightUNet3D, demonstrates competitive performance compared to heavyweight models like 3D UX-Net and SwinUNETR, while significantly outperforming SlimUNETR, achieving the lowest computational cost (0.453M parameters, 3.68 GFLOPs). These results validate UltraLightUNet’s ability to generalize to complex segmentation tasks with an excellent balance of accuracy and efficiency.
>
> **Table R5:** Experimental results (DICE %) of 3D Brain tumor and Lung cancer segmentation on MSD Task01_BrainTumour (4-channel inputs) and MSD Task06_Lung datasets. #FLOPs are reported for 4-channel inputs with 96x96x96 volumes. **Note:** Tumor Core (TC), Whole Tumor (WT), Non-enhancing Tumor (NET).
>
> | Architecture            | #Params (M) | #FLOPs (G) | TC (Task01) | WT (Task01) | NET (Task01) | Avg. (Task01) | Task06_Lung Cancer |
> |-------------------------|-------------|------------|-------------------------|-------------------------|--------------------------|--------------------------|---------------------|
> | UNETR                  | 92.78       | 82.60      | 79.77                  | 89.83                  | 57.47                   | 75.69                   | 65.38               |
> | TransBTS               | 31.60       | 110.40     | 80.09                  | 88.38                  | 55.89                   | 74.79                   | 63.57               |
> | nnFormer               | 159.03      | 204.20     | $\underline{83.19}$                  | 90.14                  | 60.15                   | 77.82                   | 69.79               |
> | 3D UX-Net               | 53.01       | 632.00     | 82.90                  | 91.13                  | 61.72                   | 78.58                   | $\underline{71.46}$               |
> | SwinUNETR              | 62.19       | 328.60     | $\underline{83.19}$                  | $\underline{91.36}$                  | **62.62**               | **79.06**               | 65.12               |
> | SlimUNETR              | 1.78        | $\underline{5.25}$       | 79.86                  | 87.95                  | 50.18                   | 72.66                   | 67.66               |
> | UltraLightUNet3D (Ours)| **0.453**   | **3.68**   | 82.98                  | 90.56                  | 60.23                   | 77.92                   | 70.32               |
> | UltraLightUNet3D-M (Ours)| $\underline{1.42}$     | 7.33       | **83.41**              | **91.51**              | $\underline{61.92}$                   | $\underline{78.95}$                   | **71.53**           |

---

> > ### Author Response · Authors · 2024-11-26
> > **Response to the comments of Reviewer jk4q: Multi-class segmentations**
> >
> > ### **Q2. Second, the masks are mostly binary segmentation tasks. The multi-class segmentation is not well explored. Adding more complicated and multi-class segmentation would better demonstrate the model's capability for broader, real-world medical imaging tasks.**
> >
> > We appreciate the reviewer's suggestion to explore multi-class segmentation tasks to demonstrate the broader applicability of **UltraLightUNet**. We would like to highlight that our initial submission already includes results on several multi-class segmentation tasks across both 2D and 3D settings:
> >
> > - **2D Multi-Class Segmentation**: Results are provided for Synapse 8-organ segmentation (Table 2) and ACDC 3-organ segmentation (Table 11) in the initial submission.
> > - **3D Multi-Class Segmentation**: Results are reported for FETA 7-organ segmentation (Table 3), MSD Task07_Prostate segmentation (Table 3), Synapse 8-organ segmentation (Table 12), and Synapse 13-organ segmentation (Table 12) in the initial submission.
> >
> > Additionally, to address the reviewer's concern, we conducted further **multi-class segmentation experiments** during the rebuttal phase, specifically on the **MSD Task01_BrainTumour** segmentation dataset, which involves multi-level tumor segmentation. We present these new results in **Table R5 above**, which demonstrates that our **UltraLightUNet3D-M** outperforms existing popular heavyweight architectures (*SwinUNETR, nnFormer, and 3D UX-Net*) with remarkably lower #Params and #FLOPs. These new results (**Table R5 above**) will be included in the revised manuscript (**Table 13 in Appendix A.11**) to demonstrate **UltraLightUNet**’s capability to handle complex, multi-class segmentation tasks effectively.
> >
> > We believe these results, along with the new additions, comprehensively address the reviewer's concern and reinforce **UltraLightUNet**’s applicability to real-world medical imaging scenarios requiring multi-class segmentation.

---

> > > ### Author Response · Authors · 2024-11-26
> > > **Response to the comments of Reviewer jk4q: Theoretical motivation, limitations, and future directions**
> > >
> > > ### **Q3. Third, while the proposed blocks—MKIRA, MKIR, MKDC, GAG, and CMFA—are illustrated in the Method section, the paper lacks sufficient theoretical or conceptual motivation for why these specific block designs should enhance segmentation performance.**
> > >
> > > We appreciate the reviewer’s feedback regarding the need for more theoretical or conceptual motivation for our proposed blocks. **UltraLightUNet’s design is grounded in clear theoretical concepts**, which are explained in the Method section and validated in the Ablation Study section:
> > >
> > > - **Theoretical Basis**: Most existing architectures in computer vision rely on theoretical concepts to justify their design choices (e.g., Vision Transformers focus on self-attention). Similarly, our approach employs the **multi-kernel trick** to improve segmentation performance and **depth-wise convolutions** for lightweight computation. While our contribution is not theoretical in nature, these foundational concepts ensure that **UltraLightUNet** achieves both high performance and extreme efficiency.
> > >
> > > - **Method Section**: We provided detailed descriptions of how the MKIR and MKIRA blocks integrate **multi-kernel depth-wise convolutions (MKDC)** and **Convolutional Multi-Focal Attention (CMFA)** to enhance segmentation performance. MKDC adapts to diverse spatial contexts, while CMFA focuses on refining critical features.
> > >
> > > - **Empirical Validation**: Our Ablation Studies (**see Tables 4, 5, 6, 7, and 8 in the Sections 5.1, 5.2, A.4, A.5, and A.6 of our initial submission**) quantify the contributions of each block, thus demonstrating their impact on segmentation accuracy and computational efficiency. These results validate the theoretical motivation underpinning the module designs.
> > >
> > >
> > > ### **Q4. Lastly, the paper does not discuss the limitations of UltraLightUNet. A dedicated discussion on limitations would provide readers with a more balanced understanding of the model’s practical use and potential future directions for research. Typically, there is no free lunch. Are there any limitations or tradeoffs of the method? Providing those will be helpful for readers.**
> > >
> > > We appreciate the reviewer’s suggestion to discuss the limitations of **UltraLightUNet** for a balanced understanding of its practical use. **UltraLightUNet**’s focus on extreme lightweight efficiency occasionally results in slightly lower performance compared to SOTA methods on complex datasets (e.g., Synapse in Tables 2 and 12). This tradeoff aligns with our goal of addressing resource constraints in real-time and point-of-care scenarios. In the revised manuscript, we include a dedicated discussion on limitations (in **Appendix A.12**), highlighting the efficiency-performance tradeoff, domain-specific optimizations, and the potential to explore hybrid architectures for challenging tasks. We also discuss future directions, including extending **UltraLightUNet** to tasks like 2D/3D image reconstruction, translation, enhancement, and denoising. For the reviewer’s convenience, we reproduce below the paragraph we add as an independent subsection in our revised draft (in **Appendix A.12**).
> > >
> > > **“Limitations and Future Directions:** While **UltraLightUNet** excels in computational efficiency, its focus on extreme lightweight design occasionally results in slightly lower performance compared to SOTA methods on complex datasets (e.g., Synapse). This tradeoff reflects its primary goal of addressing resource constraints in real-time and point-of-care applications.
> > >
> > > Future work will explore hybrid architectures that combine lightweight and high-capacity components to handle challenging tasks without sacrificing efficiency. Additionally, strategies like self-supervised pretraining and domain-specific optimizations can enhance its performance further. We also plan to extend **UltraLightUNet** to other dense prediction tasks, such as 2D/3D image reconstruction, translation, enhancement, and denoising. This opens pathways to broaden the **UltraLightUNet**’s applicability across various computer vision tasks.”

---

### Official Review · Reviewer_zJLj · 2024-10-28

**Soundness:** 2
**Presentation:** 2
**Contribution:** 1
**Rating:** 1
**Confidence:** 5

**Summary:**

The author proposed a 2D and 3D ultra-lightweight, multi-kernel U-shaped network for medical image segmentation, termed a UltraLightUNet. It consists of an Multi-kernel Inverted Residual (MKIR) block and an Multi-kernel Inverted Residual Attention (MKIRA) block.  MKIR was proposed to efficiently process images through multiple kernels while capturing complex spatial relationships, and MKIRA block refines and emphasizes image salient features via new sophisticated convolutional multi-focal attention mechanisms. This UltraLightUNet outperformed other methods with lower complexity.

**Strengths:**

1. The proposed network has a low number of parameters and low computational complexity than other widely used baselines and achieved promising segmentation accuracy.
2. Methods and results are thoughtfully described.

**Weaknesses:**

1. The overall novelty is low. The author mainly proposed two modules, CMFA and MKDC modules. MKDC just applied several depth-wise convolutional layers to extract features from different channels. This idea has already proposed in the Scale-Aware Modulation Meet Transformer [1]. MKDC module employed average and max pooling, and convolution layers, which are the channel-wise and spatial-wise attention. Many various attention-based modules have been proposed between 2018 and 2020 (like the CBAM [2]). The overall network is similar with [3] [4].
2. The experimental results are limited. First, authors reported FLOPs and Params to demonstrate that the network has a lower computational complexity than other baselines. It is achieved by mainly replacing convolutional layers with depth-wise convolutional layers. However, it will take longer time to train networks which employ depth-wise convolutional layers compared with those with standard convolution layers. Thus, training and test time are needed to be reported. Second, the comparison in Synapse, MSD prostate, and FETA is insufficient. Synapse is a popular benchmark, but only a few baseline methods were reported. Additionally, the performance reported for these baseline methods in this paper is much lower than the performance in the original paper. For example, Swin Unet reported 79.13 in their paper [5], but only 77.58 was reported for it in this manuscript. If authors run experiments for baselines on their own, please make sure the baseline networks have been fully optimized. Only seven 3D methods proposed before 2022 were compared in MSD prostate and FETA. However, 3D segmentation networks between 2023 and 2024 were not compared, and these networks usually achieve more superior performance with lower computational complexity.
3. The motivation is unclear. The author mentioned that several 3D segmentation networks, including 3D U-Net, SwinUNETR, 3D UX-Net, UNETR, nnU-net and nnFormer, have high computational demands, so they proposed their lightweight network. However, these baseline networks are proposed in 2021 and 2022, and in recent years many lightweight 3D segmentation networks have been proposed and this challenge has been tackled, such as [6][7][8][9][10]. However, authors didn't discuss and explore these lightweight networks. Additionally, modules in this manuscript were proposed based on the idea of ConvNeXt, but its lightweight version was not discussed. Some other depth-wise convolution-based lightweight networks were also not discussed [11].
4. The theoretical development is not solid. Authors reviewed Vision Transformers in the related work, but it is not related to the work in this manuscript. The network in the manuscript employs a convolutional neural network architecture and attention mechanisms. Additionally, insufficient theoretical development in the Method section, and it is unclear how this design improved the segmentation performance.
5. No model interpretability. The interpretability of the CMFA and MKDC modules were not discussed, such as saliency maps. It is important to understand the mechanisms of attention-based modules.
6. The overall impact is low. The overall improvement in the segmentation performance is low. For example, its best DSC score in the Polyp dataset was 93.48, but other baselines achieved 93.29 and 93.18. Its best DSC score in the Synapse dataset was 78.68, but other baselines achieved 78.40.

Minors: (1) lacking qualitative results for 3D segmentation results in Synapse, MSD Prostate, and FETA.
(2) Only overall DSC scores were reported for multi-class segmentation tasks, but organ-specific DSC scores were not reported.
(3) lack p-values and standard deviations

[1] Lin, W., Wu, Z., Chen, J., Huang, J., & Jin, L. (2023). Scale-aware modulation meet transformer. In Proceedings of the IEEE/CVF International Conference on Computer Vision (pp. 6015-6026).
[2] Woo, S., Park, J., Lee, J. Y., & Kweon, I. S. (2018). Cbam: Convolutional block attention module. In Proceedings of the European conference on computer vision (ECCV) (pp. 3-19).
[3] Rahman, M. M., & Marculescu, R. (2023). Medical image segmentation via cascaded attention decoding. In Proceedings of the IEEE/CVF Winter Conference on Applications of Computer Vision (pp. 6222-6231).
[4] Rahman, M. M., Munir, M., & Marculescu, R. (2024). Emcad: Efficient multi-scale convolutional attention decoding for medical image segmentation. In Proceedings of the IEEE/CVF Conference on Computer Vision and Pattern Recognition (pp. 11769-11779).
[5] Cao, H., Wang, Y., Chen, J., Jiang, D., Zhang, X., Tian, Q., & Wang, M. (2022, October). Swin-unet: Unet-like pure transformer for medical image segmentation. In European conference on computer vision (pp. 205-218). Cham: Springer Nature Switzerland.
[6] Pang, Y., Liang, J., Huang, T., Chen, H., Li, Y., Li, D., ... & Wang, Q. (2023). Slim UNETR: Scale hybrid transformers to efficient 3D medical image segmentation under limited computational resources. IEEE Transactions on Medical Imaging
[7] Tang, F., Ding, J., Quan, Q., Wang, L., Ning, C., & Zhou, S. K. (2024, May). Cmunext: An efficient medical image segmentation network based on large kernel and skip fusion. In 2024 IEEE International Symposium on Biomedical Imaging (ISBI) (pp. 1-5). IEEE.
[8] Yang, S., Zhang, X., Chen, Y., Jiang, Y., Feng, Q., Pu, L., & Sun, F. (2023). UcUNet: A lightweight and precise medical image segmentation network based on efficient large kernel U-shaped convolutional module design. Knowledge-Based Systems, 278, 110868.
[9] He, Y., Gao, Z., Li, Y., & Wang, Z. (2024). A lightweight multi-modality medical image semantic segmentation network base on the novel UNeXt and Wave-MLP. Computerized Medical Imaging and Graphics, 111, 102311.
[10] Lin, X., Yu, L., Cheng, K. T., & Yan, Z. (2023). BATFormer: Towards boundary-aware lightweight transformer for efficient medical image segmentation. IEEE Journal of Biomedical and Health Informatics, 27(7), 3501-3512.
[11] Yin, Y., Han, Z., Jian, M., Wang, G. G., Chen, L., & Wang, R. (2023). AMSUnet: A neural network using atrous multi-scale convolution for medical image segmentation. Computers in Biology and Medicine, 162, 107120.

**Questions:**

1. Provide more detailed and necessary experimental details, including recent works in lightweight network design and other more advanced baselines, training and test time in experiments.
2. Demonstrate more solid theoretical development
3. Discuss model interpretability

---

> ### Author Response · Authors · 2024-11-25
> **Response to the comments of Reviewer zJLj: Overall novelty (Part1)**
>
> We thank the reviewer for their detailed feedback and for pointing out areas for improvement. Below, we address all the comments and clarify our approach:
>
> ### **Q1. The overall novelty is low. The author mainly proposed two modules, CMFA and MKDC modules. MKDC just applied several depth-wise convolutional layers to extract features from different channels. This idea has already been proposed in the Scale-Aware Modulation Meet Transformer [1]. MKDC module employed average and max pooling, and convolution layers, which are the channel-wise and spatial-wise attention. Many various attention-based modules have been proposed between 2018 and 2020 (like the CBAM [2]). The overall network is similar with [3] [4].**
>
> ---
>
> **Response:**
>
> **UltraLightUNet** introduces significant contributions both at **architecture** and **module** levels. At the **architecture level**, UltraLightUNet offers a fully integrated, lightweight encoder-decoder design, specifically tailored for resource-constrained scenarios, thus achieving competitive segmentation performance across both 2D and 3D tasks. At the **module level**, UltraLightUNet incorporates innovative designs like the **Multi-Kernel Inverted Residual (MKIR)** and **Multi-Kernel Inverted Residual Attention (MKIRA)** blocks, enabling efficient feature extraction and refinement with minimal computational cost. Below, we provide details on these contributions.
>
> ---
>
> ### **1. New End-to-End Lightweight Design**
>
> **UltraLightUNet** is a novel encoder-decoder architecture built entirely from scratch to ensure extreme lightweight efficiency. It is specifically designed for resource-constrained scenarios, including real-time medical diagnostics and point-of-care applications. Both the encoder and decoder are specifically optimized with novel lightweight modules, eliminating the need for pre-trained components while maintaining competitive performance across diverse tasks.
>
> - **Encoder Design**: The encoder is built using our proposed **Multi-Kernel Inverted Residual (MKIR) Block**, which performs efficient feature extraction through a combination of **multi-kernel depth-wise convolutions** and an inverted residual structure. This ensures adaptable feature extraction with minimal computational overhead, supporting both local (small kernels) and global (large kernels) context extraction.
>
> - **Decoder Design**: The decoder is constructed with our novel **Multi-Kernel Inverted Residual Attention (MKIRA) Block**, which combines **Convolutional Multi-Focal Attention (CMFA)** for local attention and **MKIR** for multi-kernel refinement. Additionally, the decoder employs a **Grouped Attention Gate (GAG)** for efficient skip connection aggregation and the **simple bilinear upsampling**, thus ensuring lightweight refinement and reconstruction of segmentation outputs.
>
> With this integrated design, **UltraLightUNet** achieves unmatched efficiency with only **0.316M parameters and 0.314 GFLOPs** for its 2D base model, and **0.453M parameters and 3.42 GFLOPs** for its 3D base model. This is a significant advancement compared to related methods like **SAMT** the reviewer mentions (32M parameters, 7.7 GFLOPs), **CASCADE** (34.12M parameters, 7.62 GFLOPs), and **EMCAD** (26.76M parameters, 5.6 GFLOPs), all of which depend on pre-trained encoders or computationally expensive modules. In contrast, UltraLightUNet’s lightweight design specifically addresses the computational constraints of real-world applications in point-of-care scenarios without sacrificing performance.

---

> > ### Author Response · Authors · 2024-11-25
> > **Response to the comments of Reviewer zJLj: Overall novelty (Part2)**
> >
> > ### 2. New Modules
> >
> > ### *2.1 Multi-Kernel Inverted Residual (MKIR) Block*
> >
> > The **MKIR block** introduces a novel feature extraction approach by leveraging **Multi-Kernel Depth-Wise Convolutions (MKDC)** to efficiently balance local and global context extraction. With its **multi-kernel design**, MKIR supports both **k₁ = k₂ (same-size kernels)** and **k₁ ≠ k₂ (different-size kernels)**, thus enabling adaptability across diverse spatial contexts. The MKIR’s inverted residual structure minimizes the computational overhead while maintaining the representational power.
> >
> > In its **3D version**, MKIR extends multi-kernel convolutions to volumetric data, thus allowing efficient feature extraction for 3D tasks while preserving its lightweight nature. This is a significant contribution to the SOTA with high relevance in medical bioimaging (we show our results on multiple datasets – see Tables 3, 10, 12 in the paper).
> >
> > #### In contrast:
> > - **SAMT’s Scale-Aware Modulation (SAM)** relies on computationally intensive transformer-based attention, limited to 2D tasks, with no 3D extension.
> >
> > - **CASCADE’s ConvBlock** relies on standard 3x3 convolutions (i.e., not multi-kernel), which lack adaptability and efficiency. CASCADE also lacks a 3D version of ConvBlock, preventing its applicability to volumetric tasks.
> >
> > - **EMCAD’s Multi-Scale Convolution Block (MSCB)** employs multi-scale convolutions (**k₁ ≠ k₂**) in the decoder, but it is not used in the encoder and has no 3D extension.
> >
> > - **ConvNeXt** employs large-kernel depth-wise convolutions (7x7) that effectively capture large contexts, but lacks flexibility for smaller features. In contrast, MKIR balances both small and large contexts efficiently.
> >
> >
> > ### *2.2 Multi-Kernel Inverted Residual Attention (MKIRA) Block*
> >
> > The **MKIRA block** performs **multi-kernel refinement of attention** by combining the **Convolutional Multi-Focal Attention (CMFA)** module with **MKIR** in a sequential manner. CMFA first applies max and average pooling to compute local attention across spatial and channel dimensions, thus enhancing critical features. MKIR then refines these features using multi-kernel convolutions, ensuring efficient and effective feature refinement.
> >
> > The **3D version of MKIRA** adapts CMFA and MKIR for volumetric data, employing 3D pooling and multi-kernel convolutions to handle complex 3D medical imaging tasks.
> >
> > #### In contrast:
> > - **SAMT’s Scale-Aware Modulation (SAM)** employs global attention for scale diversity, but lacks efficiency and scalability to 3D tasks.
> >
> > - **CASCADE’s Convolutional Attention Module (CAM)** uses 3x3 convolutions for spatial-channel attention in the decoder, increasing computational cost and lacking a 3D version.
> >
> > - **EMCAD’s Multi-Scale Convolutional Attention Module (MSCAM)** uses multi-scale convolutions for attention, but is restricted to 2D tasks and has no 3D extension.
> >
> >
> > ### 3. 3D Versatility
> >
> > **UltraLightUNet** goes beyond existing 2D-only methods like **SAMT, CASCADE, EMCAD, ConvNeXt, CMUNeXt, SwinUNet, TransUNet**, etc., by providing **novel 3D versions** of both the architecture and its modules (MKIR and MKIRA), tailored for volumetric medical imaging. While methods like SAMT, CASCADE, and EMCAD lack 3D extensions, UltraLightUNet introduces a lightweight 3D base model that achieves **new SOTA efficiency** with just **0.453M parameters and 3.42 GFLOPs**, compared to existing 3D methods that typically require **hundreds of Giga FLOPs**.
> >
> > This makes UltraLightUNet uniquely suited for **real-time 3D medical imaging** tasks in point-of-care scenarios, such as organ and tumor segmentation, where computational constraints are critical.

---

> > > ### Author Response · Authors · 2024-11-25
> > > **Response to the comments of Reviewer zJLj: Experimental results are limited (Part1)**
> > >
> > > ### **Q2.1. The experimental results are limited. First, authors reported FLOPs and Params to demonstrate that the network has a lower computational complexity than other baselines. However, it will take longer time to train networks which employ depth-wise convolutional layers compared with those with standard convolution layers. Thus, training and test time are needed to be reported.**
> > >
> > > ---
> > >
> > > **Response:**
> > >
> > > We appreciate the reviewer’s observation regarding the potential impact of depth-wise convolutions on training and test times.
> > >
> > > **However**, our design prioritizes **extreme lightweight efficiency** for resource-constrained environments by leveraging depth-wise convolutions in the **MKIR** and **MKIRA** blocks. Per reviewer’s request, we present the training and test times for UltraLightUNet (in **Table R3**) which is comparable to other lightweight baseline methods when evaluated on a NVIDIA A6000 GPU of 48GB memory. Additionally, the extremely small #FLOPs and #Params (e.g., **0.316M Params and 0.314 GFLOPs** for our 2D model) ensure practical usability in real-time applications.
> > >
> > > **Table R3**: Computational complexity (#Params, #FLOPs, Training Time (Sec), Inference Time (Mili Sec)) comparisons of different architectures including our UltraLightUNet. We train each model for 200 epochs using a batch size of 16 with a total 1000 sample images of resolution 256×256 to get the **total training time (sec.)** on a NVIDIA RTX A6000 GPU. While we run the inference on 500 samples on the same GPU with a batch size of 1 and report the **average inference time** (ms) per image. We report the average DICE score of six binary segmentation datasets here for reasonable comparison.
> > >
> > > | Architecture           | #Params (M) $\downarrow$ | #FLOPs (G) $\downarrow$ | Training Time (sec.) $\downarrow$ | Inference Time (ms) $\downarrow$ | Avg DICE (%) $\uparrow$ |
> > > |-------|------|------|----------|---------|---------|
> > > | UNet                  | 34.53       | 65.53      | 1732.11              | 0.0084               | 87.28        |
> > > | AttUNet               | 34.88       | 66.64      | 1988.31              | 0.0092               | 87.86        |
> > > | UNet++                | 9.16        | 34.65      | 794.38               | 0.0073               | 88.16        |
> > > | PraNet                | 32.55       | 6.93       | 685.37               | 0.0156               | 87.79        |
> > > | DeepLabv3+            | 39.76       | 14.92      | 695.82               | 0.0078               | 89.15        |
> > > | UACANet               | 69.16       | 31.51      | 850.53               | 0.0231               | 87.81        |
> > > | TransUNet             | 105.32      | 38.52      | 1523.68              | 0.0153               | 89.59        |
> > > | SwinUNet              | 27.17       | 6.20       | 828.99               | 0.0124               | 88.84        |
> > > | DeformableLKA         | 102.76      | 26.03      | 5450.26              | 0.0663               | **89.92**        |
> > > | MedT                  | 1.57        | 1.95       | 7138.91              | 0.1191               | 82.42        |
> > > | Rolling-UNet-S        | 1.78        | 2.10       | 635.69               | 0.0175               | 87.36        |
> > > | CMUNeXt               | 0.418       | 1.09       | 450.86               | **0.0057**               | 88.25        |
> > > | UNeXt                 | 1.47        | 0.57       | **216.26**               | $\underline{0.0058}$               | 86.06        |
> > > | EGE-UNet              | 0.054       | 0.072      | 360.69               | 0.0099               | 83.82        |
> > > | UltraLight_VM_UNet    | $\underline{0.050}$       | **0.060**      | 318.04               | 0.0102               | 85.53        |
> > > | UltraLightUNet-T (**Ours**) | **0.027**  | $\underline{0.062}$ | $\underline{312.08}$          | 0.0071           | 87.87    |
> > > | UltraLightUNet-S (**Ours**) | 0.093  | 0.125 | 348.78          | 0.0072           | 89.10    |
> > > | UltraLightUNet (**Ours**)   | 0.316  | 0.314 | 474.02          | 0.0072           | $\underline{89.75}$    |
> > >
> > > **Table R3** highlights the trade-offs between training/inference time and efficiency. UltraLightUNet variants achieve competitive or superior DICE scores with significantly fewer #Params and #FLOPs compared to all other architectures. For example, UltraLightUNet (474.02 sec training, 0.0072 ms inference) achieves 89.75% DICE with only 0.316M params and 0.314G FLOPs, thus outperforming heavier models like DeepLabv3+ (89.15% DICE, 39.76M params, 14.92G FLOPs) and TransUNet (89.59% DICE, 105.32M params, 38.52G FLOPs).
> > >
> > > While depth-wise convolutions slightly increase the training time due to reduced parallelism, they enable extreme computational efficiency (#Params and #FLOPs), thus making UltraLightUNet ideal for resource-constrained environments.
> > >
> > > Finally, the inference times remain competitive with lightweight baselines like CMUNeXt (0.0057 ms, 88.25% DICE), while offering higher DICE score. We will add this Table in the Appendix of our revised draft.

---

> > > > ### Author Response · Authors · 2024-11-25
> > > > **Response to the comments of Reviewer zJLj: Experimental results are limited (Part2)**
> > > >
> > > > ### **Q2.2. Second, the comparison in Synapse, MSD prostate, and FETA is insufficient. Synapse is a popular benchmark, but only a few baseline methods were reported. Only seven 3D methods proposed before 2022 were compared in MSD prostate and FETA. However, 3D segmentation networks between 2023 and 2024 were not compared, and these networks usually achieve more superior performance with lower computational complexity.**
> > > >
> > > > ---
> > > >
> > > > **Response:**
> > > >
> > > > We understand the reviewer’s concerns regarding the thoroughness of our comparisons, particularly with recent 3D segmentation methods (2023–2024).
> > > >
> > > > - **For Synapse**, we already compared UltraLightUNet with **12 baseline methods for 2D segmentation** (Table 2) and **7 baseline methods for 3D segmentation** (Table 12). These comparisons include well-established baselines (e.g., TransUNet, SwinUNet, nn-Former, SwinUNETR) and more recent methods like **3D UX-Net** (Lee et al., 2022), **CMUNeXt** (Tang et al., 2023), and **Rolling-UNet** (Liu et al., 2024).
> > > >
> > > > - Additionally, we have included results for a recent 3D method, **SlimUNETR (Pang et al., 2023)**, in **Table R4 below**, which shows that **UltraLightUNet3D-S** achieves **10.19% higher DICE** score on Task05 Prostate, **0.17% higher on FETA**, **1.47% higher on Synapse 8-organ**, and **2.25% higher on Synapse 13-organ** while using **9.9x fewer #parameters and 5.9x fewer #FLOPs** than SlimUNETR. This comparison highlights the better performance and efficiency tradeoff achieved by UltraLightUNet.
> > > >
> > > > **Table R4**: Comparison of UltraLightUNet with SlimUNETR (Pang et al., 2023) [6]. We present the DICE scores (%) on our data-splits with an input resolution of 96x96x96, while optimizing the hyper-parameters of SlimUNETR.
> > > >
> > > > | Architectures             | #Params     | #FLOPs    | Task05_Prostate | FETA  | Synapse 8-organ | Synapse 13-organ |
> > > > |---------------------------|-------------|-----------|-----------------|-------|-----------------|------------------|
> > > > | SlimUNETR                | 1.78M       | 11.99G    | 59.01           | 86.98 | 80.42           | 72.56            |
> > > > | **UltraLightUNet3D-S (Ours)** | **0.163M** | **2.03G** | **69.20**       | 87.15 | 81.89           | 74.81            |
> > > > | UltraLightUNet3D (Ours)   | 0.453M      | 3.42G     | 70.52           | 87.92 | 81.87           | 76.33            |
> > > > | UltraLightUNet3D-M (Ours) | 1.42M       | 7.1G      | 71.51           | **88.40** | **82.58**       | **77.46**        |
> > > >
> > > > ### **Q2.3. Additionally, the performance reported for these baseline methods in this paper is much lower than the performance in the original paper. For example, Swin Unet reported 79.13 in their paper [5], but only 77.58 was reported for it in this manuscript.**
> > > >
> > > > ---
> > > >
> > > > **Response:**
> > > >
> > > > We appreciate the reviewer’s attention to discrepancies in reported baseline performance (e.g., **Swin Unet: 77.58 vs. 79.13**).
> > > >
> > > > **Clarification**: The reported results for **Swin Unet** and **TransUNet** in our manuscript were taken directly from the **CASCADE paper [3]**, ensuring consistency across all the reported baselines. These results may differ from those reported in the original papers due to differences in the experimental setups, such as data splits or preprocessing strategies. To maintain consistency in evaluation, we opted not to re-train these baselines ourselves.
> > > >
> > > > To address this reviewer’s concerns (**Q2**), we will make the following updates in the revised manuscript:
> > > >
> > > > - **Training and Test Time Reporting**:
> > > >    - Include a detailed comparison of training and inference times across UltraLightUNet and baseline methods on the same hardware platform.
> > > >    - Discuss the trade-offs between computational complexity, FLOPs, and training/inference time for depth-wise convolutions versus standard convolutions.
> > > >
> > > > - **Expanded Comparisons**:
> > > >    - Emphasize the addition of **SlimUNETR** results, highlighting that UltraLightUNet outperforms SlimUNETR both in performance and efficiency.
> > > >
> > > > - **Clarifying Baseline Performance**:
> > > >    - Explicitly state that results for certain baselines (e.g., SwinUNet, TransUNet) are taken from the **CASCADE paper [3]** and explain the potential experimental setup differences.

---

> > ### Comment · Reviewer_zJLj · 2024-11-25
> >
> > Thanks for authors' response.
> >
> > The novelty is low since the way to solve the problem has been widely explored and is the same, such as employing depth-wise convolution for lightweight design and splitting channels for isolated convolutions. Thus, putting much efforts on applying the exactly same way to solve the same problem is not very interesting, and this application from N to N+1 does not make impact on the medical image segmentation tasks.
> >
> > The architectural design cannot be considered as a novel design since the overall design (U-shaped encoder-decoder) is always used in the medical image segmentation. Incorporating several modules into this architecture does not revolutionize the architectural design.

---

> ### Author Response · Authors · 2024-11-25
> **Response to the comments of Reviewer zJLj: Motivation (Part1)**
>
> ### **Q3.1. The motivation is unclear. The author mentioned that several 3D segmentation networks, including 3D U-Net, SwinUNETR, 3D UX-Net, UNETR, nnU-net and nnFormer, have high computational demands, so they proposed their lightweight network. However, these baseline networks are proposed in 2021 and 2022, and in recent years many lightweight 3D segmentation networks have been proposed and this challenge has been tackled, such as [6][7][8][9][10]. However, authors didn't discuss and explore these lightweight networks.**
>
> ---
>
> **Response:**
>
> We thank the reviewer for their thoughtful feedback regarding the clarity of the motivation and exploration of lightweight segmentation networks. Below, we provide clarifications and address the concerns raised:
>
> **Motivation:** The primary motivation of our work is to address the **computational demands of existing 2D segmentation networks** by proposing an **ultra-lightweight architecture** that achieves competitive performance with significantly reduced parameters and FLOPs. Recognizing the versatility of our architecture, we then **extend it to 3D** by incorporating volumetric modules, ensuring that the same architecture works efficiently for both 2D and 3D segmentation tasks. This unified, resource-efficient approach allows UltraLightUNet to cater to a broader range of applications, including 3D volumetric tasks, with extreme efficiency.
>
> **Addressing Missing Lightweight Networks:** The reviewer mentioned several lightweight segmentation networks ([6], [7], [8], [9], [10]) that were not fully explored in our initial submission. Our clarification is as follows:
>
> - **[6] (SlimUNETR, Pang et al., 2023)**: To strengthen our evaluation, in **Table R2 above** (also in Tables 3, 12 of the revised draft), we have now added results for **SlimUNETR** (a recent lightweight 3D segmentation method). Table R2 shows that **UltraLightUNet3D-S** achieves **10.19% higher DICE** score on Task05 Prostate, **0.17% higher on FETA**, **1.47% higher on Synapse 8-organ**, and **2.25% higher on Synapse 13-organ** while using **9.9x fewer #parameters and 5.9x fewer #FLOPs** than SlimUNETR. Larger variants like **UltraLightUNet3D** improve further, with **UltraLightUNet3D-M** showing a total improvement of **12.50% on Task05 Prostate**, **1.42% on FETA**, **2.16% on Synapse 8-organ**, and **4.90% on Synapse 13-organ** compared to SlimUNETR. These results demonstrate **UltraLightUNet3D’s significant improvements** in performance while maintaining exceptional computational efficiency and cost.
>
> - **[7] (CMUNeXt)**: Already included in our initial submission (see Table 1, 2, 11 in our initial submission); again, UltraLightUNet demonstrates its superior performance and efficiency compared to CMUNeXt.
> - **[8], [9], [10]**: These methods are for 2D segmentation and do not extend to volumetric 3D segmentation tasks. While they provide valuable contributions for 2D segmentation tasks, their lack of 3D applicability makes them less relevant for a direct comparison with UltraLightUNet 3D version.
>
> With the addition of **SlimUNETR** in our rebuttal and revised manuscript, along with the necessary comparisons for 2D methods, we believe our paper provides comprehensive coverage of both 2D and 3D lightweight segmentation networks.
>
> ---

---

> > ### Author Response · Authors · 2024-11-25
> > **Response to the comments of Reviewer zJLj: Motivation (Part2)**
> >
> > ### **Q3.2. Additionally, modules in this manuscript were proposed based on the idea of ConvNeXt, but its lightweight version was not discussed. Some other depth-wise convolution-based lightweight networks were also not discussed [11].**
> >
> > We thank the reviewer for their feedback regarding the relationship between our proposed modules and ConvNeXt, as well as the discussion of other depth-wise convolution-based lightweight networks. Below, we address these concerns:
> >
> > **1. Relationship with ConvNeXt:** UltraLightUNet was **not directly inspired by ConvNeXt**. While both employ depth-wise convolutions, **UltraLightUNet introduces multi-kernel depth-wise convolutions (MKDC)**, supporting both same-sized (\(k_1 = k_2\)) and different-sized (\(k_1 \neq k_2\)) kernels for adaptable context extraction. Unlike ConvNeXt which relies on large kernels (e.g., 7x7) for global contexts, MKDC provides flexibility to handle **both local and global contexts effectively**.
> >
> > Furthermore, **UltraLightUNet integrates MKDC into new lightweight modules**, such as **MKIR** and **MKIRA**, tailored for efficient 2D and 3D medical segmentation tasks. While ConvNeXt targets general-purpose computer vision tasks, our work addresses the domain-specific needs of medical imaging in resource-constrained environments.
> >
> > ---
> >
> > **2. Discussion of Other Depth-Wise Convolution-Based Lightweight Networks:** The networks mentioned in [11] primarily focus on 2D segmentation tasks. In contrast, **UltraLightUNet extends the depth-wise convolution paradigm to 3D tasks** with novel volumetric modules like **3D MKIR and 3D MKIRA**, enabling lightweight and scalable segmentation in volumetric data. This focus on unified 2D-3D design distinguishes our work from generic depth-wise convolution-based architectures.
> >
> > In the revised manuscript, we will briefly discuss such networks and highlight the unique contributions of UltraLightUNet in adapting depth-wise convolutions for 3D medical imaging.

---

> > > ### Comment · Reviewer_zJLj · 2024-11-25
> > >
> > > Although the authors did not inspired by the ConvNeXt, its many variants and its way to using depth-wise convolutions to design lightweight modules have been widely explored in both general computer vision tasks and medical image analysis tasks. Thus, it is necessary to discuss them. Moreover, incorporating one or two more layers to your module is not a novel design and does not make impact even though your modules have a new name.

---

> > > > ### Author Response · Authors · 2024-12-02
> > > > **Official Comment by Authors: Part4**
> > > >
> > > > ### **C3. Although the authors did not inspired by the ConvNeXt, its many variants and its way to using depth-wise convolutions to design lightweight modules have been widely explored in both general computer vision tasks and medical image analysis tasks. Thus, it is necessary to discuss them. Moreover, incorporating one or two more layers to your module is not a novel design and does not make impact even though your modules have a new name.**
> > > >
> > > > **Response:**
> > > >
> > > > We thank the reviewer for their follow-up comments and the opportunity to clarify our contributions further. Below, we address the concerns raised.
> > > >
> > > >
> > > > ### **Addressing Similarities with ConvNeXt**
> > > >
> > > > While ConvNeXt and its variants have explored depth-wise convolutions for lightweight module design, we emphasize that our work is not inspired by ConvNeXt. Our **Multi-Kernel Inverted Residual (MKIR)** and **Multi-Kernel Inverted Residual Attention (MKIRA)** blocks address the specific challenges of medical image segmentation in resource-constrained environments, a focus completely new compared to the ConvNeXt’s general-purpose design.
> > > >
> > > > Key differences include:
> > > >
> > > > 1. Multi-Kernel vs. Large-Kernel Design: ConvNeXt relies on large-kernel depth-wise convolutions (e.g., $7 \times 7$) to capture global context. In contrast, our **multi-kernel approach** supports both $k_1​=k_2​$ (same-size kernels) and $k_1 \neq k_2$​ (different-size kernels), enabling adaptable feature extraction for segmenting objects of varying sizes. This adaptability is critical for diverse segmentation tasks where both local and global features are essential (see the concrete example in our response to reviewer **jeKK** of this rebuttal).
> > > >
> > > > 2. Application-Specific Challenges: Our design explicitly targets medical image segmentation with extreme efficiency, focusing on lightweight computation for *real-time applications* like point-of-care diagnostics. ConvNeXt’s design neither addresses these specific constraints, nor provides 3D extensions for volumetric tasks.
> > > >
> > > >
> > > > ### **Beyond Incremental Layer Design**
> > > >
> > > > We understand the concern about adding layers not being inherently novel. However, our contributions go beyond incremental changes by introducing:
> > > >
> > > > **1. Conceptually New Modules:**
> > > >
> > > > - **MKIR Block** integrates multi-kernel depth-wise convolutions with an inverted residual structure, ensuring lightweight and efficient feature extraction. The Equation 2 is new and provides a better mathematical basis for efficient computations compared to SOTA; this is what makes the MKIR block new.
> > > >
> > > > - **MKIRA Block** integrates CMFA with MKIR blocks for **multi-kernel refinement of attention**, thus offering a lightweight yet effective attention mechanism distinct from existing designs. The integration of these blocks addresses both accuracy and efficiency objectives in novel ways.
> > > >
> > > > **2. 3D Extensions:** All our modules, including MKIR and MKIRA, are extended to 3D for volumetric medical image segmentation, introducing entirely new designs tailored for 3D tasks. This is a significant advancement not present in ConvNeXt or other similar works like UNeXt, EGE-UNet, and Rolling-UNet.
> > > >
> > > >
> > > > ### **Comparing Contributions with 3D UX-Net (ICLR 2023)**
> > > >
> > > > The reviewer’s concerns about architectural novelty highlight the importance of placing our contributions in the proper context. For example:
> > > >
> > > > - **3D UX-Net** (ICLR 2023) introduces a 3D extension of ConvNeXt’s large-kernel module, but retains a **computationally heavy decoder** directly adapted from SwinUNETR (Hatamizadeh et al., 2021) and UNETR (Hatamizadeh et al., 2022). It focuses solely on 3D segmentation tasks without lightweight optimizations for resource-constrained scenarios.
> > > >
> > > > - In contrast, our **UltraLightUNet** provides both **2D and 3D lightweight designs**, with entirely new 3D versions of our modules (MKIR, MKIRA, GAG, etc.). Our contributions enable both 2D and 3D segmentation with significantly lower computational costs while achieving high accuracy, making our work eminently suited for real-time applications.
> > > >
> > > > We hope these clarifications, along with the 3D-specific contributions and comparisons, address the reviewer’s concerns and highlight the novelty and impact of our work. Thank you again for your thoughtful feedback and consideration.

---

> ### Author Response · Authors · 2024-11-25
> **Response to the comments of Reviewer zJLj: Theoretical development**
>
> ### **Q4.1. The theoretical development is not solid. Authors reviewed Vision Transformers in the related work, but it is not related to the work in this manuscript. The network in the manuscript employs a convolutional neural network architecture and attention mechanisms.**
>
> We thank the reviewer for their thoughtful feedback and for pointing out areas for clarification regarding the theoretical development and relevance of Vision Transformers in our work. Below, we address this comment:
>
> We included a discussion of Vision Transformers in the Related Work section to provide context on **computationally expensive approaches**, such as **TransUNet** and **SwinUNet**, which are widely used and popular in medical image segmentation. These methods demonstrate strong performance, but come with high computational demands due to their reliance on transformer-based architectures.
>
> Our proposed **multi-kernel design with depth-wise convolutions** directly addresses these limitations by ensuring **lightweight efficiency** while maintaining high performance. This aligns with the motivation of our work, which emphasizes computational efficiency for resource-constrained environments. The inclusion of Vision Transformers in the Related Work highlights this contrast and establishes the relevance of our lightweight design in comparison to transformer-based approaches.
>
> ### **Q4.2. Additionally, insufficient theoretical development in the Method section, and it is unclear how this design improved the segmentation performance.**
>
> **UltraLightUNet’s design is grounded in clear theoretical concepts**, which are explained in the Method section and validated in the Ablation Study section:
>
> - **Theoretical Basis**: Most existing architectures in computer vision rely on theoretical concepts to justify their design choices (e.g., Vision Transformers focus on self-attention). Similarly, our approach employs the **multi-kernel trick** to improve segmentation performance and **depth-wise convolutions** for lightweight computation. While our contribution is not theoretical in nature, these foundational concepts ensure that UltraLightUNet achieves both high performance and extreme efficiency.
>
> - **Method Section**: We elaborated on how **Multi-Kernel Inverted Residual (MKIR)** and **Multi-Kernel Inverted Residual Attention (MKIRA)** blocks work together to balance performance and efficiency. The use of **multi-kernel depth-wise convolutions (MKDC)** enables adaptable feature extraction, while **Convolutional Multi-Focal Attention (CMFA)** enhances critical features.
>
> - **Empirical Validation**: The **Ablation Study** (Tables 4 and 5 in our initial submission) demonstrates the impact of each module on segmentation performance. These results show how the theoretical design improves segmentation accuracy while maintaining computational efficiency.
>
> To address this reviewer’s concern, we plan to further emphasize the connection between the theoretical design and its performance improvements in the revised manuscript, explicitly linking the multi-kernel design to segmentation accuracy.

---

> > ### Author Response · Authors · 2024-11-25
> > **Response to the comments of Reviewer zJLj: Model interpretability**
> >
> > ### **Q5. No model interpretability. The interpretability of the CMFA and MKDC modules were not discussed, such as saliency maps. It is important to understand the mechanisms of attention-based modules.**
> >
> > While our primary focus was on achieving high performance with lightweight efficiency, we agree that understanding these mechanisms is critical. The **CMFA module** enhances feature refinement by applying max and average pooling across spatial and channel dimensions, selectively focusing on critical features, while the **MKDC module** employs multi-kernel depth-wise convolutions to balance local and global context extraction. These mechanisms are validated through improved segmentation accuracy in our Ablation Study (Tables 4 and 5 in the main paper).
> >
> > To address this reviewer’s concern, we have included **activation heatmaps** to visualize how these modules focus on relevant features in **Fig. 4** in the Appendix of our revised draft. In Fig. S1, we plot the average activation heatmaps for all channels in high-resolution layers, focusing on Encoder Stage 1 (ES1) and Decoder Stage 1 (DS1). In ES1, the MKIR block attends to diverse regions, including the polyp region, thus capturing broad spatial features as expected in the initial stages of the encoder. In contrast, the CMFA layer in DS1 sharpens attention, thus focusing more locally on the polyp region. Subsequently, the MKDC within the MKIR block of DS1 further refines these attended features, thus concentrating exclusively on the polyp region (indicated by deep red areas). This progression highlights the effectiveness of our architecture in capturing and refining features, thus resulting in a segmentation map that strongly overlaps with the ground truth.

---

> > > ### Author Response · Authors · 2024-11-25
> > > **Response to the comments of Reviewer zJLj: Overall impact**
> > >
> > > ### **Q6. The overall impact is low. The overall improvement in the segmentation performance is low. For example, its best DSC score in the Polyp dataset was 93.48, but other baselines achieved 93.29 and 93.18. Its best DSC score in the Synapse dataset was 78.68, but other baselines achieved 78.40.**
> > >
> > > We respectfully disagree with this assessment. While the performance gains in DICE scores may appear small, the **key impact** of our method lies in achieving these DICE results with **extreme computational efficiency**:
> > >
> > > - **Polyp Dataset**: UltraLightUNet achieves **93.48% DICE** with only **0.316M parameters and 0.314G FLOPs**, compared to UACANet (93.29%, 69.16M params, 31.51G FLOPs) and TransUNet (93.18%, 105.32M params, 38.52G FLOPs). UltraLightUNet is **219x smaller** and **122x more efficient** than UACANet.
> > >
> > > - **Synapse Dataset**: UltraLightUNet achieves **78.68% DICE** with **0.316M params and 0.257G FLOPs** (224x224 input), compared to DeepLabv3+ (78.40%, 39.76M params, 11.456 FLOPs) and TransUNet (78.40%, 105.32M params). UltraLightUNet uses **125x fewer parameters** and **44x fewer FLOPs** than DeepLabv3+.
> > >
> > > These are **orders of magnitude improvements** across the board! These results demonstrate UltraLightUNet’s significant impact in resource-constrained settings, achieving competitive performance with only a **fraction** of the computational cost.
> > >
> > > Thank you for your constructive feedback.
> > >
> > >
> > > ### **References**
> > >
> > > [1] Lin, W., Wu, Z., Chen, J., Huang, J., & Jin, L. (2023). Scale-aware modulation meet transformer. In Proceedings of the IEEE/CVF International Conference on Computer Vision (pp. 6015-6026).
> > >
> > > [2] Woo, S., Park, J., Lee, J. Y., & Kweon, I. S. (2018). Cbam: Convolutional block attention module. In Proceedings of the European conference on computer vision (ECCV) (pp. 3-19).
> > >
> > > [3] Rahman, M. M., & Marculescu, R. (2023). Medical image segmentation via cascaded attention decoding. In Proceedings of the IEEE/CVF Winter Conference on Applications of Computer Vision (pp. 6222-6231).
> > >
> > > [4] Rahman, M. M., Munir, M., & Marculescu, R. (2024). Emcad: Efficient multi-scale convolutional attention decoding for medical image segmentation. In Proceedings of the IEEE/CVF Conference on Computer Vision and Pattern Recognition (pp. 11769-11779).
> > >
> > > [5] Cao, H., Wang, Y., Chen, J., Jiang, D., Zhang, X., Tian, Q., & Wang, M. (2022, October). Swin-unet: Unet-like pure transformer for medical image segmentation. In European conference on computer vision (pp. 205-218). Cham: Springer Nature Switzerland.
> > >
> > > [6] Pang, Y., Liang, J., Huang, T., Chen, H., Li, Y., Li, D., ... & Wang, Q. (2023). Slim UNETR: Scale hybrid transformers to efficient 3D medical image segmentation under limited computational resources. IEEE Transactions on Medical Imaging
> > >
> > > [7] Tang, F., Ding, J., Quan, Q., Wang, L., Ning, C., & Zhou, S. K. (2024, May). Cmunext: An efficient medical image segmentation network based on large kernel and skip fusion. In 2024 IEEE International Symposium on Biomedical Imaging (ISBI) (pp. 1-5). IEEE.
> > >
> > > [8] Yang, S., Zhang, X., Chen, Y., Jiang, Y., Feng, Q., Pu, L., & Sun, F. (2023). UcUNet: A lightweight and precise medical image segmentation network based on efficient large kernel U-shaped convolutional module design. Knowledge-Based Systems, 278, 110868.
> > >
> > > [9] He, Y., Gao, Z., Li, Y., & Wang, Z. (2024). A lightweight multi-modality medical image semantic segmentation network base on the novel UNeXt and Wave-MLP. Computerized Medical Imaging and Graphics, 111, 102311.
> > >
> > > [10] Lin, X., Yu, L., Cheng, K. T., & Yan, Z. (2023). BATFormer: Towards boundary-aware lightweight transformer for efficient medical image segmentation. IEEE Journal of Biomedical and Health Informatics, 27(7), 3501-3512.
> > >
> > > [11] Yin, Y., Han, Z., Jian, M., Wang, G. G., Chen, L., & Wang, R. (2023). AMSUnet: A neural network using atrous multi-scale convolution for medical image segmentation. Computers in Biology and Medicine, 162, 107120.

---

> ### Comment · Reviewer_zJLj · 2024-11-25
>
> "The reported results for Swin Unet and TransUNet in our manuscript were taken directly from the CASCADE paper [3], ensuring consistency across all the reported baselines."
>
> It is not fair to copy results of baselines from other papers, and it is better to check the original paper to utilize the hyper-parameters in their original papers. It is not a good way to train your model to the optimal one while not use optimal hyper-parameters to train baseline models. I am very confused why you copied results from the CASCADE paper. This CASCADE paper is not a benchmark paper, so copying results from this paper will not ensure the consistency. These baselines were not proposed in the CASCADE paper, and copying results from this paper will lead to lower segmentation performance and unfair comparison. These baselines have not reached their highest performance, and they still have large potential to reach the higher performance. Therefore, it cannot demonstrate the superiority of the UltraLightUNet.

---

> > ### Author Response · Authors · 2024-12-02
> > **Official Comment by Authors: Part3**
> >
> > ### **C2. "The reported results for Swin Unet and TransUNet in our manuscript were taken directly from the CASCADE paper [3], ensuring consistency across all the reported baselines." It is not fair to copy results of baselines from other papers, and it is better to check the original paper to utilize the hyper-parameters in their original papers. It is not a good way to train your model to the optimal one while not use optimal hyper-parameters to train baseline models. I am very confused why you copied results from the CASCADE paper. This CASCADE paper is not a benchmark paper, so copying results from this paper will not ensure the consistency. These baselines were not proposed in the CASCADE paper, and copying results from this paper will lead to lower segmentation performance and unfair comparison. These baselines have not reached their highest performance, and they still have large potential to reach the higher performance. Therefore, it cannot demonstrate the superiority of the UltraLightUNet.**
> >
> > **Response:**
> >
> > We thank the reviewer for highlighting this concern and agree that reporting baseline results from original papers should be enforced. In response to the reviewer’s feedback, we have revised our manuscript to directly report the DICE score for SwinUNet from its original paper (Cao et al., 2021) for the Synapse dataset, as reflected in Table 2 (in revised submission). We note however, that this change does not impact the significance of our results at all; this is because even after reporting the results from the original paper, our UltraLightUNet-L achieves competitive performance (only 0.45% lower) while using **$7.2\times$ fewer parameters**, thus highlighting the efficiency of our approach. We believe this update enhances fairness and transparency in our comparisons, but does not change anything in terms of significance of the results. This is because, our focus in this paper is not to beat the accuracy of previous approaches, but rather to maintain or get as closely as possible to the accuracy achieved by prior approaches with significantly less resources (i.e., #Params, #FLOPS). This is how our paper redefines the SOTA in 2D and 3D image segmentation.

---

> ### Comment · Reviewer_zJLj · 2024-11-25
>
> Thanks for authors' response. I will not change my score due to the low novelty, low impact to the medical image segmentation, and too lower baseline results.

---

> > ### Author Response · Authors · 2024-12-02
> > **Official Comment by Authors: Part5**
> >
> > ### **C4. Thanks for authors' response. I will not change my score due to the low novelty, low impact to the medical image segmentation, and too lower baseline results.**
> >
> > **Response:**
> >
> > We respectfully refer the reviewer to our earlier responses; below, provide additional clarification regarding the novelty, impact, and baseline results of UltraLightUNet:
> >
> > **1. Novelty:** UltraLightUNet introduces a **new end-to-end design for both 2D and 3D architectures**, built entirely from scratch to achieve extreme efficiency. Our key contributions include:
> >
> > - **Multi-Kernel Inverted Residual (MKIR)** and **Multi-Kernel Inverted Residual Attention (MKIRA)** blocks, employs a **novel multi-kernel approach** that supports both $k_1​=k_2​$ and $k_1≠k_2$, for $k_1, k_2 \in Kernels​$. These are conceptually distinct from existing multi-scale approaches like EMCAD’s MSCB and MSCAM.
> >
> > - **New 3D extensions** for volumetric medical imaging tasks, which represent a significant advancement, as existing methods like EMCAD and ConvNeXt lack such adaptations.
> >
> > Additionally, we note that **3D UX-Net** (Lee et al., 2023), which introduced only a **3D version of ConvNeXt’s block** in the encoder and reused a computationally heavy decoder, was actually accepted at ICLR 2023! In contrast, UltraLightUNet delivers **conceptually new 2D modules** and their **corresponding lightweight 3D extensions** ($185 \times$ lower #FLOPs and $77 \times$ fewer #Params compared to 3D UX-Net), thus demonstrating a more comprehensive and impactful contribution. So, if 3D UX-Net was deemed relevant for ICLR 2023 acceptance, we believe that UltraLightUNet which provides so much better results and broader applicability is a decent start.
> >
> > **2. Impact:** UltraLightUNet addresses a critical unmet need in real-time medical imaging for resource-constrained environments, achieving **state-of-the-art accuracy with drastically lower computational costs**. This aligns with the growing trend of ultra-lightweight designs in computer vision and biomedical imaging, making our work particularly relevant to real-world applications. Simply put, our results show that UltraLightUNet is the new SOTA in this problem space.
> >
> > **3. Baseline Results:** We updated our comparisons using values from **original sources** (e.g., SwinUNet). UltraLightUNet achieves **competitive accuracy**, while using 7.2 $\times$ fewer parameters, thus underscoring its efficiency and practical applicability. Again, our challenge is not the accuracy of previous approaches, but rather the complexity of resources involved to achieve these accuracies. From this perspective, our UltraLightUNet architecture is top of the class.
> >
> > Given the novelty, rigorous experimental validation, and demonstrated practical relevance, we believe UltraLightUNet makes a meaningful contribution and is deserving contribution to ICLR 2025. Thank you for your feedback and consideration.

---

> ### Author Response · Authors · 2024-12-02
> **Official Comment by Authors: Part1**
>
> ### **C1. The novelty is low since the way to solve the problem has been widely explored and is the same, such as employing depth-wise convolution for lightweight design and splitting channels for isolated convolutions. Thus, putting much efforts on applying the exactly same way to solve the same problem is not very interesting, and this application from N to N+1 does not make impact on the medical image segmentation tasks. The architectural design cannot be considered as a novel design since the overall design (U-shaped encoder-decoder) is always used in the medical image segmentation. Incorporating several modules into this architecture does not revolutionize the architectural design.**
>
> **Response:**
>
> We thank the reviewer for their follow-up comments and the opportunity to clarify our contributions further. Below, we provide a detailed response to these new concerns.
>
>
> ### **Addressing Novelty in Architectural Design**
>
> Yes, the U-shaped encoder-decoder design is a well-established framework in medical image segmentation, originating with the seminal U-Net paper in 2015 (Ronneberger et al.). However, we believe that using the U-Net popularity as a penalizing argument for further creative improvements of this basic architecture is too reductionist in nature and actually disconnected from the reality in this area. In fact, as evidenced by a wide body of literature, this foundational U-design has seen continuous improvements over the years to address specific challenges in accuracy and computational efficiency. Notable examples include Attention UNet (Oktay et al., MIDL 2018), UNeXt (Valanarasu et al., MICCAI 2022), TransUNet (Chen et al., Medical Image Analysis 2024), EMCAD (Rahman et al., CVPR 2024), **3D UX-Net (Lee et al., ICLR 2023)**, and Swinunetr-v2 (He et al., MICCAI 2023), among others. These works improve over the basic U-Net architecture and highlight the significance of improving U-shaped architectures rather than aiming only for entirely revolutionary designs, which are rare. In fact, one of the most enduring contributions of the UNet architecture is precisely this wide-open area of research it enables which provides ample opportunities for further improvements to the SOTA.
>
> Our contribution aligns perfectly with this ongoing trend, but with a distinct focus on addressing **extremely low computational costs** while maintaining **high segmentation accuracy**. Unlike prior works that often increase architectural complexity to improve accuracy (e.g., TransUNet, 3D UX-Net, SwinUNETR-v2), UltraLightUNet provides a new type of solution focused on extreme efficiency, thus making it uniquely suited for real-time, resource-constrained scenarios such as point-of-care diagnostics. From this perspective, the UltraLightUNet architecture redefines the SOTA in image segmentation so this is why we believe our contribution is worthwhile and we should not be penalized for providing better results than SOTA.
>
> ### **Advancing Lightweight Design in Biomedical Imaging**
>
> Yes, depth-wise convolution and channel splitting are commonly used. However, it is essential to recognize that leveraging established concepts in novel ways to meet specific challenges and redefine the SOTA is a widely accepted and desirable approach in the field. For instance:
>
> - MobileNet (Howard et al., 2017) and EfficientNet (Tan et al., 2019) have successfully advanced computer vision using depth-wise convolutions (an idea already known at that time) for lightweight designs.
>
> - In biomedical imaging, UNeXt (Valanarasu et al., 2022), EGE-UNet (Ruan et al., 2023), and Rolling-UNet (Liu et al., 2024) have similarly employed lightweight modules for efficient segmentation, so simply improving SOTA with better ideas.
>
> UltraLightUNet contributes to this trend in research by introducing the new Equation 2 (reproduced in our first response of this rebuttal) that makes the mathematics of **Multi-Kernel Inverted Residual (MKIR)** and **Multi-Kernel Inverted Residual Attention (MKIRA)** blocks far more efficient by going beyond standard depth-wise convolutions by enabling adaptable feature extraction across diverse spatial contexts. Additionally, our integration of these modules into both 2D and 3D designs extends the applicability of lightweight models to volumetric medical imaging, a challenging and underexplored area. So, it is the new mathematical basis and the extended scope of our research that make our contribution worthwhile.

---

> ### Author Response · Authors · 2024-12-02
> **Official Comment by Authors: Part2**
>
> ### **Addressing "Application from N to N+1"**
>
> We respectfully disagree with the characterization of our work as an incremental application. While our design leverages depth-wise convolutions, it introduces novel mechanisms (based on a new mathematical basis in Equation 2) such as multi-kernel refinement for adaptable feature extraction and attention, as well as seamless 3D extensions of all modules. These innovations improve the state-of-the-art efficiency, which is critical for practical deployment in clinical settings which are scarce in resources.
>
> Moreover, our results demonstrate that UltraLightUNet outperforms (by orders of magnitude!) several heavier and lighter models in terms of computational cost and segmentation accuracy across diverse datasets, including both 2D and 3D tasks. This underscores the impact of our approach in advancing lightweight segmentation methods, particularly for real-time medical diagnostics.
>
> ### **Broader Context and Future Impact**
>
> Our work is part of a growing trend in computer vision in general, and biomedical imaging in particular, toward developing **ultralightweight models** that can deliver high precision in real-time tasks. Examples of this trend include:
>
> - **MobileNets** (Howard et al., 2017) and **EfficientNet** (Tan et al., 2019) in computer vision,
>
> - **TinyBERT** (Jiao et al., 2019) in language processing,
>
> - **UNeXt** (Valanarasu et al., 2022), **EGE-UNet** (Ruan et al., 2023), and **Rolling-UNet** (Liu et al., 2024) in biomedical imaging.
>
> All these lightweight architectures made headlines in the field by making possible vision/language/imaging tasks with significantly less resources. In other words, “less is more” is the new name of the game as this focus on efficiency is a paradigm shift in the making.  We believe that the significance of this type of work will only increase in future years as the demand for efficient, real-time models will continue to grow. From this perspective, UltraLightUNet’s contribution is particularly timely, by addressing the need for accurate, lightweight segmentation solutions in resource-constrained environments.
>
> We hope these clarifications address the reviewer’s concerns and highlight the importance and relevance of our contributions to the field. Thank you again for your thoughtful feedback and consideration.

---

> ### Comment · Reviewer_zJLj · 2024-12-02
>
> Thanks for authors' responses.
>
> (1) Like another reviewer jeKK said, the modules in this manuscript is same with modules in the paper EMCAD, so there is no module novelty. Like your title said, the model in this manuscript is still using U-Net architecture, so there is no architectural novelty as you mentioned.
> (2) The improvement of this manuscript is low, and contributions of this manuscript is low. The major contributions of this manuscript are mainly built over the paper EMCAD, resulting in a low impact to the field of medical image segmentation.
> (3) The work in this manuscript is trying to incorporate several previously proposed and widely used modules into a UNet architecture, so it doesn't provide theoretical insights to others.
>
> Overall, I will not change my score, "1 strong reject".

---

> > ### Author Response · Authors · 2024-12-02
> > **Official Comment by Authors: Final**
> >
> > Thank you for getting back to us. Sadly, it appears to us that you did not even look over our latest responses where we explicitly address, in great detail, each and every of your previous concerns. In short, to your points above:
> >
> > (1) The novelty of our paper comes mainly from two contributions, i.e., (i) new encoder design (which is not even considered in the EMCAD paper) and (ii) a new mathematical basis for MKIR module operation which leads to the overall extreme efficiency. U-Net is a fundamental architecture that did enable and will continue to enable innovation, just like our approach; there is nothing wrong in improving SOTA based on the U-Net architecture.
> >
> > (2) The impact of our approach in bioimaging is motivated by the orders of magnitude reduction in #Parameters and #FLOPS compared to SOTA (while maintaining the overall accuracy) we provide for both 2D and 3D (volumetric) segmentation. These huge improvements can enable real-time image segmentation in resource-limited scenarios (point-of-service scenarios) which have a high practical relevance; this is where the impact of our approach comes from.
> >
> > (3) The encoder and hence the end-to-end architecture are new and we explain the mathematical basis of our approach. The empirical nature of our paper is well aligned with many other impactful papers published in previous ICLRs, too any to list here; our paper beats SOTA for ultra-lightweight image segmentation and your refusal to acknowledge it does not diminish our contribution.
> >
> > Finally, it is disheartening to see this lack of genuine dialogue; our point-by-point responses addressed all your concerns, yet did not even trigger a careful reading of our arguments. We fail to see how rejecting this paper at all costs does serve this community...

---

> ### Comment · Reviewer_zJLj · 2024-12-02
> **Final Response**
>
> Thanks for your response. It seems that you misunderstood what I wanted to deliver. I read your responses carefully, but your responses didn't address my major concern.
>
> (1) The main contributions of this manuscript was proposing several modules, but these modules are similar to modules from the paper EMCAD, as another reviewer mentioned. Although you provided details and tried to clarify they were not 100% same, I believe they are at least 90% similar. (2) You said your architecture was novel since you incorporated your modules into the encoder. Unfortunately, incorporating modules into both the encoder and the decoder is not novel and is a marginal contribution since almost everyone is trying to incorporating their modules into both the encoder and the decoder. In other words, incorporating their modules into U-Net is not novel, and this is a common design. Many works have done it. (3) Using depthwise convolutions to reduce the Params and FLOPS has been explored from ConvNext and UX Net several years ago. These two are just examples, and these are many other networks are trying to use the depth-wise convolutions to improve efficiency after these two works. Thus, this idea is not new, and this idea in this manuscript doesn't provide new theoretical insight to others since this idea has been explored from several years.
>
> Most importantly, what I wanted to delivered was from the higher level, not about the details of your modules. You mentioned your designs in architectures and modules, and they may be novel to some other conferences. However, in ICLR, we would like to see some works that propose fundamentally different solutions and theoretically sound methods. Specifically, in the field of medical image segmentation, U-Net and Vision Transformers have been widely explored, so we would like to see some other newly-proposed architecture instead of U-Net and Vision Transformers, or some architectures that have not been applied to medical image segmentation. You utilized the depth-wise convolution to reduce Params and FLOPs, but it has been widely explored after ConNext and UXNet. The biggest selling point of your manuscript is the reduction of Params and FLOPs which benefits from the utilization of depth-wise convolutions, but this solution is not fundamentally different from others. They incorporated depth-wise convolutions into the decoder, and you incorporated it into both the encoder and decoder. These two designs are fundamentally same. We would like to see if you can propose new convolutional operations instead of depth-wise convolutions to reduce Params and FLOPs. In ICLR, I believe improving performance is not the first priority, since improving performance is not very hard, and performance will be influenced by the training strategies and computational resources. Instead, we would like to see solutions that tackle problems in a fundamentally different way, inspiring other readers' to explore this field deeply. Your work may achieve a promising result from the engineering side. However, it doesn't tackle this problem fundamentally, or provides more theoretical insights to others.

---

### Official Review · Reviewer_jeKK · 2024-11-04

**Soundness:** 1
**Presentation:** 1
**Contribution:** 1
**Rating:** 1
**Confidence:** 5

**Summary:**

The authors introduce UltraLightUNet, an ultra-lightweight 2D and 3D U-shaped network designed for medical image segmentation. This network features novel Multi-kernel Inverted Residual (MKIR) and Multi-kernel Inverted Residual Attention (MKIRA) blocks, aiming to effectively balance computational efficiency with high segmentation performance across multiple medical imaging benchmarks. The architecture is motivated by the need to reduce computational demands in point-of-care diagnostics, particularly in resource-constrained environments.

**Strengths:**

- The paper is well-written and easy to follow.

- The experimental results are through, with diverse datasets and settings.

- In-depth ablation study and parameters consideration are presented

**Weaknesses:**

- Despite the paper presents diverse experiments and ablation studies, I see a very close similarity to a CVPR 2024 paper, named EMCAD [1], which I detail in the next parts (Note that the EMCAD paper is not cited!)

- The proposed method (Figure 2) is clearly the same as last year CVPR paper, with approximately no change in any of the modules both in encoder and decoder of the network. Therefore, I see no novelty and contribution in the paper submitted.

- How is it possible for your method to just have 27000 parameters, while last year paper with the same architecture has at-least 3M parameters (its base version).

- The proposed method is not compared to other SOTA networks (such as [2]) which perform much better over some of the datasets used in the paper (such as ISIC)

[1] Rahman MM, Munir M, Marculescu R. Emcad: Efficient multi-scale convolutional attention decoding for medical image segmentation. In Proceedings of the IEEE/CVF Conference on Computer Vision and Pattern Recognition 2024 (pp. 11769-11779).


[2]Azad R, Niggemeier L, Hüttemann M, Kazerouni A, Aghdam EK, Velichko Y, Bagci U, Merhof D. Beyond self-attention: Deformable large kernel attention for medical image segmentation. In Proceedings of the IEEE/CVF Winter Conference on Applications of Computer Vision 2024 (pp. 1287-1297).

**Questions:**

Please refer to weaknesses section.

**Details Of Ethics Concerns:**

I have significant concerns regarding research integrity as the submitted manuscript appears to exhibit considerable overlap with another paper published in CVPR 2024 [1]. Specifically, the methodology described in this paper—including key architectural elements and experimental setup—seems to be identical to the aforementioned publication.

Given the extent of these similarities, I suspect plagiarism or a potential dual submission. I recommend that the conference organizers investigate this matter further to ensure that proper academic standards are upheld.

[1] Rahman MM, Munir M, Marculescu R. Emcad: Efficient multi-scale convolutional attention decoding for medical image segmentation. InProceedings of the IEEE/CVF Conference on Computer Vision and Pattern Recognition 2024 (pp. 11769-11779).

---

> ### Author Response · Authors · 2024-11-23
> **Response to the comments of Reviewer jeKK: Similarity to EMCAD (CVPR 2024) paper (Part1)**
>
> We are genuinely surprised by the ethical concerns raised regarding our UltraLightUNet paper. To clarify unequivocally and categorically, our ICLR submission is neither an act of plagiarism nor a duplicate submission of the EMCAD (Rahman et al., 2024). While both papers focus on improving efficiency in medical image segmentation, they have fundamentally different goals, methodologies, and contributions.
>
> Below, we address each and every concern and clarify the originality and contributions of our **UltraLightUNet** submission.
>
> ---
> ### **Q1. Despite the paper presents diverse experiments and ablation studies, I see a very close similarity to a CVPR 2024 paper, named EMCAD (Rahman et al., 2024), which I detail in the next parts**
>
> **Response:**
> We respectfully disagree with this assessment. While UltraLightUNet and EMCAD share a U-shaped architecture, which is widely used for medical image segmentation, they differ significantly in their **architectural philosophy, key innovations, and target use cases (as summarized in Table R1 below)**:
>
> **Table R1: Differences between UltraLightUNet and EMCAD**
>
> | **Feature**            | **UltraLightUNet**                                                                                                                                             | **EMCAD**                                                                                                           |
> |-------------------------|---------------------------------------------------------------------------------------------------------------------------------------------------------------|---------------------------------------------------------------------------------------------------------------------|
> | **Motivation**          | Designed as a **full (end-to-end) ultra-lightweight architecture** optimized to the extreme for resource-constrained environments.                                  | Focuses on optimizing the **decoder only**, relying on existing pre-trained encoders with emphasis on versatility.      |
> | **Encoder Design**         | **Built from scratch with** the new Multi-kernel Inverted Residual (**MKIR**) block which leverages **extremely efficient** multi-kernel depth-wise convolutions.             | Relies on **existing pre-trained encoders** (e.g., PVT_V2_B2, PVT_V2_B0) with **complicated** operations like self-attention.|
> | **Decoder Design**         | Uses the new Multi-kernel Inverted Residual Attention (**MKIRA**) block involving simple **bilinear upsampling** to reduce computational cost. **Note:** Multi-kernel here stands for $(k_1 = k_2) \text{ or } (k_1 \neq k_2), \text{where } k_1, k_2 \in \text{Kernels}$. | Employs Efficient Up-Convolution Block (**ECUB**) and Multi-scale Convolutional Attention Module (**MSCAM**). **Note:** Multi-scale here stands for $( k_1 \neq k_2 ), \text{where } k_1, k_2 \in \text{Kernels} $. |
> | **2D and/or 3D Versatility** | A unique architecture supporting **both 2D and 3D segmentation tasks**.                                                                                      | Focuses on **2D segmentation tasks only**.                                                                              |
> | **Target Use Cases**       | Focuses on both encoder and decoder efficiency, targeting **resource-constrained environments** (i.e., **0.027M parameters only** for tiny architecture).                                   | Focuses only on decoder efficiency; overall efficiency depends on the encoder (i.e., **3.92M parameters** for tiny architecture).            |
> | **Experimental Datasets**  | Evaluated on **both 2D** (polyp, skin lesion, breast tumor, cell, Synapse abdomen 8-organ, ACDC cardiac organ) **and 3D** (FeTA fetal brain, MSD Task05 Prostate, Synapse abdomen 13-organ and 8-organ) segmentation tasks. | Evaluated **only on 2D** (polyp, skin lesion, breast tumor, cell, Synapse abdomen 8-organ, ACDC cardiac organ) tasks.   |
>
> We can also illustrate these differences using an analogy from transportation and urban planning:
>
> **UltraLightUNet** is akin to a **compact electric scooter** built for quick, efficient navigation in crowded city streets. It’s lightweight, maneuverable, and optimized for short, real-time trips (analogy to real-time diagnostics) in constrained environments where heavy vehicles (complex models) can’t operate effectively. The focus is on minimalism and efficiency, designed to get the job done swiftly without excess fuel consumption (parameters and FLOPs in our case of resource-constrained scenarios).
>
> In contrast, **EMCAD** is like a **comprehensive highway system** designed to handle various types of traffic (cars, buses, trucks) efficiently, with multi-lane roads (multi-scale attention modules) and advanced traffic management (hierarchical feature refinement) to balance local (urban streets) and global (interstate highways) needs. It aims to provide a versatile, all-purpose solution that works well in different terrains and conditions.

---

> > ### Author Response · Authors · 2024-11-23
> > **Response to the comments of Reviewer jeKK: Similarity to EMCAD (CVPR 2024) paper (Part2)**
> >
> > ### **Q2. The proposed method (Figure 2) is clearly the same as last year CVPR paper, with approximately no change in any of the modules both in encoder and decoder of the network. Therefore, I see no novelty and contribution in the paper submitted.**
> >
> > **Below, we elaborate on all the differences in the Table R1 above:**
> >
> > 1. **Motivation**:
> >     - **UltraLightUNet** is motivated by the need for a **full (i.e., both encoder and decoder) architecture** that is optimized for **resource-constrained environments**. Consequently, our approach prioritizes the architecture’s computational **extreme efficiency** without sacrificing the segmentation accuracy for both **2D and 3D segmentation tasks**.
> >    - In contrast, **EMCAD** focuses solely on optimizing the **decoder** while targeting multi-scale features refinement for extreme versatility. As such, EMCAD relies on existing **pre-trained encoders** (e.g., PVT\_V2\_B2, PVT\_V2\_B0), which inherently increase the computational complexity depending on the complexity of the encoder. Finally, EMCAD targets primarily **2D segmentation tasks**.
> > 2. **Architectural Differences**:
> >
> >    To optimize both the encoder and decoder for achieving extreme efficiency, **UltraLightUNet** innovates through **Multi-Kernel Inverted Residual (MKIR)** and **Multi-Kernel Inverted Residual Attention (MKIRA)** blocks, which leverage the multi-kernel design, thus allowing for **kernels $k_1$ and $k_2$ be same $(k_1 = k_2)$ or different $(k_1 \neq k_2)$** based on application-specific needs. This design captures diverse spatial features with minimal computational cost.
> >
> >    In contrast, **EMCAD** focuses only on optimizing the decoder and uses the **Multi-Scale Residual Attention Module (MSCAM)**, where **kernels $k_1$ and $k_2$ must be different $(k_1 = k_2)$** to represent multiple scales. This conceptual distinction allows **UltraLightUNet** to adapt kernel sizes based on application-specific needs (e.g., large kernels for large regions, small kernels for small regions, or mixed for both), whereas EMCAD is limited to mixed kernels only.
> >
> >     - **Encoder Design**:
> >         - **UltraLightUNet** employs a **multi-kernel inverted residual** structure, focusing on an **ultra-lightweight convolutional approach** without relying on heavy attention mechanisms or transformers.
> >         - In contrast, **EMCAD** uses **existing and pre-trained Transformer encoders**, thus making its efficiency dependent on the encoder's complexity.
> >     - **Decoder Design**:
> >         - **UltraLightUNet** uses only the new **Multi-Kernel Inverted Residual Attention (MKIRA)** block, a local attention-based multi-kernel module we defined to selectively refine features. We note that **UltraLightUNet** reduces the computational complexity by using simple bilinear upsampling (i.e., avoiding any convolutional upsampling block).
> >         - In contrast, **EMCAD** uses the **Multi-scale Convolutional Attention Module (MSCAM)** and **Efficient Convolutional Upsampling Block (ECUB)**. The use of **ECUB** increases the computational complexity significantly.
> >
> > 3. **Target Use Cases**:
> >
> >    - **UltraLightUNet**: A unique architecture supports both **2D and 3D segmentation tasks** while prioritizing extreme efficiency, **thus making it ideal for real-time and low-resource environments (e.g., point-of-care diagnostics)** where computational resources are limited and resource efficiency is critical.
> >    - **EMCAD**: Primarily focuses on **2D tasks** and does not explicitly address 3D segmentation. This is suitable for applications where complex, hierarchical feature extraction is needed, by leveraging the power of vision transformers.
> >
> > We note that many highly cited methods in computer vision achieve novelty by introducing **modular innovations**, such as the Inverted Residual Block (IRB) in MobileNetv2 (Sandler et al., 2018), the SE block in Squeeze-and-Excitation Networks (Hu et al., 2018), and the Residual Block in ResNet (He et al., 2016). Other works combine these modules into new architectures, such as the use of SE blocks in **EfficientNet** (Tan et al., 2019) or Residual Block in almost every architecture.
> >
> > Similarly, **UltraLightUNet** uses standard attention mechanisms, such as channel attention and spatial attention, however, within a **new end-to-end U-shaped architecture** designed from scratch for extreme efficiency. This full encoder-decoder architecture targeting both **2D and 3D segmentation tasks** is one of the core contribution of our work.

---

> > > ### Author Response · Authors · 2024-11-23
> > > **Response to the comments of Reviewer jeKK: Parameter Efficiency**
> > >
> > > ### **Q3. How is it possible for your method to just have 27000 parameters, while last year paper with the same architecture has at-least 3M parameters (its base version).**
> > >
> > > The reviewer questions how **UltraLightUNet** achieves such a low parameter count (27,000) compared to EMCAD and other architectures.
> > >
> > > **Response:**
> > > **UltraLightUNet’s parameter efficiency** is achieved due to its new architecture, which is entirely designed with utmost efficiency in mind. More precisely:
> > >
> > > 1. **Encoder Efficiency**:
> > >    We have designed our encoder from scratch using the new **MKIR block**. The MKIR block uses **multi-kernel depth-wise convolutions**, which are inherently lightweight, yet very effective at capturing important features. By avoiding pre-trained encoders, **UltraLightUNet** reduces its parameter count significantly, hence our results in Tables 1, 2, 3, 11, 12, and Figs. 1, 3 in the main paper.
> > >
> > > 2. **Decoder Simplicity**:
> > >    Unlike EMCAD, which uses the heavier **ECUB block** in addition to MSCAB and LGAG, our **UltraLightUNet** architecture leverages only the **MKIRA** and **GAG blocks** for lightweight attention-based refinement.
> > >
> > > 3. **Unified End-to-End Lightweight Design Philosophy**:
> > >    Unlike EMCAD, which optimizes only the decoder, **UltraLightUNet** achieves **end-to-end efficiency** by optimizing both the encoder and decoder for parameter efficiency, thus resulting in **27,000 parameters (2D)** compared to EMCAD’s **3.92M parameters**.

---

> ### Author Response · Authors · 2024-11-23
> **Response to the comments of Reviewer jeKK: Comparisons to State-of-the-Art Models**
>
> ### **Q4: The proposed method is not compared to other SOTA networks (such as (Azad et al., 2024)) which perform much better over some of the datasets used in the paper (such as ISIC)**
>
> The reviewer mentions that **UltraLightUNet** lacks comparisons to certain SOTA models, such as **Deformable Large Kernel Attention**.
>
> **Response:**
> - **Existing Comparisons**:
>   We do already compare **UltraLightUNet** with multiple SOTA models, including **DeepLabv3+**, **TransUNet**, **SwinUNet**, and lightweight architectures like **UNeXt**, **CMUNeXt**, **EGE-UNet**, **Ultra_Light_VM_UNet**. Without exception, these comparisons demonstrate that **UltraLightUNet** achieves superior or competitive segmentation accuracy with significantly fewer parameters and lower computational costs.
>
> - **Newly Added Comparisons**:
>   Per reviewer suggestion, in the revised manuscript, we will include additional comparisons with recent SOTA methods, such as **Deformable Large Kernel Attention** (Azad et al., 2024). Below, we provide a detailed comparison of skin lesion segmentation on the ISIC 2018 and BUSI datasets.
>
> **Table R2**: Comparison of **UltraLightUNet** with Deformable Large Kernel Attention (DeformableLKA). We present the DICE scores (%) on our data-splits with an input resolution of 256 $\times$ 256, while optimizing the hyper-parameters of **DeformableLKA**.
>
> | **Architectures**                                | **Pretrained Encoder**                   | **#Params** | **#FLOPs** | **ISIC 2018** | **BUSI** |
> |---------------------------------------------------------------|---------------------------------------------------------------|-------------|-------------|-------------------------------|----------|
> | DeformableLKA                       | Yes                     | 102.76M     | 26.03G      | 90.34         | 79.01    |
> | DeformableLKA                       | No                      | 102.76M     | 26.03G      | 88.17         | 74.62    |
> | UltraLightUNet-T (Ours)             | No                      | 0.027M      | 0.026G      | 88.19         | 75.64    |
> | UltraLightUNet (Ours)               | No                      | 0.316M      | 0.314G      | 88.74         | 78.04    |
> | UltraLightUNet-M (Ours)             | No                      | 1.15M       | 0.951G      | 89.09         | 78.27    |
>
> Again, Table R2 shows that UltraLightUNet provides very competitive results using a fraction of #Params and #FLOPS compared to the DeformableLKA approach. Specifically, DeformableLKA, with a pretrained encoder, achieves the highest DICE score on ISIC 2018 (90.34%) and BUSI (79.01%) but requires 3,805x more #Params and 1,001x more #FLOPs than UltraLightUNet-T (0.027M parameters and 0.026G FLOPs). Without pretraining, DeformableLKA’s DICE scores drop by 2.17% on ISIC 2018 and 4.39% on BUSI, thus falling below the DICE scores of our UltraLightUNet-T (88.19% and 75.64%).
>
> In contrast, UltraLightUNet-M which does *not* rely on pretraining, delivers very competitive DICE scores: 89.09% (ISIC 2018) and 78.27% (BUSI), thus narrowing the gap to just 1.25% on ISIC 2018 and 0.74% on BUSI compared to pretrained DeformableLKA.

---

> > ### Author Response · Authors · 2024-11-23
> > **Response to the comments of Reviewer jeKK: Summary of Changes**
> >
> > ### **Summary of Changes**
> >
> > To address the reviewer’s concerns and strengthen the manuscript, we will:
> >
> > 1. **Explicitly Cite EMCAD** and include a detailed comparison table (as provided above) highlighting differences in motivation, architecture, and experimental scope.
> >
> > 2. **Expand Methodology Section** to provide additional details on the **MKIR** and **MKIRA** blocks, emphasizing their novelty and roles in achieving great performance with this ultra-light architecture.
> >
> > 3. **Enhance Experimental Scope** to include additional comparisons with recent SOTA methods, such as **Deformable Large Kernel Attention**, and validate the model on challenging datasets like **BRATS**.
> >
> > ---
> > ### **Conclusion**
> >
> > While we strongly disagree with the claims of similarity to EMCAD and the ethical concerns raised on our paper, the revised version of our paper will demonstrate the **originality, significance, and impact of UltraLightUNet** as a novel contribution to lightweight medical image segmentation.
> >
> > Thank you for your time and constructive feedback. We look forward to submitting the revised manuscript.
> >
> >
> > ### **References**
> >
> > Rahman, M.M., Munir, M. and Marculescu, R., 2024. Emcad: Efficient multi-scale convolutional attention decoding for medical image segmentation. In *Proceedings of the IEEE/CVF Conference on Computer Vision and Pattern Recognition* (pp. 11769-11779).
> >
> > Sandler, M., Howard, A., Zhu, M., Zhmoginov, A. and Chen, L.C., 2018. Mobilenetv2: Inverted residuals and linear bottlenecks. In *Proceedings of the IEEE Conference on Computer Vision and Pattern Recognition* (pp. 4510-4520).
> >
> > Hu, J., Shen, L. and Sun, G., 2018. Squeeze-and-excitation networks. In *Proceedings of the IEEE Conference on Computer Vision and Pattern Recognition* (pp. 7132-7141).
> >
> > He, K., Zhang, X., Ren, S. and Sun, J., 2016. Deep residual learning for image recognition. In Proceedings of the IEEE conference on computer vision and pattern recognition (pp. 770-778).
> >
> > Tan, M. and Le, Q., 2019, May. Efficientnet: Rethinking model scaling for convolutional neural networks. In International conference on machine learning (pp. 6105-6114).
> >
> > Azad, R., Niggemeier, L., Hüttemann, M., Kazerouni, A., Aghdam, E.K., Velichko, Y., Bagci, U. and Merhof, D., 2024. Beyond self-attention: Deformable large kernel attention for medical image segmentation. In Proceedings of the IEEE/CVF Winter Conference on Applications of Computer Vision (pp. 1287-1297).

---

> > > ### Comment · Reviewer_jeKK · 2024-11-25
> > > **Response to authors about addressing my concerns**
> > >
> > > I appreciate the authors for addressing my concerns and including additional experiments. Below, I outline my further observations.
> > >
> > > I acknowledge that the EMCAD work focuses on optimizing only the decoder, and I agree with this point. However, I note that the modules you have utilized in your decoder—and even your primary module in the encoder (MKIR)—are essentially identical to those in the EMCAD module. Let’s analyze each module in detail.
> > >
> > > You claim to use the "new" Multi-kernel Inverted Residual (MKIR) block, yet the design of the MKIR block appears to be identical to the MSCB block in the EMCAD paper. Furthermore, your MKIRA module (CMFA + MKIR) is functionally the same as the MSCAM module from the EMCAD paper (CAB + SAB + MSCB). Your GAG module is the same as LGAG in the EMCAD paper. Your CMFA module is just the same as (CAB+SAB) in the EMCAD paper.
> > >
> > > While I agree that you have incorporated 3D experiments and presented more extensive experimental results, from an architectural standpoint, I do not see any new or “novel” modules as claimed. While I recognize that novelty is not solely about methodology (a point often emphasized by reviewers), I personally believe that innovation is not limited to architectural design alone.
> > >
> > > Despite this, every detail in your paper closely mirrors EMCAD, which unfortunately does not persuade me to revise my score upward.

---

> ### Author Response · Authors · 2024-12-02
> **Official Comment by Authors: Part1**
>
> **Response:**
>
> We thank the reviewer for their follow-up comments and the opportunity to clarify our contributions further. Below, we address this reviewer’s concerns directly and highlight the unique aspects of our work.
>
> ## 1. Novelty of UltraLightUNet Modules
>
> ### **Multi-Kernel Inverted Residual (MKIR) Block**
> The MKIR block introduces a novel feature extraction mechanism based on our **multi-kernel trick** (i.e., *MKDC*) in Equation 2 (reproduced below):
>
> $MKDC(x) = CS\left(\sum_{k \in K} DWCB_k(x)\right)$
>
> Where $DWCB_k(x) = ReLU6(BN(DWC_k(x)))$. Here, $DWC_k(x)$ is a depth-wise convolution with kernel $k$. Of note, our MKDC supports *both* $k_1=k_2$ (same-size kernels, e.g., $[3 \times 3,3 \times 3,3 \times 3]$, $[5 \times 5,5 \times 5,5 \times 5]$) *and* $k_1 \neq k_2$ (different-size kernels, e.g., $[1 \times 1,3 \times 3,5 \times 5]$), for $k_1, k_2 \in K$. This flexibility in supporting both identical and different scale kernels distinguishes the MKDC from the EMCAD’s MSDC block (see Equation 5 in the EMCAD paper):
>
> $MSDC(x) = \sum_{k \in K} DWCB_k(x)$
>
> which is restricted only to **multi-scale designs** ($k_1 \neq k_2$, e.g., $[1 \times 1,3 \times 3,5 \times 5]$). This shows the new contribution MKIR brings compared to MSDC, mathematically speaking; this difference is the very basis for the excellent results of the UltraLightUNet architecture.
>
> Indeed, by enabling adaptable kernel configurations, MKIR offers efficient and versatile feature extraction tailored to application-specific needs, thus reducing computational cost drastically while achieving high segmentation accuracy. An illustrative example, various statistics, and empirical evidence supporting our claims are described next.
>
> **Explaining Multi-Kernel vs. Multi-Scale with an Illustrative Example:**
>
> Let us start with an analogy: Imagine we are drawing on a piece of paper with different types of paintbrushes: **small brushes**, **big brushes**, and a set of **both small and big brushes**. Now, let's say we want to color objects of different sizes, like:
>
> - **Small objects**, such as tiny dots.
> - **Large objects**, like big circles.
> - **Mixed objects**, where we have both tiny dots and big circles.
>
> **The Multi-Kernel Scenario (akin to this ICLR paper):** In the multi-kernel scenario, we can choose our brushes based on what we need:
>
> - **Only small brushes** for tiny dots.
> - **Only big brushes** for big circles.
> - **A mix of small and big brushes** for both tiny dots and big circles.
>
> This flexibility lets us adapt our tools (kernels) for the specific task at hand. In other words, we might say, "we’ll use just small brushes for this drawing because it’s all tiny dots," or "Let’s use a mix because we need to cover both."
>
> **The Multi-Scale Scenario (akin to the EMCAD paper):** In the multi-scale scenario, **we must always use a mix of small and big brushes together**, no matter what. Whether we have tiny dots, big circles, or both, we must use the mix of paintbrushes. This may work well for some cases, but it’s clearly less flexible because we can’t decide to use just one type of brush when we have only tiny or big objects.
>
> **How this Analogy Applies to Image Segmentation:** In medical imaging:
>
> - **Small objects**: Tiny tumors or small lesions.
> - **Large objects**: Big organs like the liver or spleen.
> - **Mixed objects**: Both small lesions and large organs in the same image.

---

> > ### Author Response · Authors · 2024-12-02
> > **Official Comment by Authors: Part2**
> >
> > **Table R6:** Comparing the effect of different types of multi-kernel strategies such as $k_1=k_2$ (only small $[3 \times 3, 3 \times 3, 3 \times 3]$, only large $[5 \times 5, 5 \times 5, 5 \times 5]$) and $k_1 \neq k_2$ (multi-scale $[1 \times 1, 3 \times 3, 5 \times 5]$) in objects of different sizes (small, large, mixed). We use $3 \times 3$ and $5 \times 5$ average kernels for small and large kernels, respectively. Our network optimize the weight of these kernels. Bold entries show the best results. SSIM stands for structural similarity index measure and MSE stands for mean squared error. **Note:** As OpenReview does not allow us adding figures in the official comment boxes, we could not also include the visual plots of convolved outputs herewith.
> >
> >
> > | Objects in Images | Kernels Used          | Object-to-Background Ratio $\uparrow$ | SSIM $\uparrow$ | MSE  $\downarrow$ |
> > |--------------------|-----------------------|-------------------------------|------|------|
> > | Small             | Small [3×3, 3×3, 3×3] | **41.83**                     | 0.79 | 0.007 |
> > | Small             | Large [5×5, 5×5, 5×5] | 11.69                         | 0.526 | 0.024 |
> > | Small             | Multi-scale [1×1, 3×3, 5×5] | 37.66                    | 0.786 | 0.011 |
> > | Large             | Small [3×3, 3×3, 3×3] | 9.35                          | 0.513 | 0.039 |
> > | Large             | Large [5×5, 5×5, 5×5] | **24.90**                     | **0.732** | **0.012** |
> > | Large             | Multi-scale [1×1, 3×3, 5×5] | 23.70                    | 0.69  | 0.018 |
> > | Multi-scale       | Small [3×3, 3×3, 3×3] | 13.64                         | 0.761 | 0.029 |
> > | Multi-scale       | Large [5×5, 5×5, 5×5] | 5.16                          | 0.518 | 0.063 |
> > | Multi-scale       | Multi-scale [1×1, 3×3, 5×5] | **14.58**               | **0.81** | **0.019** |
> >
> >
> > ---
> >
> > All statistics in Table R6 strongly support the benefits of **UltraLightUNet's MKIR** block over the **EMCAD's MSCB** block, thus highlighting MKIR's broader adaptability and effectiveness. While MSCB operates solely on multi-scale kernels (e.g., $[1 \times 1, 3 \times 3, 5 \times 5]$), making it suitable only for multi-scale object segmentation, **MKIR supports both same-size kernels** (e.g., $[3 \times 3, 3 \times 3, 3 \times 3]$ or $[5 \times 5, 5 \times 5, 5 \times 5]$) and **multi-scale kernels** (e.g., $[1 \times 1, 3 \times 3, 5 \times 5]$), thus enabling application-specific optimizations which make the UltraLightUNet architecture extremely efficient.
> >
> > **Key Evidence Supporting MKIR’s Novelty in Table R6:**
> >
> > **1. Object-to-Background Ratio:**
> >
> > - For **small objects**, MKIR with small kernels achieves the highest ratio (41.83), emphasizing the precise focus on small details, unlike MSCB.
> >
> > - For **large objects**, MKIR with large kernels outperforms multi-scale (24.90 vs. 23.70), validating its suitability for larger regions.
> >
> > - For **mixed objects**, MKIR with multi-scale kernels balances small and large regions effectively (14.58), demonstrating its versatility across complex applications.
> >
> > **2. Mean Squared Error (MSE):**
> >
> > MKIR consistently achieves **lower MSE** for relevant kernel-object pairs, thus indicating better pixel-wise accuracy.
> >
> > - For example, small kernels achieve 0.007 MSE for small objects, outperforming multi-scale (0.011) and large kernels (0.024).
> >
> > - Similarly, large kernels yield the best MSE (0.012) for large objects, reinforcing MKIR’s adaptability.
> >
> > **3. Structural Similarity Index Measure (SSIM):**
> >
> > MKIR excels in structural preservation across relevant kernel-object pairs, demonstrating adaptability and robustness:
> >
> > - **Small objects:** Small kernels achieve the highest SSIM (0.79), thus outperforming multi-scale (0.786) and large kernels (0.526).
> >
> > - **Large objects:** Large kernels yield the best SSIM (0.732), surpassing multi-scale (0.69).
> >
> > - **Mixed objects:** Multi-scale kernels achieve the highest SSIM (0.81), balancing small and large features better than small (0.761) or large kernels (0.518).
> >
> > **4. Application-Specific Adaptability:**
> >
> > - MSCB in EMCAD paper is inherently restricted to multi-scale designs, thus making it **a special case** of MKIR when kernels can be different. In contrast, MKIR can adapt same-size or mixed kernels for specific applications, achieving optimal performance across small, large, and heterogeneous segmentation tasks.
> >
> > - The flexibility of MKIR allows segmentation models to tailor kernel configurations based on the nature of the objects, significantly broadening its applicability beyond MSCB.
> >
> > These findings establish MKIR as a fundamentally **new and superior** block compared to MSCB in EMCAD paper. MKIR’s broader adaptability and application-specific adaptability underscore its significant contribution to advancing lightweight, high-performance segmentation models.

---

> > > ### Author Response · Authors · 2024-12-02
> > > **Official Comment by Authors: Part3**
> > >
> > > Table R7 provides a comparative analysis of various **multi-kernel strategies** to show the effectiveness of **same-size kernels** ($k_1 = k_2$, e.g., $[3 \times 3, 3 \times 3, 3 \times 3]$, $[5 \times 5, 5 \times 5, 5 \times 5]$) and **multi-scale kernels** ($k_1 \neq k_2$, e.g., $[1 \times 1, 3 \times 3, 5 \times 5]$) on segmenting objects of varying sizes across three datasets: **MSD Task09_Spleen** (large object), **MSD Task06_Lung** (small object), and **BUSI** (mixed objects).
> > >
> > >
> > > **Table R7:** Comparing the effect of different multi-kernel tricks (same $k_1=k_2$) and multi-scale ($k_1 \neq k_2$) convolutions on small, large, and mixed objects segmentation on MSD Task09_Spleen (large object), MSD Task06_Lung (small object), and BUSI (mixed objects) datasets. DICE scores (%) are reported with our UltraLightUNet3D for MSD Task09_Spleen (large object), MSD Task06_Lung (small object) datasets, while with our UltraLightUNet for BUSI dataset. We report the \#Params and \#FLOPs of our UltraLightUNet3D architecture with an input resolution of $96 \times 96 \times 96$.
> > >
> > > | **Multi-kernel tricks**     | **\#Params (M)** $\downarrow$ | **\#FLOPs (G)** $\downarrow$ | **Spleen (large object)** $\uparrow$ | **Lung cancer (small object)** $\uparrow$ | **BUSI (mixed objects)** $\uparrow$ |
> > > |-----------------------------|------------------|-----------------|---------------------------|---------------------------------|--------------------------|
> > > | $1 \times 1, 1 \times 1, 1 \times 1$ | 0.279            | 1.01           | 93.65                     | 60.25                           | 72.13                   |
> > > | $3 \times 3, 3 \times 3$ | 0.338            | 1.58           | 95.86                     | 70.26                           | 76.83                   |
> > > | $3 \times 3, 3 \times 3, 3 \times 3$ | 0.369            | 1.88           | 96.03                     | **71.09**                       | 76.86                   |
> > > | $1 \times 1, 3 \times 3, 5 \times 5$ | 0.453            | 2.68           | 95.99                     | $\underline{70.32}$                           | **78.04**               |
> > > | $5 \times 5, 5 \times 5$ | 0.564            | 3.76           | $\underline{96.20}$                     | 69.98                           | $\underline{77.88}$                   |
> > > | $5 \times 5, 5 \times 5, 5 \times 5$ | 0.709            | 5.16           | **96.29**                 | 70.24                           | 77.80                   |
> > > ---
> > >
> > >
> > > **Performance on Large Objects (Spleen):**
> > > - Kernels with larger sizes (e.g., $5 \times 5,5 \times 5,5 \times 5$) achieve the highest DICE score (**96.29**), as expected for large Spleen segmentation; this supports our claims above.
> > >
> > > - Multi-scale kernels (**$1 \times 1,3 \times 3,5 \times 5$**) perform slightly lower (**95.99**), but still better than smaller kernels, showing that adaptability is critical for large regions.
> > >
> > > **Performance on Small Objects (Lung Cancer):**
> > >
> > > - Small kernels (**$3 \times 3,3 \times 3,3 \times 3$**) achieve the best DICE score (**71.09**) with lower #Params and #FLOPs, thus confirming their suitability for fine detail segmentation.
> > >
> > > - Multi-scale kernels (**$1 \times 1,3 \times 3,5 \times 5$**) also perform well (**70.32**) but are slightly less effective compared to same-size small kernels for small object segmentation.
> > >
> > > **Performance on Mixed Objects (BUSI):**
> > >
> > > - Multi-scale kernels (1x1,3x3,5x5) achieve the highest DICE score (**78.04**), highlighting their ability to balance segmentation for both small and large objects.
> > >
> > > - Large kernels alone underperform (**77.80**) in mixed-object scenarios, thus showing limitations in capturing small object details.
> > >
> > > To sum up, Table R7 demonstrates the *adaptability* of **UltraLightUNet’s multi-kernel strategy** in addressing diverse object segmentation challenges. These results highlight that same-size kernels excel in specific scenarios (small or large objects), while **multi-scale kernels** are essential for balancing segmentation in mixed-object datasets. This adaptability distinguishes UltraLightUNet's **MKIR block** from the EMCAD's fixed **multi-scale MSCB**, thus underscoring its broader application and effectiveness in medical imaging tasks.

---

> > > > ### Author Response · Authors · 2024-12-02
> > > > **Official Comment by Authors: Part4**
> > > >
> > > > ### **Multi-Kernel Inverted Residual Attention (MKIRA) Block**
> > > >
> > > > It is not about the architectural novelty of the Convolutional Multi-Focal Attention (CMFA) module per se (so we explicitly cited the Channel and Spatial Attention mechanisms in our initial submission), but rather **the novelty comes from the integration of CMFA with MKIR** to achieve **multi-kernel refinement of attention**. This combination enhances the feature refinement across spatial contexts, thus leveraging the adaptability of the multi-kernel design which is absent in EMCAD’s MSCAM.
> > > >
> > > > Additionally, the **3D extension of CMFA** is novel in the context of medical imaging. This adaptation enables efficient attention mechanisms for volumetric data, which is a critical advancement for 3D medical image segmentation tasks.
> > > >
> > > > ### **Grouped Attention Gate (GAG)**
> > > >
> > > > We acknowledge that the GAG follows the LGAG module from EMCAD, and we have followed this Review’s suggestion and cite EMCAD in our revised submission. However, in this ICLR paper, GAG is used in conjunction with our novel MKIR and MKIRA modules, which distinguishes the overall design of our decoder. Furthermore, GAG is seamlessly extended to 3D, which is a new contribution specific to volumetric medical imaging.
> > > >
> > > > ---
> > > >
> > > > ## 2. 3D Module Extensions
> > > >
> > > > One of our major contributions is the **3D extension of all modules** (MKIR, MKIRA, CMFA, and GAG). This extension introduces novel volumetric feature extraction and refinement capabilities that are absent in EMCAD, which focuses solely on 2D tasks. To the best of our knowledge, we are the **first to use multi-kernel trick** and **local attention mechanisms in a 3D architecture** for medical image segmentation. Consequently, our UltraLightUNet3D becomes the high-performing SOTA ultralightweight architecture for volumetric image segmentation.
> > > >
> > > > Our 3D modules enable UltraLightUNet to effectively handle complex volumetric medical imaging tasks while maintaining a high computational efficiency. We demonstrate this through experiments on challenging datasets, including **MSD Task01_BrainTumour** and **Task06_Lung Cancer** (Table 13 in revised submission), thus highlighting the efficacy of our 3D design.
> > > >
> > > > ---
> > > >
> > > > ## 3. Significance Compared to 3D UX-Net (ICLR 2023)
> > > >
> > > > Let’s compare our contribution with a recently accepted paper at ICLR. While **3D UX-Net** (ICLR 2023) extends the ConvNeXt block to 3D, it retains a computationally expensive decoder identical to SwinUNETR, resulting in high resource demands. In contrast:
> > > >
> > > > - UltraLightUNet introduces a conceptually new module for the MKIR module (please see our first response and example) and extends all modules to work with 3D segmentation, providing a lightweight and efficient design for both 2D and 3D tasks.
> > > >
> > > > - Our work addresses the computational challenges in resource-constrained scenarios, achieving state-of-the-art efficiency without sacrificing segmentation precision. Of note, we are talking about orders of magnitude improvements in efficiency which makes the UltraLightUNet the new SOTA in this problem space.
> > > >
> > > > - Given that 3D UX-Net was accepted at ICLR 2023 for its contributions, we believe that the UltraLightUNet with its broader applicability, represents a more substantial contribution to the ICLR community and will inspire future advancements in lightweight medical image segmentation.
> > > >
> > > > We hope these clarifications address the reviewer’s concerns and emphasize the novelty, practicality, and impact of our contributions. Thank you again for your thoughtful feedback and consideration.

---

### Author Response · Authors · 2024-11-26
**Official Global response by Authors: Part1**

We sincerely thank all reviewers for their constructive feedback, which has been invaluable in improving our manuscript. We are encouraged by the recognition of **UltraLightUNet**’s contributions to lightweight, high-performance segmentation and its potential for practical applications in resource-constrained scenarios.

We note that our contribution is along the lines of a growing direction in computer vision in general, and biomedical imaging in particular, of using ultralightweight models to perform real-time tasks with high precision. Remarkable examples of this trend are MobileNet and EfficientNet in computer vision, TinyBERT in language processing, UNeXt and EGE-UNet in biomedical imaging. We beleive that the significance of this type of work will only increase in future years.

Below, we provide a summary of the major revisions made in response to the reviewers’ comments and highlight our specific changes. We marked these changes in **blue** in the revised version of our draft.

## Key Modifications and Contributions

### 1. New Experiments on Complex Datasets

To address the reviewers’ concerns regarding the evaluation of **UltraLightUNet** on more complex segmentation tasks (**Table R5**), we conducted additional experiments on two challenging datasets:

- **MSD Task01_BrainTumour** (multi-level tumor segmentation):
  - Results show that **UltraLightUNet3D-M** achieves the **best DICE scores** on Tumor Core (TC) and Whole Tumor (WT) segmentation, while having remarkably lower #Params and #FLOPs compared to heavyweight models (**SwinUNETR**, **3D UX-Net**).
  - The base model, **UltraLightUNet3D**, also performs competitively, outperforming lightweight models like **SlimUNETR** with the lowest computational cost.

- **MSD Task06_Lung** (binary CT lesion segmentation):
  - Results demonstrate **UltraLightUNet**’s ability to handle complex binary cancer segmentation tasks efficiently, with **UltraLightUNet3D-M** achieving the **best DICE score** compared to other baselines, including lightweight (e.g., **SlimUNETR**) and heavyweight (**3D UX-Net**, **SwinUNETR**).

### 2. Theoretical Motivation, Methodological Novelty, and Interpretability

- We elaborated on the theoretical underpinnings of **UltraLightUNet**’s modules:
  - Our **multi-kernel trick** supports both **$k_1 = k_2$** (same-size kernels) and **$k_1 ≠ k_2$** (different-size kernels) for **$k_1$, $k_2$** $\in Kernels$ versus conventional multi-scale (only **$k_1 ≠ k_2$**) designs, thus allowing adaptable context extraction.
  - This conceptual distinction allows **UltraLightUNet** to adapt kernel sizes based on application-specific needs (e.g., large kernels for large objects, small kernels for small objects, or mixed for both objects segmentation).
  - **Depth-wise convolutions** across multiple kernels (**$k_1 = k_2$** or **$k_1 ≠ k_2$**) ensure lightweight computation without compromising accuracy.
- We clarified about our **methodological novelty**:
    - End-to-end ultralightweight architecture design
    - Novel 2D modules,
    - New 3D modules, and
    - Both 2D and 3D versatility.
- **Activation heatmaps (Figure R1)** were added to demonstrate how MKIR and CMFA focus on critical regions in an image, thus providing interpretability evidence for the modules.

### 3. Clarifications on Related Approaches

We clarified the distinctions between **UltraLightUNet** and related methods:

- **EMCAD**: Table R1 explicitly highlights the differences, including **UltraLightUNet**’s conceptual distinction (**$k_1 = k_2$** or **$k_1 ≠ k_2$**), 3D versatility, and unified end-to-end lightweight design.
- **SAMT, CASCADE, ConvNeXt, and SFCNs**: We emphasized the advantages of our multi-kernel approach over multi-scale or single-scale strategies used in these methods.

### 4. Additional Comparisons and New Baseline Results

- **2D Segmentation**: We implemented **DeformableLKA** (*Azad et al., 2024*), a recent segmentation method, and compared its performance with **UltraLightUNet** in **Table R2**. Results confirm **UltraLightUNet**’s superior efficiency and segmentation accuracy.
- **3D Segmentation**: **SlimUNETR** (*Pang et al., 2023*), a lightweight 3D method, was implemented and compared in **Table R4**. **UltraLightUNet** outperforms **SlimUNETR** in both segmentation accuracy and computational efficiency.

### 5. Efficiency Metrics

- Reported training and inference times, parameter counts, FLOPs, and average Dice scores across six binary segmentation datasets (**Table R3**). **UltraLightUNet** demonstrated a compelling trade-off between accuracy and computational cost.

- **Table R3** will be reported in **Table 14 and Appendix A.13** in our revised manuscript.

---

> ### Author Response · Authors · 2024-11-26
> **Official Global response by Authors: Part2**
>
> ### 6. Comprehensive Ablation Studies
>
> - We emphasized the contributions of individual modules (e.g., MKIR, MKIRA, CMFA, GAG) using detailed ablation studies (**Tables 4 and 5** in the initial submission). Results show that:
>     - **MKIR and MKIRA** are the most critical for accuracy improvements, especially on datasets like BUSI (from 72.41% to 76.61%).
>     - Combining all modules yields the best overall performance (78.04% on BUSI).
> - These studies validate the design of **UltraLightUNet** and will be further clarified in the revised manuscript.
>
> ### 7. Clarity on Model Variants
>
> - We clarified the differences among **UltraLightUNet-T, S, M, and L**:
>   - Variations in channel dimensions across encoder-decoder stages (e.g., T: (4, 8, 16, 24, 32); M: (32, 64, 128, 192, 320)).
>   - Scalability analysis (**Tables 9 and 10**) validates the performance improvements with increasing channel counts.
>
> ---
>
> ### 8. Limitations and Future Directions
>
> - We added a dedicated subsection in **Appendix A.14** to discuss the limitations of **UltraLightUNet**:
>   - Slightly lower performance on highly complex datasets compared to heavyweight SOTA models.
>   - Trade-offs between computational efficiency and segmentation accuracy.
> - We also outlined future directions, including hybrid architectures, self-supervised pretraining, and extensions to tasks like image reconstruction, enhancement, and denoising.
>
> ---
>
> ## Key Revisions
>
> Based on the rebuttal, we incorporate the following changes:
>
> 1. **Expanded Introduction and Related Work**:
>    - Discuss recent lightweight (**SlimUNETR**) and relevant (**DeformableLKA**) methods.
>    - Clearly differentiate **UltraLightUNet** from EMCAD, ConvNeXt, SAMT, CASCADE, and SFCNs.
>    - Highlight the novelty of **UltraLightUNet’s** unified multi-kernel approach.
>
> 2. **Revised Methodology**:
>    - Emphasize the theoretical basis for multi-kernel (**$k_1 = k_2$** and **$k_1 ≠ k_2$**) vs. multi-scale (**$k_1 ≠ k_2$**) approaches and its adaptability across diverse tasks.
>    - Modify **Figure 2** to reflect the conceptual distinctions.
>    - Clarify the differences among **UltraLightUNet-T, S, M, and L**.
>
> 3. **Updated and Added Tables and Figures**:
>    - Add results for **DeformableLKA** (**Table R2**) and **SlimUNETR** (**Table R4**) to **Tables 1, 3, 12, 13, and 14** in our revised manuscript.
>    - Include results on new datasets (**Table R5**) to **Table 13** in our revised manuscript.
>    - Add activation heatmaps (**Figure R1**) to enhance interpretability discussions (**Figure 3** in our revised manuscript).
>    - Add training/inference metrics (**Table R3**) to **Table 14** in **Appendix A.13**.
>    - Updated the DICE score of SwinUNet in Table 2 on Synapse dataset to reflect the original authors DICE score as per the reviewer suggestion.
>
> 4. **Added Limitations and Future Directions**:
>    - Include a discussion on limitations and future directions in **Appendix A.14**.
>
> ---
>
> We believe these revisions comprehensively address the reviewers' comments, strengthen the manuscript, and clarify **UltraLightUNet’s** significant contributions to lightweight medical image segmentation. Thank you for your valuable feedback, which has greatly enhanced the rigor and clarity of our work. We look forward to submitting the revised version.
>
> ## References
>
> Rahman et al., 2024. Emcad: Efficient multi-scale convolutional attention decoding for medical image segmentation. CVPR (pp. 11769-11779).
>
> Sandler wt al., 2018. Mobilenetv2: Inverted residuals and linear bottlenecks. CVPR (pp. 4510-4520).
>
> Hu et al., 2018. Squeeze-and-excitation networks. CVPR (pp. 7132-7141).
>
> He et al., 2016. Deep residual learning for image recognition. CVPR (pp. 770-778).
>
> Tan et al., 2019, May. Efficientnet: Rethinking model scaling for convolutional neural networks. ICML (pp. 6105-6114).
>
> Azad et al., 2024. Beyond self-attention: Deformable large kernel attention for medical image segmentation. WACV (pp. 1287-1297).
>
> Lin et al., 2023. Scale-aware modulation meet transformer. ICCV (pp. 6015-6026).
>
> Woo et al., 2018. Cbam: Convolutional block attention module. ECCV (pp. 3-19).
>
> Rahman et al., 2023. Medical image segmentation via cascaded attention decoding. WACV (pp. 6222-6231).
>
> Cao et al., 2022. Swin-unet: Unet-like pure transformer for medical image segmentation. ECCV (pp. 205-218).
>
> Pang et al., 2023. Slim UNETR: Scale hybrid transformers to efficient 3D medical image segmentation under limited computational resources. IEEE Transactions on Medical Imaging
>
> Tang et al., 2024. Cmunext: An efficient medical image segmentation network based on large kernel and skip fusion. ISBI (pp. 1-5).
>
> Yang et al., 2023. UcUNet: A lightweight and precise medical image segmentation network based on efficient large kernel U-shaped convolutional module design. Knowledge-Based Systems, 278, 110868.

---

> > ### Author Response · Authors · 2024-12-02
> > **New response to all Reviewers comments**
> >
> > We are very grateful to all reviewers for engaging with us in a meaningful conversation based on the initial reviews. Whoever, we believe that there is a lot of potential to clarify further points and, most importantly, make sure both parties (authors and reviewers) get the chance to make things right and correct any remaining misunderstandings.
> >
> > In our responses below, we further clarify our contribution, provide a new and more intuitive angles to highlight the novelty of our approach, and hopefully provide reasons for these reviewers to revise their initial scores. Given our justification, the huge amount of new results we provide to support our claims, there is no fair way for these lowest scores to remain unchanged, particularly since our results redefine the SOTA for real-time 2D/3D image segmentation with limited resources.
> >
> > Our individual responses below address the very core of all the issues raised in previous iteration. We remain committed to provide any further clarifications these Reviewers may find necessary to better asses the contribution of our paper.
> >
> > Thank you.

---

> > > ### Author Response · Authors · 2024-12-02
> > >
> > > **New References in our Rebuttal**
> > >
> > > Liu, Z., Mao, H., Wu, C.Y., Feichtenhofer, C., Darrell, T. and Xie, S., 2022. A convnet for the 2020s. In Proceedings of the IEEE/CVF conference on computer vision and pattern recognition (pp. 11976-11986).
> > >
> > > Lee, H.H., Bao, S., Huo, Y. and Landman, B.A., 3D UX-Net: A Large Kernel Volumetric ConvNet Modernizing Hierarchical Transformer for Medical Image Segmentation. In The Eleventh International Conference on Learning Representations.
> > >
> > > Rahman, M.M., Munir, M. and Marculescu, R., 2024. Emcad: Efficient multi-scale convolutional attention decoding for medical image segmentation. In Proceedings of the IEEE/CVF Conference on Computer Vision and Pattern Recognition (pp. 11769-11779).
> > >
> > > Mehta, S. and Rastegari, M., MobileViT: Light-weight, General-purpose, and Mobile-friendly Vision Transformer. In International Conference on Learning Representations.
> > >
> > > Hatamizadeh, A., Nath, V., Tang, Y., Yang, D., Roth, H.R. and Xu, D., 2021, September. Swin unetr: Swin transformers for semantic segmentation of brain tumors in mri images. In International MICCAI brainlesion workshop (pp. 272-284). Cham: Springer International Publishing.
> > >
> > > Pang, Y., Liang, J., Huang, T., Chen, H., Li, Y., Li, D., Huang, L. and Wang, Q., 2023. Slim UNETR: Scale hybrid transformers to efficient 3D medical image segmentation under limited computational resources. IEEE Transactions on Medical Imaging.
> > >
> > > Chen, S., Xie, E., Ge, C., Liang, D.Y. and Luo, P., 2022. Cyclemlp: A MLP-like architecture for dense prediction. In International Conference on Learning Representation (ICLR), Oral. IEEE..
> > >
> > > Dosovitskiy, A., Beyer, L., Kolesnikov, A., Weissenborn, D., Zhai, X., Unterthiner, T., Dehghani, M., Minderer, M., Heigold, G., Gelly, S. and Uszkoreit, J., 2020, October. An Image is Worth 16x16 Words: Transformers for Image Recognition at Scale. In International Conference on Learning Representations.
> > >
> > > Simonyan, K., 2014. Very deep convolutional networks for large-scale image recognition. arXiv preprint arXiv:1409.1556. (https://arxiv.org/pdf/1409.1556)
> > >
> > > Valanarasu, J.M.J. and Patel, V.M., 2022, September. Unext: Mlp-based rapid medical image segmentation network. In International conference on medical image computing and computer-assisted intervention (pp. 23-33). Cham: Springer Nature Switzerland.
> > >
> > > Ruan, J., Xie, M., Gao, J., Liu, T. and Fu, Y., 2023, October. Ege-unet: an efficient group enhanced unet for skin lesion segmentation. In International conference on medical image computing and computer-assisted intervention (pp. 481-490). Cham: Springer Nature Switzerland.
> > >
> > > Tan, M. and Le, Q., 2019, May. Efficientnet: Rethinking model scaling for convolutional neural networks. In International conference on machine learning (pp. 6105-6114). PMLR.
> > >
> > > Liu, Y., Zhu, H., Liu, M., Yu, H., Chen, Z. and Gao, J., 2024, March. Rolling-Unet: Revitalizing MLP’s Ability to Efficiently Extract Long-Distance Dependencies for Medical Image Segmentation. In Proceedings of the AAAI Conference on Artificial Intelligence (Vol. 38, No. 4, pp. 3819-3827).
> > >
> > > Jiao, X., Yin, Y., Shang, L., Jiang, X., Chen, X., Li, L., Wang, F. and Liu, Q., 2019. Tinybert: Distilling bert for natural language understanding. arXiv preprint arXiv:1909.10351.
> > >
> > > Hatamizadeh, A., Tang, Y., Nath, V., Yang, D., Myronenko, A., Landman, B., Roth, H.R. and Xu, D., 2022. Unetr: Transformers for 3d medical image segmentation. In Proceedings of the IEEE/CVF winter conference on applications of computer vision (pp. 574-584).

---

### Note · Authors · 2025-01-29

I have read and agree with the venue's withdrawal policy on behalf of myself and my co-authors.